# Learning Survival Distributions with the Asymmetric Laplace Distribution

**Deming Sheng** [1]    **Ricardo Henao** [1]

## Abstract

Probabilistic survival analysis models seek to estimate the distribution of the future occurrence (time) of an event given a set of covariates. In recent years, these models have preferred nonparametric specifications that avoid directly estimating survival distributions via discretization. Specifically, they estimate the probability of an individual event at fixed times or the time of an event at fixed probabilities (quantiles), using supervised learning. Borrowing ideas from the quantile regression literature, we propose a parametric survival analysis method based on the Asymmetric Laplace Distribution (ALD). This distribution allows for closed-form calculation of popular event summaries such as mean, median, mode, variation, and quantiles. The model is optimized by maximum likelihood to learn, at the individual level, the parameters (location, scale, and asymmetry) of the ALD distribution. Extensive results on synthetic and real-world data demonstrate that the proposed method outperforms parametric and nonparametric approaches in terms of accuracy, discrimination and calibration.

## 1. Introduction

Survival models (Nagpal et al., 2021), also known as time-to-event models, are statistical frameworks designed to predict the time until a specific event of interest occurs, given a set of covariates. These models are particularly valuable in situations where the timing of the event is crucial and often subject to *censoring*, which means that in some cases the event has not yet occurred or remains unobserved by the end of the data collection period. The flexibility and adaptability of survival models have led to their widespread application in various fields, including engineering (Lai & Xie, 2006), finance (Gepp & Kumar, 2008), marketing (Jung

[1]Duke University. Correspondence to: Ricardo Henao <ricardo.henao@duke.edu>.

*Proceedings of the $42^{nd}$ International Conference on Machine Learning*, Vancouver, Canada. PMLR 267, 2025. Copyright 2025 by the author(s).

et al., 2012), and, notably, healthcare (Zhang et al., 2017; Voronov et al., 2018; Lánczky & Győrffy, 2021; Emmerson & Brown, 2021).

Survival models can be broadly categorized into parametric, semiparametric, and nonparametric methods, each offering unique strengths depending on the characteristics of the data and the underlying assumptions. Parametric survival models assume that survival times follow a specific statistical distribution, enabling explicit mathematical modeling of the survival function. Common examples include the exponential distribution for constant hazards rates (Feigl & Zelen, 1965), the Weibull distribution for flexible hazards rate modeling (Scholz & Works, 1996), and the log-normal distribution for positively skewed survival times (Royston, 2001). Semiparametric methods, such as the Cox proportional hazards model (Cox, 1972), assume a proportional hazards structure without specifying a baseline hazard distribution, which offers robustness and interpretability. Nonparametric methods, including the Kaplan-Meier estimator (Kaplan & Meier, 1958) and the Nelson-Aalen estimator (Aalen, 1978), rely solely on observed data, avoiding distributional assumptions while directly estimating survival and hazards (risk) functions. More flexible nonparametric models such as Gradient Boosting Machines (GBM) (Dembek et al., 2014) and Random Survival Forests (RSF) (Ishwaran et al., 2008) have also been proposed, which learn survival quantities directly from data using ensemble learning techniques.

More recently, neural networks have significantly advanced survival models across parametric, semiparametric, and nonparametric settings. In parametric methods, LogNorm MLE (Hoseini et al., 2017) improves estimation by fitting a single log-normal distribution using maximum likelihood, while Deep Survival Machines (DSM) (Nagpal et al., 2021) extend this by modeling survival times as mixtures of multiple parametric distributions (*e.g.*, log-normal or Weibull), allowing greater flexibility in capturing heterogeneous risk profiles. Semiparametric approaches, exemplified by DeepSurv (Katzman et al., 2018), integrate neural networks to capture nonlinear relationships while preserving the structure of models such as the Cox proportional hazards model. Nonparametric approaches, such as DeepHit (Lee et al., 2018) and CQRNN (Pearce et al., 2022), leverage deep learning to directly estimate survival functions without relying on traditional assumptions. These advances allow

survival models to handle complex, high-dimensional data with greater precision and flexibility.

Naturally, each approach has limitations that may affect its suitability for different applications. Parametric models rely on strong assumptions about the underlying distribution, which may not accurately capture true survival patterns. Semiparametric models are dependent on the proportional hazards assumption, which can be invalid in certain datasets. Nonparametric models, such as DeepHit and CQRNN, tend to be computationally intensive and require large datasets for effective training, making them less practical in resource-constrained settings. Additionally, these models often produce discrete estimates, which may compromise interpretation and summarization flexibility compared to the continuous modeling offered predominantly by parametric models. To address these limitations, we propose a parametric survival analysis method based on the Asymmetric Laplace Distribution (ALD). Our contributions are listed below.

- We introduce a flexible parametric survival model based on the Asymmetric Laplace Distribution, which offers superior flexibility in capturing diverse survival patterns compared to other distributions (parametric methods).

- The continuous nature of the ALD-based approach offers great flexibility in summarizing distribution-based predictions, thus addressing the limitations of existing discretized nonparametric methods.

- Experiments on 14 synthetic datasets and 7 real-world datasets in terms of 9 performance metrics demonstrate that our proposed framework consistently outperforms both parametric and nonparametric approaches in terms of both discrimination and calibration. These results underscore the robust performance and generalizability of our method in diverse datasets.

## 2. Background

**Survival Data.** A survival dataset $\mathcal{D}$ is represented as a set of triplets $\{(\mathbf{x}_n, y_n, e_n)\}_{n=1}^{N}$, where $\mathbf{x}_n \in \mathbb{R}^d$ denotes the set of covariates in $d$ dimensions, $y_n = \min(o_n, c_n) \in \mathbb{R}_+$ represents the observed time, and $e_n$ is the event indicator. If the event of interest is observed, *e.g.* death, then $o_n < c_n$ and the event indicator is set to $e_n = 1$, otherwise, the event is *censored* and $e_n = 0$. In this work, we make the common assumption that the distributions of observed and censored variables are conditionally independent given the covariates, *i.e.*, $o \perp\!\!\!\perp c \mid \mathbf{x}$. Moreover, while we primarily consider right-censored data, less common types of censoring can be readily implemented (Klein & Moeschberger, 2006), *e.g.*, left-censoring can be data handled by changing the likelihood accordingly (see Section 3.3 for an example of how the maximum likelihood loss proposed here can be adapted for such a case).

**Survival and Hazard Functions.** Survival and hazards functions are two fundamental concepts in survival analysis. The survival function is denoted as $S(t) = P(T > t)$, which represents the probability that an individual has *survived* beyond time $t$. It can also be expressed in terms of the cumulative distribution function (CDF), $F(t)$, which gives the probability that the event has occurred by the time $t$, as $S(t) = 1 - F(t)$. The hazards function, denoted as $\lambda(t)$, describes the instantaneous risk that the event occurs at a specific time $t$, given that the individual has survived up to that point. Formally, it is defined as:

$$\lambda(t) = \lim_{\Delta t \to 0} \frac{P(t \leq T < t + \Delta t | T \geq t)}{\Delta t}.$$

The hazards function is related to the survival function through:

$$\lambda(t) = -\frac{d}{dt} \log S(t), \text{ or } S(t) = \exp\left(-\int_0^t \lambda(u)\, du\right).$$

Furthermore, the probability density function (PDF), $f(t)$, which represents the likelihood that the event occurs at time $t$, can be derived as:

$$f(t) = -\frac{d}{dt} S(t) = \lambda(t) S(t).$$

These relationships establish a unified framework linking $S(t)$, $F(t)$, $\lambda(t)$, and $f(t)$, highlighting their interdependence in survival analysis. Importantly, for the purpose of making predictions, we are interested in distributions *conditioned* on observed covariates, namely $S(t|\mathbf{x})$, $F(t|\mathbf{x})$, $\lambda(t|\mathbf{x})$ and $f(t|\mathbf{x})$.

**Survival Models.** Survival models can be broadly classified into three main categories. Parametric models assume that the survival PDF follows a specific probability distribution as described above. These models thus use a predefined closed-form distribution to describe $f(t|\mathbf{x})$ and $F(t|\mathbf{x})$, for which a model estimating its parameters can be specified. Alternatively, semiparametric models, such as the Cox proportional hazards model (Cox, 1972), first decompose the conditional hazards function as $\lambda(t \mid \mathbf{x}) = \lambda(t)\lambda(\mathbf{x})$, then estimate $\lambda(t)$ from the data and specify a model for $\lambda(\mathbf{x})$. In contrast, nonparametric models, such as DeepHit and CQRNN (Pearce et al., 2022) circumvent directly modeling conditional distributions by discretizing $f(t|\mathbf{x})$ (DeepHit, Lee et al., 2018), learning summaries of $f(t|\mathbf{x})$ such as (a fixed set of) quantiles Pearce et al. (CQRNN, 2022), or even learning to sample from $f(t|\mathbf{x})$ (Chapfuwa et al., 2018). More details can be found in Appendix A.2.

## 3. Methods

### 3.1. Asymmetric Laplace Distribution (ALD)

**Definition 3.1** (Kotz et al. (2012))**.** A random variable $Y$ is said to have an asymmetric Laplace distribution with

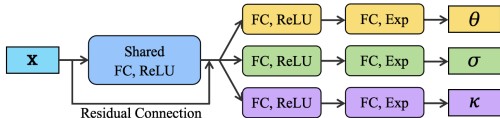

*Figure 1.* The proposed neural network architecture for predicting the parameters of the Asymmetric Laplace Distribution $\mathcal{AL}(\theta, \sigma, \kappa)$. Here, FC denotes the fully connected layer.

parameters $(\theta, \sigma, \kappa)$, if its PDF is:

$$f_{\text{ALD}}(y; \theta, \sigma, \kappa) = \frac{\sqrt{2}}{\sigma} \frac{\kappa}{1+\kappa^2} \begin{cases} \exp\left(\frac{\sqrt{2}\kappa}{\sigma}(\theta - y)\right), & \text{if } y \geq \theta, \\ \exp\left(\frac{\sqrt{2}}{\sigma\kappa}(y - \theta)\right), & \text{if } y < \theta, \end{cases} \quad (1)$$

where $\theta, \sigma > 0$, and $\kappa > 0$, are the location, scale and asymmetry parameters.

Moreover, its CDF can be expressed as:

$$F_{\text{ALD}}(y; \theta, \sigma, \kappa) = \begin{cases} 1 - \frac{1}{1+\kappa^2} \exp\left(\frac{\sqrt{2}\kappa}{\sigma}(\theta - y)\right), & \text{if } y \geq \theta, \\ \frac{\kappa^2}{1+\kappa^2} \exp\left(\frac{\sqrt{2}}{\sigma\kappa}(y - \theta)\right), & \text{if } y < \theta. \end{cases} \quad (2)$$

We denote the distribution of $Y$ as $\mathcal{AL}(y; \theta, \sigma, \kappa)$.

**Corollary 3.2.** *The Asymmetric Laplace Distribution, denoted as $\mathcal{AL}(\theta, \sigma, \kappa)$, can be reparameterized as $\mathcal{AL}(\theta, \sigma, q)$ to facilitate quantile regression (Yu & Moyeed, 2001), where $q \in (0, 1)$ is the percentile parameter that represents the desired quantile. The relationship between $q$ and $\kappa$ is given by $q = \kappa^2/(\kappa^2 + 1)$.*

Additional details are provided in Appendix A.1.

### 3.2. Model for the ALD

The structure of the proposed model is illustrated in Figure 1, where a shared encoder is followed by three independent heads to estimate the parameters $\theta$, $\sigma$, and $\kappa$ of the ALD distribution. For the purpose of the experiments in Section 5 with structured data, we use fully connected layers with ReLU activation functions. The outputs of the model connected to $\theta$, $\sigma$ and $\kappa$ are further constrained to be non-negative through an exponential (Exp) activation. In addition, a residual connection is included to enhance gradient flow and improve model stability. See Appendix B.3 for more details about the architecture of the model.

### 3.3. Learning for the ALD

We propose learning the model for the ALD through maximum likelihood estimation (MLE). Since the event of interest can be either observed or censored, we specify separate objectives for these two types of data. For observed events, for which $e = 1$, we directly seek to optimize the parameters of the model to maximize $f_{\text{ALD}}(t|\mathbf{x})$ in (1). Alternatively,

for censored events, for which $e = 0$, we optimize the parameters of the model to maximize the survival function $S_{\text{ALD}}(t|\mathbf{x}) = 1 - F_{\text{ALD}}(t|\mathbf{x})$ in (2). In this manner, the ALD objective below accounts for both the occurrence of events and their respective timing while explicitly incorporating the survival probability constraint for censored data:

$$-\mathcal{L}_{\text{ALD}} = \sum_{n \in \mathcal{D}_{\text{O}}} \log f_{\text{ALD}}(y_n \mid \mathbf{x}_n)$$
$$+ \sum_{n \in \mathcal{D}_{\text{C}}} \log S_{\text{ALD}}(y_n \mid \mathbf{x}_n), \quad (3)$$

where $\mathcal{D}_{\text{O}}$ and $\mathcal{D}_{\text{C}}$ are the subsets of $\mathcal{D} = \mathcal{D}_{\text{O}} \cup \mathcal{D}_{\text{C}}$ for which $e = 1$ and $e = 0$, respectively. Detailed derivations of the objective in (3) be found in Appendix A.1.

The simplicity of the objective in (3) is a consequence of the ability to write the relevant distributions, $f_{\text{ALD}}(t|\mathbf{x})$ and $S_{\text{ALD}}(t|\mathbf{x})$, in closed form. Moreover, we make the following remarks.

- The objective in (3) has the same form as the one used in other parametric approaches, for instance Royston (2001) for the log-normal distribution.

- We can readily adapt the loss for other forms of censoring, for instance, if events are left censored, we only have to replace the second term of (3) by $1 - S_{\text{ALD}}(t|\mathbf{x})$.

- We do not consider additional loss terms, as is usually done for other approaches, *e.g.*, DeepHit optimizes a form similar to (3), where the density function and cumulative distribution are replaced by discretized approximations, but also consider an additional loss term to improve discrimination (Lee et al., 2018).

- Although the ALD in (1) has support for $t < 0$, we have observed empirically that this is unlikely to happen, as we will demonstrate in the experiments.

### 3.4. Comparison between our Method and CQRNN

CQRNN (Pearce et al., 2022) adopts the widely-used objective for quantile regression, which is also based on the Asymmetric Laplace Distribution $\mathcal{AL}(\theta, \sigma, q)$, and uses the transformation in Corollary 3.2. Specifically, they use the maximum likelihood estimation approach to optimize the following objective:

$$\mathcal{L}_{\text{QR}}(y; \theta_q, \sigma, q) = \log \sigma - \log[q(1-q)]$$
$$+ \frac{1}{\sigma} \begin{cases} q(y - \theta_q), & \text{if } y \geq \theta_q, \\ (1-q)(\theta_q - y), & \text{if } y < \theta_q. \end{cases} \quad (4)$$

Following the quantile regression framework, their approach optimizes a model to predict $\theta_q$ for a predefined collection of quantile values, *e.g.*, $q = \{0.1, 0.2, \ldots, 0.9\}$. Effectively and similarly to ours, Pearce et al. (2022) specify a shared

encoder with multiple heads to predict $\{\theta_q\}_q$. Note that the objective in (4) does not require one to specify $\sigma$, which results in the following simplified loss:

$$\mathcal{L}_{\text{QR}}(y; \theta_q, q) = \begin{cases} q(y - \theta_q), & \text{if } y \geq \theta_q, \\ (1 - q)(\theta_q - y), & \text{if } y < \theta_q, \end{cases}$$
$$= (y - \theta_q)(q - \mathbb{I}[\theta_q > y]), \quad (5)$$

where $\mathbb{I}[\cdot]$ is the indicator function. The formulation in (5) is also known as the *pinball* or *checkmark* loss (Koenker & Bassett Jr, 1978), which is widely used in the quantile regression literature.

Importantly, unlike in the objective for our approach in (3), CQRNN does not maximize the survival probability directly. Instead, they adopt the also widely used approach based on the Portnoy's estimator (Neocleous et al., 2006), which optimizes an objective function tailored for censored quantile regression. Specifically, this approach introduces a re-weighting scheme to handle the censored data:

$$\mathcal{L}_{\text{CQR}}(y, y^*; \theta_q, q, w) = w\mathcal{L}_{\text{QR}}(y; \theta_q, q) \\ + (1 - w)\mathcal{L}_{\text{QR}}(y^*; \theta_q, q), \quad (6)$$

where where $y^*$ is a *pseudo* value set to be "sufficiently" larger than all the observed values of $y$ in the data. Specifically, in CQRNN (Pearce et al., 2022) it is defined as $y^* = 1.2 \max_i y_i$. However, Portnoy (Neocleous et al., 2006) indicates that $y^*$ could be set to any sufficiently large value approximating $\infty$. For example, Koenker (2022) sets $y^* = 1e6$. This means that in practice, this parameter often requires careful tuning based on the specific dataset, provided that different datasets exhibit varying levels of sensitivity to it. In some cases, we have observed that small perturbations in $y^*$ can lead to considerable variation on performance metrics. Consequently, optimizing this parameter can be non-trivial, making the use of CQRNN, and other censored quantile regression methods, challenging.

The other parameter in (6) that requires attention is the weight $w \in (0, 1)$, which is defined as $w = (q - q_c)/(1 - q_c)$, and where $q_c$ is the quantile at which the data point was censored ($e = 0, y = c$), with respect to the observed value distribution, *i.e.*, $p(o < c|x)$. The challenge is that $q_c$ is not known in practice. To address this issue, CQRNN proposes two strategies: a sequential grid algorithm and the quantile grid output algorithm. The core idea of both strategies is to approximate $q_c$ using the proportion $q$ corresponding to the quantile that is closest to the censoring value $c$ using the distribution of observed events $y$, which are readily available. Even with this approach, $q_c$ is an inherently inaccurate approximation. Its precision heavily depends on the initial grid of $q$ values, specifically, the intervals between consecutive $q$ values. Consequently, smaller intervals provide finer granularity, but increased computational costs, while larger

intervals may lead to coarser approximations that tend to affect model performance. This means that in some cases, the model is sensitive to the choice of the grid of $q$ values.

In contrast, our approach enjoys a simple objective function resulting in parametric estimates of several distribution summaries such as mean, median, standard deviation, and quantiles without additional cost. Additional details of CQRNN are provided for completeness in Appendix A.2.

## 4. Related Work

Survival analysis is a fundamental area of study in statistics and machine learning, focusing on modeling time-to-event data while accounting for censoring. A wide range of models has been developed that span parametric, semiparametric, and nonparametric methods.

Parametric models assume a specific distribution for the time-to-event variable, providing a structured approach to modeling survival and hazard functions. Commonly used distributions include the exponential (Feigl & Zelen, 1965), Weibull (Scholz & Works, 1996), and log-normal (Royston, 2001) distributions, as well as more flexible formulations based on mixtures of these distributions (Nagpal et al., 2021). For example, the log-normal model assumes that the logarithm of survival times follows a normal distribution, enabling straightforward parameterization of survival curves. In modern approaches (Hoseini et al., 2017), neural networks are employed to learn the parameters of the assumed distribution, *e.g.*, the mean and variance for the log-normal. This combination allows the model to leverage the power of neural networks to capture complex, nonlinear relationships between covariates and survival times, while keeping the interpretability and structure inherent to the parametric framework. However, these models face challenges despite their simplicity when the true event distribution significantly deviates from that assumed.

Semiparametric methods strike a balance between flexibility and interpretability. One notable example is the Cox proportional hazards model (Cox, 1972), which assumes a multiplicative effect of covariates on the hazard function. Building on this foundation, DeepSurv (Katzman et al., 2018), a deep learning-based extension, replaces the linear assumption with neural network architectures to model complex feature interactions. DeepSurv has demonstrated improved performance in handling high-dimensional covariates while maintaining the interpretability of hazard ratios. However, semiparametric models face challenges in effectively handling censored data, particularly when censoring rates are very high. In such cases, the limited amount of usable information can lead to degraded performance and reduced reliability of the model's estimates.

The Kaplan–Meier (KM) estimator (Kaplan & Meier, 1958)

is a widely used nonparametric method for survival analysis. KM estimates the survival function directly from the data without assuming any underlying distribution. The KM estimator is particularly effective for visualizing survival curves and computing survival probabilities. However, its inability to incorporate covariates limits its applicability in complex scenarios. To address this, more advanced nonparametric approaches have been developed. Tree-based ensemble methods such as Random Survival Forests (RSF) (Ishwaran et al., 2008) and Gradient Boosting Machines (GBM) (Dembek et al., 2014) extend traditional nonparametric modeling by incorporating covariate information while avoiding strict distributional assumptions. These models are capable of capturing nonlinear relationships and interactions, and provide robust survival estimates even in the presence of high-dimensional features. In parallel, neural network-based models such as DeepHit (Lee et al., 2018) and CQRNN (Pearce et al., 2022) have emerged, which directly estimate survival probabilities or quantiles without predefined hazard or survival function forms. These methods offer high flexibility and are particularly effective in modeling complex, high-dimensional, and heterogeneous datasets. Nevertheless, a notable shortcoming of these models (*e.g.*, DeepHit and CQRNN) is that they produce piecewise constant or point-mass distribution estimates, respectively, which lack continuity and smoothness. This can lead to survival estimates that are difficult to summarize, interpret, and use in downstream analysis.

## 5. Experiments

### 5.1. Datasets

We utilize two types of datasets, following Pearce et al. (2022): (Type 1) synthetic data with synthetic censoring and (Type 2) real-world data with real censoring. Table 1 presents a summary of general statistics for all datasets. To account for training and model initialization variability, we run all experiments 10 times with random splits of the data with partitions consistent with Table 1. The source code required to reproduce the experiments presented in this paper is available at: `https://github.com/demingsheng/ALD`.

For synthetic observed data with synthetic censoring, the input features $\mathbf{x}$ are generated uniformly as $\mathbf{x} \sim \mathcal{U}(0, 2)^d$, where $d$ represents the number of features. The observed variable $o \sim p(o|\mathbf{x})$ and the censored variable $c \sim p(c|\mathbf{x})$ follow distinct distributions, with each distribution parameterized differently, depending on the specific dataset configuration. This variability in distributions and parameters allows for the evaluation of the model's robustness under diverse synthetic data scenarios.

For real target data with real censoring, we utilize datasets that span various domains, characterized by distinct features,

*Table 1.* Dataset summaries: number of features (Feats), training/test data size, and proportion of censored events (PropCens).

| Dataset | Feats | Train data | Test data | PropCens |
|---|---|---|---|---|
| **Type 1 – Synthetic target data with synthetic censoring** | | | | |
| Norm linear | 1 | 500 | 1000 | 0.20 |
| Norm non-linear | 1 | 500 | 1000 | 0.24 |
| Exponential | 1 | 500 | 1000 | 0.30 |
| Weibull | 1 | 500 | 1000 | 0.22 |
| LogNorm | 1 | 500 | 1000 | 0.21 |
| Norm uniform | 1 | 500 | 1000 | 0.62 |
| Norm heavy | 4 | 2000 | 1000 | 0.80 |
| Norm med | 4 | 2000 | 1000 | 0.49 |
| Norm light | 4 | 2000 | 1000 | 0.25 |
| Norm same | 4 | 2000 | 1000 | 0.50 |
| LogNorm heavy | 8 | 4000 | 1000 | 0.75 |
| LogNorm med | 8 | 4000 | 1000 | 0.52 |
| LogNorm light | 8 | 4000 | 1000 | 0.23 |
| LogNorm same | 8 | 4000 | 1000 | 0.50 |
| **Type 2 – Real target data with real censoring** | | | | |
| METABRIC | 9 | 1523 | 381 | 0.42 |
| WHAS | 6 | 1310 | 328 | 0.57 |
| SUPPORT | 14 | 7098 | 1775 | 0.32 |
| GBSG | 7 | 1785 | 447 | 0.42 |
| TMBImmuno | 3 | 1328 | 332 | 0.49 |
| BreastMSK | 5 | 1467 | 367 | 0.77 |
| LGGGBM | 5 | 510 | 128 | 0.60 |

sample sizes, and censoring proportions. Four of these datasets: METABRIC, WHAS, SUPPORT, and GBSG, were retrieved from the DeepSurv GitHub repository[1]. Other details are available in Katzman et al. (2018). The remaining three datasets: TMBImmuno, BreastMSK, and LGGGBM were sourced from cBioPortal[2] for Cancer Genomics. These datasets constitute a diverse benchmark across domains such as oncology and cardiology, allowing a comprehensive evaluation of survival analysis methods. Additional details of all datasets can be found in Appendix B.1.

### 5.2. Metrics

**Predictive Accuracy Metrics**: Mean Absolute Error (MAE) and Integrated Brier Score (IBS) (Graf et al., 1999), measure the accuracy of survival time predictions. MAE quantifies the average magnitude of errors between predicted and observed survival times $\tilde{y}_i$ and $y_i$, respectively. For synthetic data, ground truth values are obtained directly from the observed distribution, while for real data, only observed events ($e = 1$) are considered. For the IBS calculation, we select 100 time points evenly from the 0.1 to 0.9 quantiles of the distribution for $y$ in the training set.

**Concordance Metrics**: Harrell's C-Index (Harrell et al., 1982) and Uno's C-Index (Uno et al., 2011), which evaluate the ability of the model to correctly order survival times in a pairwise manner, while accounting for censoring. Harrell's C-Index is known to be susceptible to bias, when the censoring rate is high. This happens because censoring dominates

---

[1]https://github.com/jaredleekatzman/DeepSurv/
[2]https://www.cbioportal.org/

the pairwise ranking when estimating the proportion of correctly ordered event pairs. Alternatively, Uno's C-Index adjusts for censoring by using inverse probability weighting, which provides a more robust estimate when the proportion of censored events is high.

**Calibration Metrics**: There are several metrics to assess calibration. We consider summaries (slope and intercept) of the calibration curves using the predicted PDF $f(t|\mathbf{x})$ or the survival distribution $S(t|\mathbf{x})$. Moreover, we use the censored D-Calibration (CensDcal) (Haider et al., 2020). For the former, $\text{Cal}[f(t \mid \mathbf{x})]$, 9 prediction interval widths are considered, *e.g.*, 0.1 for $[0.45, 0.55]$, 0.2 for $[0.4, 0.6]$, *etc*. These are used to define the time ranges for each prediction, after which we calculate the proportion of test events that fall within each interval. The calculation of the proportion of censored and observed cases follows the methodology in Goldstein et al. (2020), with further details provided in Appendix B.2. This calibration curve of expected *vs.* observed events is summarized with an ordinary least squares linear fit parameterized by its *slope* and *intercept*. A well-calibrated model is expected to have a unit slope and a zero intercept. For the survival distribution, $\text{Cal}[S(t|\mathbf{x})]$, we follow a similar procedure, however, we consider 10 non-overlapping intervals in the range $(0, 1)$, *i.e.*, $(0, 0.1]$, $(0.1, 0.2]$, *etc* and then calculating the proportion of test events that fall within each interval. The calculation of CensDcal starts with that of $\text{Cal}[S(t|\mathbf{x})]$, which is followed by computing the sum of squared residuals between the observed and expected proportions, *i.e.*, 0.1 for the 10 intervals defined above.

These three groups of metrics provide a robust framework for evaluating predictive accuracy, calibration, and concordance in survival analysis. For the results we calculate averages and standard deviations for all metrics over 10 random test sets. The metrics that require a point estimate, *i.e.*, MAE and C-Index are obtained using the expected value of $f(t|\mathbf{x})$, which can be calculated in closed form. More details about all metrics can be found in Appendix B.2.

### 5.3. Baselines

We compare the proposed method against eight baselines representative of related work to evaluate performance and effectiveness. **LogNorm** (Royston, 2001): A classical parametric survival model that assumes event times follow a log-normal distribution. **DSM (LogNorm / Weibull)** (Nagpal et al., 2021): A neural parametric model that represents survival times as mixtures of either log-normal or Weibull distributions, allowing flexible modeling of complex event-time distributions and adaptable hazard dynamics through deep learning. **DeepSurv** (Katzman et al., 2018): A semi-parametric survival model based on the Cox proportional hazards framework, leveraging neural networks to model nonlinear covariate effects on the hazard function. **RSF** (Ish-

waran et al., 2008): A nonparametric ensemble model that builds multiple decision trees to estimate survival functions without strong distributional assumptions. **GBM** (Dembek et al., 2014): A tree-based gradient boosting model adapted for survival analysis, capable of modeling complex nonlinearities in covariates. **DeepHit** (Lee et al., 2018): A deep learning-based survival model that predicts piece-wise probability distributions over event times using a fully neural network architecture. **CQRNN** (Pearce et al., 2022): A censored quantile regression model that employs a neural network architecture, and whose objective is based on the Asymmetric Laplace Distribution. Together, these baselines span parametric, semi-parametric, and nonparametric survival modeling paradigms and include both traditional statistical models and contemporary neural architectures, thereby providing a comprehensive benchmark for performance evaluation. The implementation details, including model selection, of our method and the other baselines can be found in Appendix B.3.

### 5.4. Results

Table 2 provides a comprehensive summary of the comparisons between our model and the eight baselines in 21 datasets and 9 evaluation metrics, which is 189 comparisons in total. When assessing the statistical significance of the different metrics we use a Student's $t$ test with $p < 0.05$ considered significant after correction for false discovery rate using Benjamini-Hochberg (Benjamini & Hochberg, 1995). These results underscore several key insights:

**Overall Superiority**: Our model is significantly better than the baselines consistently more often. For instance, our model significantly outperforms CQRNN in 23% of the comparisons while the opposite only occurs 12%, and these proportions are higher for the comparisons against the other baselines, namely, 41%, 52%, 55%, 60%, 69%, 71% and 73% for DeepSurv, GBM, RSF, LogNorm, DSM (Weibull), DSM (LogNorm) and DeepHit, respectively.

**Accuracy:** Our model demonstrates strong improvements in predictive accuracy. In particular, it achieves superior MAE performance on 10 out of 21 datasets relative to LogNorm, 11 datasets compared to GBM, DSM (LogNorm), and DSM (Weibull), and 12 datasets compared against DeepHit. Although performance in MAE is mixed when compared to CQRNN and DeepSurv, our method still performs competitively. Moreover, it consistently outperforms the baselines on nearly every dataset when evaluated with the IBS metric. This consistent superiority in IBS underscores our model's ability to provide accurate and reliable predictions over the entire time range, not just at specific time points. Table 4 and Figure 4 in the Appendix further support this, showing that our method achieves significantly lower IBS values, which reflects its effectiveness in learning from censored data without exacerbating bias in survival estimates.

*Table 2.* Summary of benchmarking results across 21 datasets. Each column group shows three figures: the number of datasets where our method significantly outperforms, underperforms or is comparable with the baseline indicated. The last two rows summarize the column totals and proportions to simplify the comparisons. For reference, the total number of comparisons is 189.

| Metric | *vs.* CQRNN | | | *vs.* LogNorm | | | *vs.* DeepSurv | | | *vs.* DeepHit | | |
|---|---|---|---|---|---|---|---|---|---|---|---|---|
| | Better | Worse | Same | Better | Worse | Same | Better | Worse | Same | Better | Worse | Same |
| MAE | 6 | 8 | 7 | 10 | 3 | 8 | 6 | 8 | 7 | 12 | 6 | 3 |
| IBS | 19 | 1 | 1 | 21 | 0 | 0 | 21 | 0 | 0 | 21 | 0 | 0 |
| Harrell's C-Index | 4 | 2 | 15 | 10 | 3 | 8 | 6 | 2 | 13 | 15 | 0 | 6 |
| Uno's C-Index | 2 | 3 | 16 | 9 | 2 | 10 | 6 | 1 | 14 | 15 | 0 | 6 |
| CensDcal | 8 | 4 | 9 | 10 | 1 | 10 | 8 | 5 | 8 | 15 | 1 | 5 |
| Cal $[S(t|\mathbf{x})]$(Slope) | 0 | 0 | 21 | 15 | 0 | 6 | 13 | 0 | 8 | 12 | 0 | 9 |
| Cal $[S(t|\mathbf{x})]$(Intercept) | 0 | 0 | 21 | 14 | 0 | 7 | 0 | 11 | 10 | 16 | 0 | 5 |
| Cal $[f(t|\mathbf{x})]$(Slope) | 4 | 0 | 17 | 14 | 0 | 7 | 9 | 0 | 12 | 14 | 0 | 7 |
| Cal $[f(t|\mathbf{x})]$(Intercept) | 0 | 4 | 17 | 10 | 0 | 11 | 8 | 0 | 13 | 18 | 0 | 3 |
| Total | 43 / 189 | 22 / 189 | 124 / 189 | 113 / 189 | 9 / 189 | 67 / 189 | 77 / 189 | 27 / 189 | 85 / 189 | 138 / 189 | 7 / 189 | 44 / 189 |
| Proportion | 0.228 | 0.116 | 0.656 | 0.598 | 0.048 | 0.354 | 0.407 | 0.143 | 0.450 | 0.730 | 0.037 | 0.233 |

| Metric | *vs.* GBM | | | *vs.* RSF | | | *vs.* DSM (LogNorm) | | | *vs.* DSM (Weibull) | | |
|---|---|---|---|---|---|---|---|---|---|---|---|---|
| | Better | Worse | Same | Better | Worse | Same | Better | Worse | Same | Better | Worse | Same |
| MAE | 11 | 7 | 3 | 9 | 6 | 6 | 11 | 6 | 4 | 11 | 5 | 5 |
| IBS | 17 | 1 | 3 | 14 | 2 | 5 | 19 | 1 | 1 | 19 | 1 | 1 |
| Harrell's C-Index | 14 | 2 | 5 | 16 | 2 | 3 | 17 | 0 | 4 | 15 | 0 | 6 |
| Uno's C-Index | 13 | 1 | 7 | 14 | 2 | 5 | 16 | 0 | 5 | 14 | 0 | 7 |
| CensDcal | 0 | 2 | 19 | 6 | 4 | 11 | 21 | 0 | 0 | 21 | 0 | 0 |
| Cal $[S(t|\mathbf{x})]$(Slope) | 6 | 0 | 15 | 12 | 0 | 9 | 12 | 0 | 9 | 16 | 0 | 5 |
| Cal $[S(t|\mathbf{x})]$(Intercept) | 12 | 0 | 9 | 10 | 0 | 11 | 13 | 0 | 8 | 17 | 0 | 4 |
| Cal $[f(t|\mathbf{x})]$(Slope) | 14 | 0 | 7 | 15 | 0 | 6 | 13 | 0 | 8 | 11 | 0 | 10 |
| Cal $[f(t|\mathbf{x})]$(Intercept) | 11 | 0 | 10 | 8 | 0 | 13 | 12 | 0 | 9 | 6 | 0 | 15 |
| Total | 98 / 189 | 13 / 189 | 78 / 189 | 104 / 189 | 16 / 189 | 69 / 189 | 134 / 189 | 7 / 189 | 48 / 189 | 130 / 189 | 6 / 189 | 53 / 189 |
| Proportion | 0.519 | 0.069 | 0.413 | 0.550 | 0.085 | 0.365 | 0.709 | 0.037 | 0.254 | 0.688 | 0.032 | 0.280 |

**Concordance:** Our model shows performance comparable to CQRNN for Harrell's and Uno's C-indices, yielding statistically indistinguishable results on most datasets (15 and 16 ties, respectively). However, when compared to a broader set of baselines, including LogNorm, GBM, RSF, DeepHit, and DSM variants, our method consistently achieves greater concordance. In particular, for Harrell's C-index, our model attains better results on 10 datasets compared to LogNorm, 14 compared to GBM, 15 compared to DeepHit and DSM (LogNorm), 16 compared to RSF, and 17 datasets compared to DSM (Weibull), respectively. A similar trend is evident for Uno's C-index, with our method outperforming Log-Norm on 9 datasets, GBM on 13, RSF and DSM (Weibull) on 14, DeepHit on 15, and DSM (LogNorm) on 16 datasets. These findings underscore the robustness of our model in capturing relative risk and generating accurate survival time rankings across a variety of baselines, ranging from classical parametric approaches to contemporary neural methods.

**Calibration**: Our model demonstrates strong performance in calibration metrics, particularly in CensDcal, Cal$[S(t|\mathbf{x})]$ (slope) and Cal$[f(t|\mathbf{x})]$ (slope). The results for CensDcal highlight its ability to effectively handle censored observations. Furthermore, our model shows significant superiority in slope- and intercept-related metrics, particularly compared to LogNorm, DeepHit, RSF, and DSM variants. However, the improvement over CQRNN remains relatively

subtle. These results indicate that our model achieves a better calibration of the predicted survival probabilities, ensuring a closer alignment between predictions and observed outcomes in both the survival CDF $S(t|\mathbf{x})$ and PDF $f(t|\mathbf{x})$. This underscores the reliability and robustness of our model in accurately capturing true survival behavior across diverse datasets.

For illustration purposes, Figure 2 shows the performance of all models and datasets using Harrell's C-index and CensD-cal. Specifically, Figure 2(a) provides a comparison of Harrell's C-index, highlighting the discriminative performance of our proposed model, ALD, alongside other baseline models. The x axis lists all the datasets, both synthetic datasets (*e.g.*, Gaussian linear, exponential, *etc.*) and real-world datasets (*e.g.*, METABRIC, WHAS, *etc.*), while the y axis indicates the range of corresponding mean C-index values, with error bars representing standard deviation across 10 model runs. Our method demonstrates consistently strong performance across both synthetic and real-world datasets, frequently achieving higher or comparable C-index values in relation to the baseline models. Among these, CQRNN stands out as the most competitive alternative, achieving similar levels of performance on certain datasets. However, in most cases, our model outperforms CQRNN, reflecting its robustness and superior discriminative capability. In particular, ALD excels under scenarios with high censoring

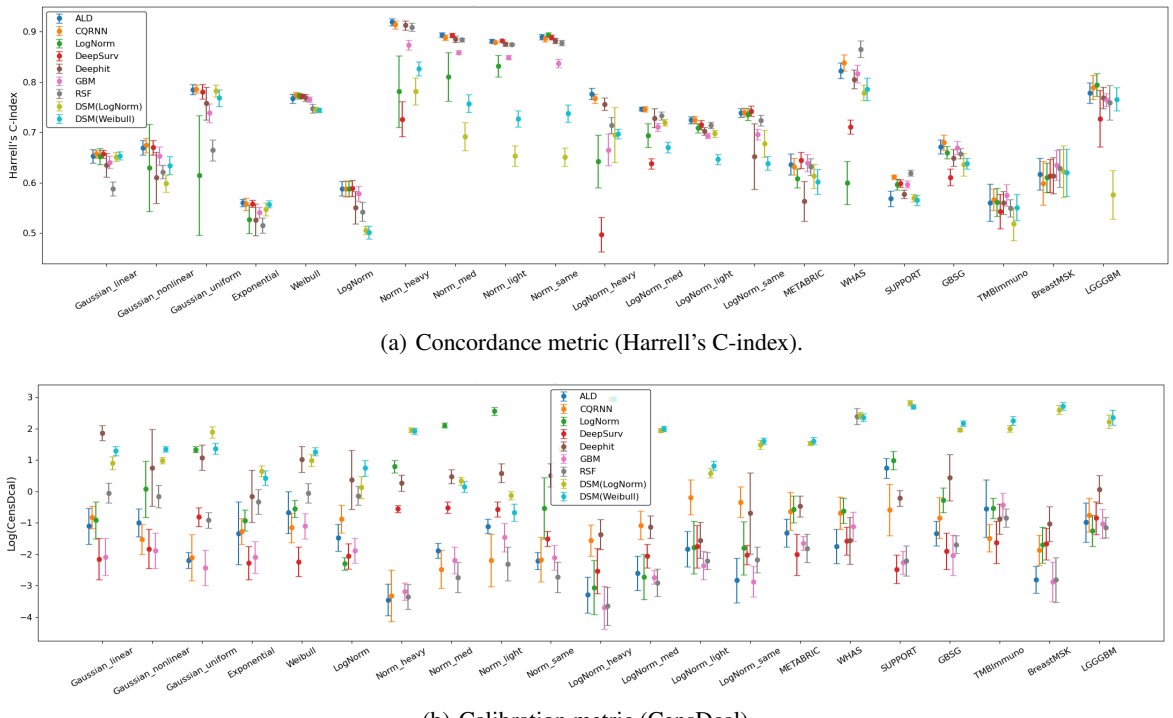

(a) Concordance metric (Harrell's C-index).

(b) Calibration metric (CensDcal).

*Figure 2.* Performance on discrimination and calibration metrics. (a) concordance and (b) calibration. Reported are test averages with standard deviations over 10 runs.

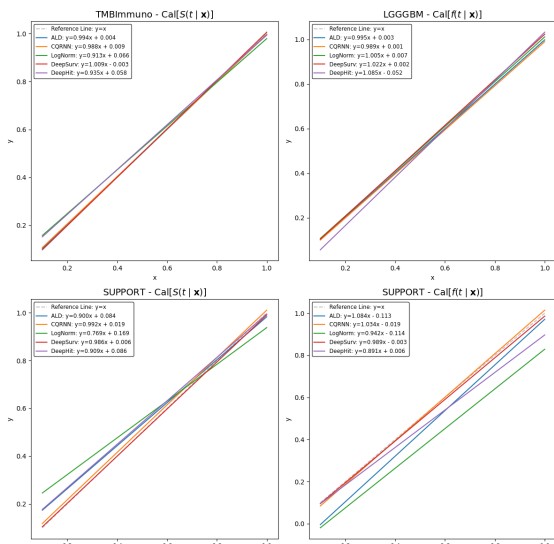

*Figure 3.* Examples of best and worst calibration curves. Slope and intercept of the linear fit are shown in the legend.

rates. For example, on Norm heavy (PropCens: 0.80), Norm med (PropCens: 0.49), LogNorm heavy (PropCens: 0.75), and LogNorm med (PropCens: 0.52), ALD consistently outperforms other models. This performance highlights ALD's ability to effectively handle challenging scenarios. Such robustness under high censorship further underscores ALD's reliability and adaptability in various survival analysis tasks.

Using a similar comparison framework, Figure 2(b) presents the calibration results using CensDcal. Concisely, the proposed model achieves consistently better (lower) CensDcal figures across most datasets, reflecting superior calibration performance compared to the baseline models.

Complementary to the CensDcal calibration metric, the slope and intercept summaries of the calibration curve provide a more intuitive (and graphical) perspective of the calibration results. Figure 3 presents the best (first row) and worst (second row) results from our model on real-world data. For comparative analysis, we include results from CQRNN, LogNorm, DeepSurv, and DeepHit. The left and right columns represent the curves for $\text{Cal}[S(t|\mathbf{x})]$ and for $\text{Cal}[f(t|\mathbf{x})]$, respectively. The gray dashed line represents the idealized result for which the slope is one and the intercept is zero. A full set of calibration visualizations across all datasets and methods is provided in Appendix C.2.

The proposed model demonstrates exceptional performance on the TMBImmuno dataset for $\text{Cal}[f(S|\mathbf{x})]$ summaries, as well as on the LGGGBM dataset for $\text{Cal}[f(t|\mathbf{x})]$ summaries indicating robust calibration across both versions of the calibration metrics. In contrast, the performance on the SUPPORT dataset is relatively weaker. This discrepancy can largely be attributed to our method's reliance on the assumption of the ALD, which may not be appropriate across all datasets. This limitation is particularly evident in datasets

like SUPPORT. Notably, the SUPPORT data exhibit high skewness with a relatively small range of $y$. Specifically, Figure 12 in the Appendix shows the event distribution for the SUPPORT data, from which we can see that it is heavily skewed. Such an skewness manifested as the concentration of events close to 0 makes it challenging to achieve good calibration in that range, *i.e.*, $t \to 0$. Similarly, our method attempted to predict smaller values for the initial quantiles but still allocated a disproportionately large weight to the first two intervals because of the small and highly concentrated predicted quantiles, which significantly reduced the capacity for the remaining intervals and ultimately degraded calibration performance. However, the calibration results for this dataset remain within reasonable ranges and more importantly, comparable to those from the baselines. Detailed results for all datasets are provided in Appendix C.1. Overall and consistent with the summary results in Table 2, our model demonstrates a clear advantage on the slope and intercept metrics, consistently achieving better performance compared to the baselines.

**Case Studies** To further support our empirical findings, we include additional case studies in Appendix C.3, providing a robustness analysis and a deeper understanding of the behavior of our method in various scenarios.

*Case Study 1: Robustness under High Censoring and Quantile Extremes.* We assess the robustness of our model under varying levels of censoring and at key survival quantiles (25%, 50%, 75%), following the experimental protocol of Nagpal et al. (2021). Figures 7 and 8 report time-dependent concordance scores for the METABRIC and SUPPORT datasets, which exhibit distinct censoring rates and distributional properties. Our method maintains strong performance across all settings and demonstrates particular robustness at higher quantiles on METABRIC, where censoring and skewness are more severe. Although performance is comparatively lower on SUPPORT, which likely reflects its deviation from the ALD assumption, our model remains comparable to DSM-based baselines, indicating stable behavior even under distributional mismatch.

*Case Study 2: Capturing Diverse Survival Patterns.* To evaluate the ability of our model to capture various survival behaviors, we perform a clustering analysis on the predicted cumulative distribution functions. Figure 9 presents six representative patterns obtained through $K$-means clustering on the estimated parameters from our method and DeepHit on seven real-world datasets. Although both models capture multiple behaviors, the CDFs produced by DeepHit tend to converge to one at 120% of the maximum observed event time ($1.2 \max_i y_i$), thus suggesting that all individuals eventually experience the event by that time horizon, which in practice is unlikely. In contrast, our model better reflects long-term survival characteristics by keeping

$F_{\mathrm{ALD}}(1.2 \max_i y_i | \mathbf{x}) < 1$ for some $\mathbf{x}$. We further validate this behavior using synthetic datasets with known ground-truth CDFs. Figures 10 and 11 compare the worst-estimated instances within each cluster for our method and Deep-Hit. Across clusters, our model consistently achieves lower Wasserstein distances, demonstrating superior accuracy and robustness in modeling heterogeneous survival distributions.

*Case Study 3: Alternative Distribution Summaries.* We also explore other distribution summaries, *i.e.*, the mode and median, to evaluate their impact on the performance of MAE and C-index. As shown in Table 5, different summaries can offer improved performance on specific datasets, demonstrating the flexibility of our probabilistic formulation to adapt the summary statistic depending of the downstream evaluation needs.

*Case Study 4: Empirical Behavior of $F_{ALD}(0|\mathbf{x})$.* Finally, recognizing that the ALD has support for $t < 0$, we summarized the empirical quantiles of the predicted $F_{\mathrm{ALD}}(0|\mathbf{x})$, *i.e.*, the probability that events occur up to $t = 0$. Interestingly, Table 6 indicates that this is not an issue as in most cases $F_{\mathrm{ALD}}(0|\mathbf{x}) \to 0$ for the majority of the predictions made by the model on the test set.

## 6. Conclusion

In this paper, we proposed a parametric survival model based on the ALD and provided a comprehensive comparison and analysis with existing methods, particularly CQRNN, which uses the same distribution. Our model produces closed-form distributions, which enables flexible summarization and interpretation of predictions. Experimental results on a diverse range of synthetic and real-world datasets demonstrate that our approach offers very competitive performance in relation to multiple baselines across accuracy, concordance, and calibration metrics.

**Limitations.** First, our method relies on the assumption of the ALD, which may not be universally applicable. This limitation was particularly evident in certain cases, such as with the SUPPORT dataset, as highlighted in Section 5.4, where the performance of our method faced challenges, especially in terms of calibration. Second, while our approach facilitates the calculation of different distribution metrics, such as mean, median, mode, and even distribution quantiles, selecting the most suitable summary statistic for specific datasets or applications remains a non-trivial task. In this study, we selected the mean as the main summary statistic, which results in relatively balanced performance metrics. However, it does not offer an advantage, for example, in terms of C-index and MAE when compared to CQRNN. Nevertheless, considering other summary statistics as part of model selection, which we did not attempt, may improve performance on these metrics for certain datasets, as detailed in Appendix C.3.

## Acknowledgments

This work was supported by grant 1R61-NS120246-02 from the National Institute of Neurological Disorders and Diseases (NINDS).

## Impact Statement

The proposed survival analysis method utilizes the Asymmetric Laplace Distribution (ALD) to deliver closed-form solutions for key event summaries, such as means and quantiles, facilitating more interpretable predictions. The method outperforms both traditional parametric and nonparametric approaches in terms of discrimination and calibration by optimizing individual-level parameters through maximum likelihood. This advancement has significant implications for applications like personalized medicine, where accurate and interpretable predictions of event timing are crucial.

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

# A. Analytical Results

This section provides the analytical results. Detailed proofs for the Asymmetric Laplace Distribution Loss can be found in Appendix A.1, while the analysis of all the baselines, including CQRNN, LogNormal MLE, DeepSurv, and DeepHit, is presented in Appendix A.2.

## A.1. Proofs for the Asymmetric Laplace Distribution Loss

**Theorem 1.** If $Y \sim \mathcal{AL}(\theta, \sigma, \kappa)$, where $\mathcal{AL}$ denotes the Asymmetric Laplace Distribution with location parameter $\theta$, scale parameter $\sigma > 0$, and asymmetry parameter $\kappa > 0$, then the ALD loss is given by:

$$\mathcal{L}_{\text{ALD}} = \mathcal{L}_{\text{o}}(y; \theta, \sigma, \kappa) + \mathcal{L}_{\text{c}}(y; \theta, \sigma, \kappa) = -\sum_{n \in \mathcal{D}_{\text{o}}} \log f_{\text{ALD}}(y_n \mid \mathbf{x}_n) - \sum_{n \in \mathcal{D}_{\text{c}}} \log\left(1 - F_{\text{ALD}}(y_n \mid \mathbf{x}_n)\right), \tag{7}$$

where $\mathcal{D}_{\text{O}}$ and $\mathcal{D}_{\text{C}}$ are the subsets of $\mathcal{D}$ for which $e = 1$ and $e = 0$, respectively. The first term maximizes the likelihood $f_{\text{ALD}}(t \mid \mathbf{x})$ for the observed data, while the second term maximizes the survival probability $S_{\text{ALD}}(t \mid \mathbf{x})$ for the censored data. To achieve this, the parameters $\theta, \sigma, \kappa$ predicted by a multi-layer perceptron (MLP) conditioned on the input features, $\mathbf{x}$, enabling the model to adapt flexibly to varying input distributions. The observed component $\mathcal{L}_{\text{o}}(y; \theta, \sigma, \kappa)$ is defined as:

$$\mathcal{L}_{\text{o}}(y; \theta, \sigma, \kappa) = \log \sigma - \log \frac{\kappa}{\kappa^2 + 1} + \frac{\sqrt{2}}{\sigma} \begin{cases} \kappa(y - \theta), & \text{if } y \geq \theta, \\ \frac{1}{\kappa}(\theta - y), & \text{if } y < \theta. \end{cases} \tag{8}$$

The censored loss component $\mathcal{L}_{\text{c}}(y; \theta, \sigma, \kappa)$ is computed using the survival probability function:

$$\mathcal{L}_{\text{c}}(y; \theta, \sigma, \kappa) = \begin{cases} \log(\kappa^2 + 1) + \frac{\sqrt{2}}{\sigma}\kappa(y - \theta), & \text{if } y \geq \theta, \\ \log(\kappa^2 + 1) - \log\left[1 + \kappa^2\left(1 - \exp\left(-\frac{\sqrt{2}}{\sigma\kappa}(\theta - y)\right)\right)\right], & \text{if } y < \theta. \end{cases} \tag{9}$$

**Proposition 2. (Mean, Mode, Variance of $Y$)** The mean, mode, variance of $Y$ are given by:

$$E[Y] = \theta + \frac{\sigma}{\sqrt{2}}\left(\frac{1}{\kappa} - \kappa\right), \tag{10}$$

$$\text{Mode}[Y] = \theta, \tag{11}$$

$$\text{Var}[Y] = \frac{\sigma^2}{2}\left(\frac{1}{\kappa^2} + \kappa^2\right). \tag{12}$$

**Proposition 3. (Quantiles of $Y$)** Let $\theta_q^{\text{ALD}}$ denotes the $q$-th quantile of $Y$. Then, the quantiles can be expressed as:

$$\theta_q^{\text{ALD}} = \begin{cases} \theta + \frac{\sigma\kappa}{\sqrt{2}} \log\left[\frac{1+\kappa^2}{\kappa^2}q\right], & \text{if } q \in \left(0, \frac{\kappa^2}{1+\kappa^2}\right], \\ \theta - \frac{\sigma}{\sqrt{2}\kappa} \log\left[(1 + \kappa^2)(1 - q)\right], & \text{if } q \in \left(\frac{\kappa^2}{1+\kappa^2}, 1\right). \end{cases} \tag{13}$$

## A.2. Analysis of All the Baselines

**CQRNN.** CQRNN (Pearce et al., 2022) combines the likelihood of the Asymmetric Laplace Distribution, $f_{\text{ALD}}(t \mid \mathbf{x})$, with the re-weighting scheme $w$ introduced by Portnoy (Neocleous et al., 2006). For the observed data, CQRNN employs the Maximum Likelihood Estimation (MLE) approach to directly maximize the likelihood of the Asymmetric Laplace Distribution $\mathcal{AL}(\theta, \sigma, q)$. The likelihood is defined over all quantiles of interest. For censored data, CQRNN splits each censored data point into two pseudo data points: one at the censoring location $y = c$ and another at a large pseudo value $y^*$. This approach enables the formulation of a weighted likelihood for censored data, resulting in the following loss function:

$$\mathcal{L}_{\text{CQR}} = \mathcal{L}_{\text{o}}(y; \theta, \sigma, q) + \mathcal{L}_{\text{c}}(y, y^*; \theta, \sigma, q, w), \tag{14}$$

where $\mathcal{L}_{\text{o}}$ represents the negative log-likelihood for observed data, and $\mathcal{L}_{\text{c}}$ accounts for the weighted negative log-likelihood of censored data using the re-weighting scheme. Expanding this, the loss can be expressed as:

$$\mathcal{L}_{\text{CQR}} = -\sum_{n \in \mathcal{D}_o} \log f_{\text{ALD}}(y_n \mid \mathbf{x}_n) - \sum_{n \in \mathcal{D}_c} [\log f_{\text{ALD}}(y_n \mid \mathbf{x}_n) + (1 - w) f_{\text{ALD}}(y^* \mid \mathbf{x}_n)]. \tag{15}$$

where $\mathcal{D}_{\text{o}}$ and $\mathcal{D}_{\text{c}}$ are the subsets of $\mathcal{D}$ for which $e = 1$ and $e = 0$, respectively. Here, CQRNN utilizes the Asymmetric Laplace Distribution $\mathcal{AL}(\theta, \sigma, q)$ to model the data. The Asymmetric Laplace Distribution, denoted as $\mathcal{AL}(\theta, \sigma, \kappa)$, can be reparameterized as $\mathcal{AL}(\theta, \sigma, q)$ to facilitate quantile regression within a Bayesian inference framework (Yu & Moyeed, 2001), where $q \in (0, 1)$ is the percentile parameter that represents the desired quantile. The relationship between $q$ and $\kappa$ is given by:

$$q = \frac{\kappa^2}{\kappa^2 + 1}. \tag{16}$$

Thus, the probability density function for $Y \sim \mathcal{AL}(\theta, \sigma, q)$ is:

$$f_{\text{ALD}}(y; \theta, \sigma, q) = \frac{q(1 - q)}{\sigma} \begin{cases} \exp\left(\frac{q}{\sigma}(\theta - y)\right), & \text{if } y \geq \theta, \\ \exp\left(\frac{(1-q)}{\sigma}(y - \theta)\right), & \text{if } y < \theta. \end{cases} \tag{17}$$

And the cumulative distribution function is:

$$F_{\text{ALD}}(y; \theta, \sigma, q) = \begin{cases} 1 - (1 - q) \exp\left(\frac{q}{\sigma}(\theta - y)\right), & \text{if } y \geq \theta, \\ q \exp\left(\frac{1-q}{\sigma}(y - \theta)\right), & \text{if } y < \theta. \end{cases} \tag{18}$$

Thus, the negative log-likelihood $\mathcal{L}_{\text{QR}}(y; \theta, \sigma, q)$ then can be explicitly derived as:

$$\mathcal{L}_{\text{QR}}(y; \theta, \sigma, q) = \log \sigma - \log[q(1 - q)] + \frac{1}{\sigma} \begin{cases} q(y - \theta), & \text{if } y \geq \theta \\ (1 - q)(\theta - y), & \text{if } y < \theta \end{cases}. \tag{19}$$

In their implementation, the scale parameter $\sigma$ is omitted, and the percentile parameter $q$ is predefined, typically set to values such as $q = \{0.1, 0.2, \ldots, 0.9\}$. A multi-layer perceptron (MLP) in CQRNN, conditioned on the input features $\mathbf{x}$, predicts $\theta_q$ for the predefined quantile values, corresponding to the location parameter $\theta$. The negative log-likelihood $\mathcal{L}_{\text{QR}}(y; \theta, \sigma, q)$ is then further simplified as:

$$\mathcal{L}_{\text{QR}}(y; \theta_q, q) = \begin{cases} q(y - \theta_q), & \text{if } y \geq \theta_q, \\ (1 - q)(\theta_q - y), & \text{if } y < \theta_q. \end{cases} = (y - \theta_q)(q - \mathbb{I}[\theta_q > y]). \tag{20}$$

This formulation is also referred to as the pinball loss or "checkmark" loss (Koenker & Bassett Jr, 1978), which is widely used in quantile regression to directly optimize the $q$-th quantile estimate. For censored data, CQRNN adopts Portnoy's estimator (Neocleous et al., 2006), which minimizes a specific objective function tailored for censored quantile regression. This approach introduces a re-weighting scheme to handle all censored data, with the formula defined as:

$$\mathcal{L}_{\text{c}}(y, y^*; \theta_q, q, w) = w \mathcal{L}_{\text{QR}}(y; \theta_q, q) + (1 - w) \mathcal{L}_{\text{QR}}(y^*; \theta_q, q), \tag{21}$$

where $y^*$ is a pseudo value set to be significantly larger than all observed values of $y$ in the dataset. Specifically, it is defined as $y^* = 1.2 \max_i y_i$ in CQRNN (Pearce et al., 2022). The weight parameter $w$ is apportioned between each pair of pseudo-data points as:

$$w = \frac{q - q_c}{1 - q_c}, \tag{22}$$

where $q_c$ is the quantile at which the data point was censored ($e = 0, y = c$) with respect to the observed value distribution, i.e., $p(o < c \mid \mathbf{x})$. However, the exact value of $q_c$ is not accessible in practice. To address this issue, CQRNN approximates $q_c$ using the proportion $q$ corresponding to the quantile that is closest to the censoring value $c$, based on the distribution of observed events $y$, which are readily available.

**LogNormal MLE.** LogNormal MLE (Hoseini et al., 2017) enhances parameter estimation using neural networks for LogNormal distributions. Specifically, a random variable $Y$ follows a LogNormal distribution if the natural logarithm of $Y$, denoted as $\ln(Y)$, follows a Normal distribution, i.e., $\ln(Y) \sim \mathcal{N}(\mu, \eta^2)$. Here, $\mu$ represents the mean, and $\eta$ is the standard deviation (SD) of the normal distribution. The probability density function of the LogNormal distribution is given by:

$$f_{\text{LogNormal}}(y; \mu, \eta) = \frac{1}{y\eta\sqrt{2\pi}} \exp\left(-\frac{(\ln y - \mu)^2}{2\eta^2}\right), \tag{23}$$

where $y > 0$ and $\eta > 0$. The cumulative distribution function is expressed as:

$$F_{\text{LogNormal}}(y, \mu, \eta) = \Phi\left(\frac{\ln(y) - \mu}{\eta}\right), \tag{24}$$

where $\Phi(z)$ is the standard normal cumulative distribution function:

$$\Phi(z) = \frac{1}{\sqrt{2\pi}} \int_{-\infty}^{z} \exp\left(-\frac{t^2}{2}\right) dt. \tag{25}$$

The maximum likelihood estimation (MLE) loss with censored data is then defined as:

$$\mathcal{L}_{\text{LogNormal}} = -\sum_{n \in \mathcal{D}_o} \log f_{\text{LogNormal}}(y_n \mid \mathbf{x}_n) - \sum_{n \in \mathcal{D}_c} \log\left(1 - F_{\text{LogNormal}}(y_n \mid \mathbf{x}_j)\right). \tag{26}$$

A multi-layer perceptron (MLP) in LogNormal MLE, conditioned on the input features $\mathbf{x}$, is used to predict the mean $\mu$ and the standard deviation $\eta$ of the corresponding normal distribution. The quantiles $\theta_q^{\text{LogNormal}}$ for the LogNormal distribution can be expressed as:

$$\theta_q^{\text{LogNormal}} = \exp(\mu + \eta\Phi^{-1}(q)), \tag{27}$$

where $\Phi^{-1}(q)$ is the inverse CDF (quantile function) of the standard normal distribution.

**DeepSurv.** DeepSurv (Katzman et al., 2018) is a semi-parametric survival model based on the Cox proportional hazards framework, leveraging deep neural networks for feature representation. A multi-layer perceptron (MLP) in DeepSurv, conditioned on the input features $\mathbf{x}$, is used to predict the log hazard function $h(\mathbf{x})$:

$$\lambda(t \mid \mathbf{x}) = \lambda_0(t)e^{h(\mathbf{x})}, \tag{28}$$

where $\lambda_0(t)$ is the baseline hazard function. The hazard function is defined as:

$$\lambda(t \mid \mathbf{x}) = \lim_{\Delta t \to 0} \frac{P(t \leq T < t + \Delta t \mid T \geq t, \mathbf{x})}{\Delta t}. \tag{29}$$

This can be rewritten as:

$$\lambda(t \mid \mathbf{x}) = -\frac{dS(t \mid \mathbf{x})/dt}{S(t \mid \mathbf{x})}, \tag{30}$$

where $S(t \mid \mathbf{x}) = P(T > t \mid \mathbf{x})$ is the survival function. By integrating both sides, we have:

$$\int \lambda(t \mid \mathbf{x}) \, dt = \int -\frac{dS(t \mid \mathbf{x})}{S(t \mid \mathbf{x})}, \tag{31}$$

which simplifies to:

$$\Lambda(t \mid \mathbf{x}) = -\log S(t \mid \mathbf{x}) + C, \tag{32}$$

where $C$ is the constant of integration and $\Lambda(t \mid \mathbf{x})$ is the cumulative hazard function:

$$\Lambda(t \mid \mathbf{x}) = \Lambda_0(t)e^{h(\mathbf{x})}, \tag{33}$$

where $\Lambda_0(t)$ is the baseline cumulative hazard function. For survival analysis, $C$ is typically set to 0 when starting from $t = 0$. Thus, the survival function can be expressed as:

$$S(t \mid \mathbf{x}) = e^{-\Lambda(t|\mathbf{x})} = e^{-\Lambda_0(t)e^{h(\mathbf{x})}} = [S_0(t)]^{e^{h(\mathbf{x})}}, \tag{34}$$

where $S_0(t)$ is the baseline survival function, typically estimated by the Kaplan-Meier method (Kaplan & Meier, 1958) using the training data. The cumulative distribution function (CDF) can then be derived as:

$$F_{\text{DeepSurv}}(t \mid \mathbf{x}) = 1 - S(t \mid \mathbf{x}) = 1 - [S_0(t)]^{e^{h(\mathbf{x})}}. \tag{35}$$

The quantiles $\theta_q^{\text{DeepSurv}}$ for DeepSurv can be obtained from the inverse CDF $F_{\text{DeepSurv}}^{-1}(t \mid \mathbf{x})$ (quantile function).

**DeepHit.** A multi-layer perceptron (MLP) in DeepHit (Lee et al., 2018), conditioned on the input features $\mathbf{x}$, is used to predict the probability distribution $f(t \mid \mathbf{x})$ over event times using a fully nonparametric approach. The quantiles $\theta_q^{\text{DeepHit}}$ can be obtained from the inverse cumulative distribution function $F_{\text{DeepHit}}^{-1}(t \mid \mathbf{x})$, where $F_{\text{DeepHit}}(t \mid \mathbf{x}) = \sum f_{\text{DeepHit}}(t \mid \mathbf{x})$.

**Gradient Boosting Machine (GBM).** GBM for survival analysis (Dembek et al., 2014) is a nonparametric, additive ensemble of decision trees trained to minimize a loss function tailored to censored data, such as the Cox partial likelihood or the Brier score. Similar to DeepHit, the model directly learns a discrete probability distribution $f(t \mid \mathbf{x})$ over event times. The quantiles $\theta_q^{\text{GBM}}$ can be obtained via the inversion of cumulative distribution function $F_{\text{GBM}}^{-1}(t \mid \mathbf{x})$, where $F_{\text{GBM}}(t \mid \mathbf{x}) = \sum f_{\text{GBM}}(t \mid \mathbf{x})$.

**Random Survival Forests (RSF).** RSF (Ishwaran et al., 2008) is a nonparametric ensemble method that extends Breiman's random forests to survival analysis. Each decision tree is trained on a bootstrap sample of the data and uses the log-rank test statistic to determine optimal splits. The survival function $S_{\text{RSF}}(t \mid \mathbf{x})$ for a given input $\mathbf{x}$ is estimated by aggregating the survival estimates from all trees in the forest:

$$S_{\text{RSF}}(t \mid \mathbf{x}) = \frac{1}{B} \sum_{b=1}^{B} S_b(t \mid \mathbf{x}), \tag{36}$$

where $S_b(t \mid \mathbf{x})$ is the survival function from the $b$-th tree and $B$ is the total number of trees. The corresponding cumulative distribution function is $F_{\text{RSF}}(t \mid \mathbf{x}) = 1 - \hat{S}_{\text{RSF}}(t \mid \mathbf{x})$. The quantiles $\theta_q^{\text{RSF}}$ then can be obtained via the inversion of cumulative distribution function $F_{\text{RSF}}^{-1}(t \mid \mathbf{x})$, where $F_{\text{RSF}}(t \mid \mathbf{x}) = \sum f_{\text{RSF}}(t \mid \mathbf{x})$.

**Deep Survival Machines (DSM).** DSM (Nagpal et al., 2021) is a neural parametric model that represents the survival distribution as a mixture of $K$ components from a chosen parametric family (*e.g.*, LogNormal, Weibull). Given input $\mathbf{x}$, the model outputs both the parameters $\{\mu_k, \eta_k\}_{k=1}^K$ for each component and a mixture weight $\pi_k(\mathbf{x})$ using a neural network. The probability density function is expressed as:

$$f_{\text{DSM}}(t \mid \mathbf{x}) = \sum_{k=1}^K \pi_k(\mathbf{x}) f_k(t; \mu_k, \eta_k), \tag{37}$$

and the cumulative distribution function is:

$$F_{\text{DSM}}(t \mid \mathbf{x}) = \sum_{k=1}^K \pi_k(\mathbf{x}) F_k(t; \mu_k, \eta_k), \tag{38}$$

where $f_k$ and $F_k$ are the PDF and CDF of the $k$-th parametric component (*e.g.*, LogNormal or Weibull). The quantiles $\theta_q^{\text{DSM}}$ are obtained from the inverse CDF $\theta_q^{\text{DSM}} = F_{\text{DSM}}^{-1}(q \mid \mathbf{x})$.

**Summary of Comparative Advantages.** In summary, our proposed method offers a unified and efficient solution that bridges the strengths of both classical and modern survival modeling approaches. Compared to traditional parametric models, it retains closed-form, differentiable PDF and CDF formulations that facilitate stable neural optimization and achieve robust empirical performance across diverse data settings. Compared to nonparametric models, it avoids the drawbacks of temporal discretization and excessive memory usage, providing compact, continuous, and interpretable estimates of survival quantities. Finally, relative to mixture-based models such as DSM, our method is simpler, faster, and more robust—requiring fewer parameters, introducing less architectural complexity, and demonstrating greater stability across datasets and censoring regimes. These advantages collectively establish the ALD framework as a compelling and scalable alternative for survival modeling in both synthetic and real-world applications.

# B. Experimental Details

This section provides additional details about the experiments conducted. The experiments were implemented using the PyTorch framework. Detailed information about the datasets, metrics, baselines and implementation details can be found in Appendix B.1, Appendix B.2, and Appendix B.3, respectively.

**Hardware.** All experiments were conducted on a MacBook Pro with an Apple M3 Pro chip, featuring 12 cores (6 performance and 6 efficiency cores) and 18 GB of memory. CPU-based computations were utilized for all experiments, as the models primarily relied on fully-connected neural networks.

## B.1. Datasets

Our datasets are designed following the settings outlined in Pearce et al. (2022). The first type of dataset consists of synthetic target data with synthetic censoring. In these datasets, the input features, $\mathbf{x}$, are generated uniformly as $\mathbf{x} \sim \mathcal{U}(0, 2)^D$, where $D$ denotes the number of features. The observed variable, $o \sim p(o \mid \mathbf{x})$, and the censored variable, $c \sim p(c \mid \mathbf{x})$, follow distinct distributions, with their parameters varying based on the specific dataset configuration. Table 3 provides detailed descriptions of the distributions for the observed and censored variables. Additionally, the coefficient vector used in some datasets is defined as $\beta = [0.8, 0.6, 0.4, 0.5, -0.3, 0.2, 0.0, -0.7]$.

The other type of dataset comprises real-world target data with real censoring, sourced from various domains and characterized by distinct features, sample sizes, and censoring proportions:

- **METABRIC (Molecular Taxonomy of Breast Cancer International Consortium):** This dataset contains genomic and clinical data for breast cancer patients. It includes 9 features, 1523 training samples, and 381 testing samples, with a censoring proportion of 0.42. Retrieved from the DeepSurv Repository.

- **WHAS (Worcester Heart Attack Study):** This dataset focuses on predicting survival following acute myocardial infarction. It includes 6 features, 1310 training samples, and 328 testing samples, with a censoring proportion of 0.57. Retrieved from the DeepSurv Repository.

*Table 3.* Characteristics of synthetic datasets encompassing the number of features, parameterized distributions of observed variables, and censored variables, as utilized in the experimental framework.

| Synthetic Dataset | Feats ($D$) | Observed Variables $o \sim p(o \mid \mathbf{x})$ | Censored Variables $c \sim p(c \mid \mathbf{x})$ |
|---|---|---|---|
| Norm linear | 1 | $\mathcal{N}(2\mathbf{x} + 10, (\mathbf{x} + 1)^2)$ | $\mathcal{N}(4\mathbf{x} + 10, (0.8\mathbf{x} + 0.4)^2)$ |
| Norm non-linear | 1 | $\mathcal{N}(\mathbf{x}\sin(2\mathbf{x}) + 10, (0.5\mathbf{x} + 0.5)^2)$ | $\mathcal{N}(2\mathbf{x} + 10, 2^2)$ |
| Exponential | 1 | $\mathrm{Exp}(2\mathbf{x} + 4)$ | $\mathrm{Exp}(-3\mathbf{x} + 15)$ |
| Weibull | 1 | $\mathrm{Weibull}(\mathbf{x}\sin(2\mathbf{x} - 2) + 10, 5)$ | $\mathrm{Weibull}(-3\mathbf{x} + 20, 5)$ |
| LogNorm | 1 | $\mathrm{LogNorm}((\mathbf{x} - 1)^2, \mathbf{x}^2)$ | $\mathcal{U}(0, 10)$ |
| Norm uniform | 1 | $\mathcal{N}(2\mathbf{x}\cos(2\mathbf{x}) + 13, (\mathbf{x} + 0.5)^2)$ | $\mathcal{U}(0, 18)$ |
| Norm heavy | 4 | $\mathcal{N}(3\mathbf{x}_0 + \mathbf{x}_1^2 - \mathbf{x}_2^2 + 2\sin(\mathbf{x}_2\mathbf{x}_3) + 6, (\mathbf{x} + 0.5)^2)$ | $\mathcal{U}(0, 12)$ |
| Norm med | 4 | —"— | $\mathcal{U}(0, 20)$ |
| Norm light | 4 | —"— | $\mathcal{U}(0, 40)$ |
| Norm same | 4 | —"— | Equal to observed dist. |
| LogNorm heavy | 8 | $\mathrm{LogNorm}(\sum_{i=1}^{8} \beta_i\mathbf{x}_i, 1)/10$ | $\mathcal{U}(0, 0.4)$ |
| LogNorm med | 8 | —"— | $\mathcal{U}(0, 1.0)$ |
| LogNorm light | 8 | —"— | $\mathcal{U}(0, 3.5)$ |
| LogNorm same | 8 | —"— | Equal to observed dist. |

- **SUPPORT (Study to Understand Prognoses Preferences Outcomes and Risks of Treatment):** This dataset provides survival data for critically ill hospitalized patients. It includes 14 features, 7098 training samples, and 1775 testing samples, with a censoring proportion of 0.32. Covariates include demographic information and basic diagnostic data. Retrieved from the DeepSurv Repository.

- **GBSG (Rotterdam & German Breast Cancer Study Group):** Originating from the German Breast Cancer Study Group, this dataset tracks survival outcomes of breast cancer patients. It includes 7 features, 1785 training samples, and 447 testing samples, with a censoring proportion of 0.42. Retrieved from the DeepSurv Repository.

- **TMBImmuno (Tumor Mutational Burden and Immunotherapy):** This dataset predicts survival time for patients with various cancer types using clinical data. It includes 3 features, 1328 training samples, and 332 testing samples, with a censoring proportion of 0.49. Covariates include age, sex, and mutation count. Retrieved from the cBioPortal.

- **BreastMSK:** Derived from the Memorial Sloan Kettering Cancer Center, this dataset focuses on predicting survival time for breast cancer patients using tumor-related information. It includes 5 features, 1467 training samples, and 367 testing samples, with a censoring proportion of 0.77. Retrieved from the cBioPortal.

- **LGGGBM:** This dataset integrates survival data from low-grade glioma (LGG) and glioblastoma multiforme (GBM), frequently used for model validation in cancer genomics. It includes 5 features, 510 training samples, and 128 testing samples, with a censoring proportion of 0.60. Retrieved from the cBioPortal.

## B.2. Metrics

We employ nine distinct evaluation metrics to assess model performance comprehensively: Mean Absolute Error (MAE), Integrated Brier Score (IBS) (Graf et al., 1999), Harrell's C-Index (Harrell et al., 1982), Uno's C-Index (Uno et al., 2011), censored D-calibration (CensDcal) (Haider et al., 2020), along with the slope and intercept derived from two versions of censored D-calibration (Cal $[S(t|\mathbf{x})]$ (Slope), Cal$[S(t|\mathbf{x})]$(Intercept), Cal$[f(t|\mathbf{x})]$(Slope), and Cal$[f(t|\mathbf{x})]$(Intercept)). These metrics provide a holistic evaluation framework, effectively capturing the survival models' predictive accuracy, discriminative ability, and calibration quality.

- **MAE:**

$$\mathrm{MAE} = \frac{1}{N} \sum_{i=1}^{N} |y_i - \tilde{y}_i|, \tag{39}$$

where $y_i$ represents the observed survival times, $\tilde{y}_i$ denotes the predicted survival times, and $N$ is the total number of data points in the test set.

- **IBS:**

$$\text{BS}(t) = \frac{1}{N} \sum_{i=1}^{N} \left[ \frac{\left(1 - \tilde{F}(t \mid \mathbf{x}_i)\right)^2 \mathbb{I}(y_i \leq t, e_i = 1)}{\tilde{G}(y_i)} + \frac{\tilde{F}(t \mid \mathbf{x}_i)^2 \mathbb{I}(y_i > t)}{\tilde{G}(t)} \right], \tag{40}$$

$$\text{IBS} = \frac{1}{t_2 - t_1} \int_{t_1}^{t_2} \text{BS}(y) \, dy, \tag{41}$$

where $\text{BS}(t)$ represents the Brier score at time $t$, and 100 time points are evenly selected from the 0.1 to 0.9 quantiles of the $y$-distribution in the training set. $\tilde{F}(t \mid \mathbf{x}_i)$ denotes the estimated cumulative distribution function of the survival time for test subjects, $\mathbb{I}(\cdot)$ is the indicator function, and $e_i$ is the event indicator ($e_i = 1$ if the event is observed). $\mathbf{x}_i$ represents the covariates, and $\tilde{G}(\cdot)$ refers to the Kaplan-Meier estimate (Kaplan & Meier, 1958) of the censoring survival function.

- **Harrell's C-Index:**

$$\text{C}_\text{H} = P(\phi_i > \phi_j \mid y_i < y_j, e_i = 1) = \frac{\sum_{i \neq j} \left[\mathbb{I}(\phi_i > \phi_j) + 0.5 * \mathbb{I}(\phi_i = \phi_j)\right]\mathbb{I}(y_i < y_j)e_i}{\sum_{i \neq j} \mathbb{I}(y_i < y_j)e_i}, \tag{42}$$

where $\phi_i = \tilde{S}(y_i \mid \mathbf{x}_i) = 1 - \tilde{F}(t \mid \mathbf{x}_i)$ represents the risk score predicted by the survival model. For implementation, we utilize the `concordance_index_censored` function from the `sksurv.metrics` module, as documented in the scikit-survival API.

- **Uno's C-Index:**

$$\begin{aligned} \text{C}_\text{U} &= P(\phi_i > \phi_j \mid y_i < y_j, y_i < y_\tau) \\ &= \frac{\sum_{i=1}^{n} \sum_{j=1}^{n} \tilde{G}(y_i)^{-2}[\mathbb{I}(\phi_i > \phi_j) + 0.5 * \mathbb{I}(\phi_i = \phi_j)]\mathbb{I}(y_i < y_j, y_i < y_\tau)e_i}{\sum_{i=1}^{n} \sum_{j=1}^{n} \tilde{G}(y_i)^{-2}\mathbb{I}(y_i < y_j, y_i < y_\tau)e_i}, \end{aligned} \tag{43}$$

where $y_\tau$ is the cutoff value for the survival time. For implementation, we utilize the `concordance_index_ipcw` function from the `sksurv.metrics` module, as documented in the scikit-survival API.

- **CensDcal:**

$$\text{CensDcal} = 100 \times \sum_{j=1}^{10} \left( (q_{j+1} - q_j) - \frac{1}{N}\zeta \right)^2, \tag{44}$$

where $\zeta$ is defined by (Goldstein et al., 2020) as:

$$\zeta = \sum_{i \in \mathcal{S}_\text{observed}} \mathbb{I}[\tilde{\theta}_{i,q_j} < y_i \leq \tilde{\theta}_{i,q_{j+1}}] + \sum_{i \in \mathcal{S}_\text{censored}} \frac{(q_{j+1} - q_i)\mathbb{I}[\tilde{\theta}_{i,q_j} < y_i \leq \tilde{\theta}_{i,q_{j+1}}]}{1 - q_i} + \frac{(q_{j+1} - q_j)\mathbb{I}[q_i < q_j]}{1 - q_i}. \tag{45}$$

Here, the percentile parameter $q_j$ is predefined as $[0.1, 0.2, \ldots, 0.9]$ at the outset, and $q_i$ is the quantile at which the data point was censored ($e = 0, y = c$) with respect to the observed value distribution, *i.e.*, $p(o < c \mid \mathbf{x})$. $\tilde{\theta}_{i,q_j}$ represents the estimated $q$th quantile of $y_i$.

- **Slope & Intercept:** The Slope and Intercept metrics evaluate the calibration quality of predicted survival quantiles relative to observed data under censoring. We utilize the `np.polyfit` function from the `NumPy` module, as documented in the NumPy API, to fit the 10 points $\left\{ \left( 0.1j, \sum_j \frac{1}{N}\zeta_j \right) \right\}_{j=1}^{10}$ and subsequently obtain the Slope and Intercept metrics. Two versions of the Slope and Intercept (Cal$[S(t|\mathbf{x})]$(Slope), Cal$[S(t|\mathbf{x})]$(Intercept), Cal$[f(t|\mathbf{x})]$(Slope), and Cal$[f(t|\mathbf{x})]$(Intercept)) are calculated, differing in how the quantile intervals are defined:

  - **Version 1 (Measuring $S(t \mid \mathbf{x})$):** The predicted survival probabilities are divided into intervals based on the target proportions, *i.e.*, $q = [0.1, 0.2, \ldots, 0.9, 1.0]$. For each quantile interval, the proportion of ground truth values (observed survival times) that fall within the corresponding predicted quantile $\frac{1}{N}\zeta$ is calculated. For example, the ratio for 0.1 ($j = 1$) is calculated within the interval $[0, 0.1]$, and for 0.2 ($j = 2$), within $[0, 0.2]$. Thus, the horizontal axis represents the target proportions $0.1j$, while the vertical axis represents the observed proportions $\sum_j \frac{1}{N}\zeta_j$ derived from predictions. In the end, this metric is suitable for evaluating the Survival Function $S(t \mid \mathbf{x})$ (or CDF $F(t \mid \mathbf{x})$).

    – **Version 2 (Measuring $f(t \mid \mathbf{x})$):** Narrower intervals centered around target proportions are used, *i.e.*, $q = [\ldots, 0.4, 0.45, 0.55, 0.6, \ldots]$. For each quantile, the observed proportions are calculated within these narrower intervals. For example, the ratio for 0.1 is calculated within the interval $[0.45, 0.55]$, and for 0.2, within $[0.4, 0.6]$. In the end, this metric is ideal for assessing the probability density function (PDF) $f(t \mid \mathbf{x})$.

### B.3. Implementation Details

**Baselines.** We compare our method against four baselines to evaluate performance and effectiveness: **LogNorm** (Royston, 2001), **DeepSurv** (Katzman et al., 2018), **DeepHit** (Lee et al., 2018), and **CQRNN** (Pearce et al., 2022). All methods were trained using the same optimization procedure and neural network architecture to ensure a fair comparison. The implementations for **CQRNN** and **LogNorm** were sourced from the official CQRNN repository (GitHub Link). The implementations for **DeepSurv** and **DeepHit** were based on the `pycox.methods` module (GitHub Link).

**Hyperparameter settings.** All experiments were repeated across 10 random seeds to ensure robust and reliable results. The hyperparameter settings were as follows:

- **Default Neural Network Architecture:** Fully-connected network with two hidden layers, each consisting of 100 hidden nodes, using ReLU activations.

- **Default Epochs:** 200

- **Default Batch Size:** 128

- **Default Learning Rate:** 0.01

- **Dropout Rate:** 0.1

- **Optimizer:** Adam

- **Batch Norm:** FALSE

**Our Method.** We incorporate a residual connection between the shared feature extraction layer and the first hidden layer to enhance gradient flow. To satisfy the parameter constraints of the Asymmetric Laplace Distribution (ALD), the final output layer applies an exponential (Exp) activation function, ensuring that the outputs of the $\theta$, $\sigma$ and $\kappa$ branches remain positive. Each of the two hidden layers contains 32 hidden nodes. A validation set is created by splitting 20% of the training set. Early stopping is utilized to terminate training when the validation performance ceases to improve.

**CQRNN.** We followed the hyperparameter settings tuned in the original paper (Pearce et al., 2022), where three random splits were used for validation (ensuring no overlap with the random seeds used in the final test runs). The following settings were applied:

- **Weight Decay:** 0.0001

- **Grid Size:** 10

- **Pseudo Value:** $y^* = 1.2 \times \max_i y_i$

- **Dropout Rate:** 0.333

The number of epochs and dropout usage were adjusted based on the dataset type:

- **Synthetic Datasets:**
  - **Norm linear, Norm non-linear, Exponential, Weibull, LogNorm, Norm uniform:** 100 epochs with dropout disabled.
  - **Norm heavy, Norm medium, Norm light, Norm same:** 20 epochs with dropout disabled.
  - **LogNorm heavy, LogNorm medium, LogNorm light, LogNorm same:** 10 epochs with dropout disabled.

- **Real-World Datasets:**

- **METABRIC:** 20 epochs with dropout disabled.
- **WHAS:** 100 epochs with dropout disabled.
- **SUPPORT:** 10 epochs with dropout disabled.
- **GBSG:** 20 epochs with dropout enabled.
- **TMBImmuno:** 50 epochs with dropout disabled.
- **BreastMSK:** 100 epochs with dropout disabled.
- **LGGGBM:** 50 epochs with dropout enabled.

**LogNorm.** The output dimensions of the default neural network architecture are 2, where the two outputs represent the mean and standard deviation of a Log-Normal distribution. To ensure the standard deviation prediction is always positive and differentiable, the output representing the standard deviation is passed through a `SoftPlus` activation function. We followed the hyperparameter settings tuned in the original paper (Pearce et al., 2022), with a **Dropout Rate** of 0.333. The number of epochs and dropout usage were adjusted based on the dataset type as follows:

- **Synthetic Datasets:** The same settings as described above for **CQRNN**.

- **Real-World Datasets:**

    - **METABRIC:** 10 epochs with dropout disabled.
    - **WHAS:** 50 epochs with dropout disabled.
    - **SUPPORT:** 20 epochs with dropout disabled.
    - **GBSG:** 10 epochs with dropout enabled.
    - **TMBImmuno:** 50 epochs with dropout disabled.
    - **BreastMSK:** 50 epochs with dropout disabled.
    - **LGGGBM:** 20 epochs with dropout enabled.

**DeepSurv.** We adhered to the official hyperparameter settings from the `pycox.methods` module (GitHub Link). Each of the two hidden layers contains 32 hidden nodes. A validation set was created by splitting 20% of the training set. Early stopping was employed to terminate training when the validation performance ceased to improve. Batch normalization was applied.

**DeepHit.** We adhered to the official hyperparameter settings from the `pycox.methods` module (GitHub Link). Each of the two hidden layers contains 32 hidden nodes. A validation set was created by splitting 20% of the training set. Early stopping was employed to terminate training when the validation performance ceased to improve. Batch normalization was applied, with additional settings: `num_durations` = 100, `alpha` = 0.2, and `sigma` = 0.1.

**GBM.** We implemented GBM model using the `GradientBoostingSurvivalAnalysis` class from the `sksurv.ensemble` module.[3] The model was configured with `n_estimators` = 100, `learning_rate` = 0.01, and `max_depth` = 3, following standard practices for tree-based boosting in survival analysis.

**RSF.** The RSF model was implemented using the `RandomSurvivalForest` class from `sksurv.ensemble`.[4] We used the default configuration with `n_estimators` = 100, which has been shown to provide a good trade-off between performance and efficiency.

**DSM.** We adopted the Deep Survival Machines (DSM) model from the `auton-survival` library[5], using the `DeepSurvivalMachines` class. The neural network consisted of two hidden layers, each with 32 hidden units. For the LogNorm variant, we used $k = 10$ mixture components, as higher values of $k$ were observed to degrade performance due to overfitting and instability. For the Weibull variant, we followed the library's default configuration with $k = 100$, which was necessary to maintain sufficient model capacity. Training was performed using observed event times and censoring indicators, and the final predictions were computed by evaluating the learned mixture distribution over a fixed 1000-point discretized time grid to estimate the cumulative distribution function (CDF).

---

[3] https://scikit-survival.readthedocs.io/en/stable/api/ensemble.html
[4] https://scikit-survival.readthedocs.io/en/stable/api/ensemble.html
[5] https://autonlab.org/auton-survival/models/dsm/index.html

# C. Additional Results

This section presents additional empirical results to support the evaluation of our method. The results include comprehensive comparisons across datasets and metrics, as well as calibration plots and case studies to illustrate key behaviors. Full benchmarking results are provided in Appendix C.1, calibration curve visualizations in Appendix C.2, and detailed case analyses in Appendix C.3.

## C.1. Overall Results.

Table 5 summarizes the full results across 21 datasets, comparing our method with 8 baselines across 9 metrics. Figure 4 visualizes these results for a more intuitive comparison. In Table 5, the best performance is highlighted in bold. Figure 4 provides a graphical representation of nine distinct evaluation metrics to comprehensively assess predictive performance, including Mean Absolute Error (MAE), Integrated Brier Score (IBS), Harrell's C-Index, Uno's C-Index, Censored D-calibration (CensDcal), and the slope and intercept derived from two versions of censored D-calibration (Cal[$S(t|\mathbf{x})$](Slope), Cal[$S(t|\mathbf{x})$](Intercept), Cal[$f(t|\mathbf{x})$](Slope), and Cal[$f(t|\mathbf{x})$](Intercept)). Specifically, the following transformations were applied to enhance the clarity of the results:

- MAE and CensDcal were log-transformed to better illustrate their value distributions and differences.

- For Cal[$S(t|\mathbf{x})$](Slope) and Cal[$f(t|\mathbf{x})$](Slope), $|1 - \text{Cal}[S(t|\mathbf{x})](\text{Slope})|$ and $|1 - \text{Cal}[f(t|\mathbf{x})](\text{Slope})|$ were computed to measure their deviation from the ideal value of 1.

- For Cal[$S(t|\mathbf{x})$](Intercept) and Cal[$f(t|\mathbf{x})$](Intercept), $|\text{Cal}[S(t|\mathbf{x})](\text{Intercept})|$ and $|\text{Cal}[f(t|\mathbf{x})](\text{Intercept})|$ were computed to measure their deviation from the ideal value of 0.

These transformations allow for a more intuitive comparison of the performance differences across metrics and models. In the end, each subfigure in Figure 4 provides a comparison of its corresponding metric. The x-axis lists all the datasets, both synthetic datasets (*e.g.*, Gaussian linear, exponential, *etc.*) and real-world datasets (*e.g.*, METABRIC, WHAS, *etc.*), the y-axis indicates the range of corresponding its metric, and error bars represent standard deviation across 10 model runs.

*Table 4.* Full results for all datasets, methods, and metrics. The values represent the mean ± 1 standard error for the test set over 10 runs.

| Dataset | Method | MAE | IBS | Harrell's C-index | Uno's C-index | CensDcal | Cal[$S(t|\mathbf{x})$](Slope) | Cal[$S(t|\mathbf{x})$](Intercept) | Cal[$f(t|\mathbf{x})$](Slope) | Cal[$f(t|\mathbf{x})$](Intercept) |
|---|---|---|---|---|---|---|---|---|---|---|
| Norm_linear | ALD | 0.865 ± 1.337 | **0.278 ± 0.008** | 0.653 ± 0.014 | 0.648 ± 0.011 | 0.407 ± 0.343 | 1.025 ± 0.016 | 0.005 ± 0.030 | 1.027 ± 0.042 | -0.016 ± 0.037 |
| | CQRNN | 0.278 ± 0.144 | 0.326 ± 0.034 | **0.657 ± 0.008** | **0.651 ± 0.007** | 0.466 ± 0.150 | **1.001 ± 0.062** | **-0.003 ± 0.026** | **1.007 ± 0.039** | -0.020 ± 0.047 |
| | LogNorm | 0.372 ± 0.228 | 0.709 ± 0.028 | 0.652 ± 0.016 | 0.646 ± 0.014 | 0.496 ± 0.399 | 0.965 ± 0.024 | 0.005 ± 0.014 | 0.978 ± 0.041 | 0.014 ± 0.067 |
| | DeepSurv | **0.239 ± 0.114** | 0.676 ± 0.026 | 0.657 ± 0.008 | 0.651 ± 0.007 | 0.139 ± 0.071 | 0.983 ± 0.018 | 0.015 ± 0.016 | **1.007 ± 0.018** | -0.005 ± 0.014 |
| | DeepHit | 1.481 ± 0.527 | 0.503 ± 0.025 | 0.635 ± 0.024 | 0.628 ± 0.025 | 6.540 ± 1.458 | 0.967 ± 0.036 | 0.098 ± 0.070 | 1.216 ± 0.051 | -0.302 ± 0.029 |
| | GBM | 0.631 ± 0.054 | 0.305 ± 0.007 | 0.641 ± 0.012 | 0.633 ± 0.010 | 0.146 ± 0.078 | 0.997 ± 0.023 | **0.003 ± 0.018** | 1.008 ± 0.014 | -0.014 ± 0.014 |
| | RSF | 1.234 ± 0.120 | 0.328 ± 0.009 | 0.588 ± 0.014 | 0.584 ± 0.013 | 0.994 ± 0.256 | 0.886 ± 0.015 | -0.025 ± 0.017 | 0.880 ± 0.021 | 0.069 ± 0.020 |
| | DSM(LogNorm) | 1.039 ± 0.046 | 0.324 ± 0.007 | 0.652 ± 0.009 | 0.646 ± 0.008 | 2.534 ± 0.523 | 1.028 ± 0.017 | -0.016 ± 0.016 | 0.968 ± 0.019 | 0.071 ± 0.022 |
| | DSM(Weibull) | 1.045 ± 0.050 | 0.323 ± 0.007 | 0.654 ± 0.008 | 0.647 ± 0.007 | 3.659 ± 0.593 | 1.104 ± 0.018 | 0.004 ± 0.017 | 1.068 ± 0.010 | **0.000 ± 0.015** |
| Norm_nonlinear | ALD | 0.243 ± 0.080 | **0.212 ± 0.006** | 0.670 ± 0.015 | 0.644 ± 0.016 | 0.406 ± 0.179 | 1.072 ± 0.021 | -0.011 ± 0.015 | 1.038 ± 0.025 | -0.016 ± 0.040 |
| | CQRNN | **0.117 ± 0.037** | 0.507 ± 0.026 | **0.674 ± 0.014** | **0.651 ± 0.014** | 0.241 ± 0.099 | 0.983 ± 0.026 | **0.002 ± 0.018** | 0.987 ± 0.012 | 0.011 ± 0.027 |
| | LogNorm | 0.396 ± 0.432 | 0.560 ± 0.058 | 0.630 ± 0.087 | 0.617 ± 0.074 | 2.136 ± 3.886 | 1.003 ± 0.052 | 0.051 ± 0.054 | 1.097 ± 0.060 | -0.098 ± 0.059 |
| | DeepSurv | 0.197 ± 0.047 | 0.623 ± 0.013 | 0.670 ± 0.015 | 0.650 ± 0.014 | 0.196 ± 0.128 | 1.015 ± 0.019 | 0.007 ± 0.016 | 1.019 ± 0.022 | -0.007 ± 0.026 |
| | DeepHit | 1.099 ± 0.130 | 0.515 ± 0.049 | 0.610 ± 0.051 | 0.596 ± 0.040 | 3.886 ± 3.682 | **0.999 ± 0.064** | -0.007 ± 0.061 | 1.064 ± 0.067 | -0.161 ± 0.084 |
| | GBM | 0.323 ± 0.027 | 0.226 ± 0.004 | 0.653 ± 0.018 | 0.636 ± 0.016 | **0.179 ± 0.104** | 1.004 ± 0.017 | 0.004 ± 0.018 | **1.012 ± 0.027** | -0.013 ± 0.031 |
| | RSF | 0.489 ± 0.045 | 0.242 ± 0.004 | 0.622 ± 0.013 | 0.604 ± 0.012 | 0.910 ± 0.300 | 0.885 ± 0.023 | -0.017 ± 0.012 | 0.890 ± 0.019 | 0.063 ± 0.025 |
| | DSM(LogNorm) | 0.510 ± 0.038 | 0.234 ± 0.005 | 0.598 ± 0.016 | 0.565 ± 0.016 | 2.683 ± 0.282 | 0.988 ± 0.017 | 0.080 ± 0.022 | 1.072 ± 0.034 | **0.004 ± 0.035** |
| | DSM(Weibull) | 0.477 ± 0.029 | 0.235 ± 0.005 | 0.634 ± 0.018 | 0.610 ± 0.016 | 3.861 ± 0.316 | 0.988 ± 0.015 | 0.135 ± 0.016 | 1.157 ± 0.022 | -0.067 ± 0.026 |
| Norm uniform | ALD | 0.473 ± 0.344 | **0.045 ± 0.002** | 0.785 ± 0.010 | 0.703 ± 0.019 | 0.115 ± 0.030 | 1.019 ± 0.020 | **0.002 ± 0.016** | 1.016 ± 0.015 | **-0.006 ± 0.021** |
| | CQRNN | **0.301 ± 0.104** | 0.535 ± 0.015 | **0.786 ± 0.009** | **0.706 ± 0.015** | 0.162 ± 0.141 | 1.018 ± 0.033 | -0.013 ± 0.013 | 1.002 ± 0.015 | -0.007 ± 0.017 |
| | LogNorm | 17.079 ± 5.833 | 0.387 ± 0.013 | 0.615 ± 0.118 | 0.578 ± 0.083 | 3.799 ± 0.354 | 0.951 ± 0.059 | 0.159 ± 0.043 | 1.186 ± 0.016 | -0.129 ± 0.021 |
| | DeepSurv | 0.627 ± 0.180 | 0.516 ± 0.009 | 0.781 ± 0.014 | 0.701 ± 0.020 | 0.466 ± 0.149 | 1.038 ± 0.017 | 0.022 ± 0.013 | 1.069 ± 0.012 | -0.051 ± 0.016 |
| | DeepHit | 1.468 ± 0.458 | 0.364 ± 0.048 | 0.758 ± 0.033 | 0.688 ± 0.028 | 3.150 ± 1.142 | 1.015 ± 0.045 | 0.024 ± 0.047 | 1.128 ± 0.051 | -0.209 ± 0.027 |
| | GBM | 1.134 ± 0.093 | 0.058 ± 0.003 | 0.739 ± 0.019 | 0.674 ± 0.015 | **0.101 ± 0.054** | **0.991 ± 0.013** | 0.013 ± 0.009 | 1.013 ± 0.012 | -0.008 ± 0.017 |
| | RSF | 1.160 ± 0.115 | 0.055 ± 0.003 | 0.665 ± 0.021 | 0.621 ± 0.019 | 0.412 ± 0.101 | 0.915 ± 0.011 | **-0.002 ± 0.007** | 0.930 ± 0.013 | 0.043 ± 0.012 |
| | DSM(LogNorm) | 1.319 ± 0.023 | 0.063 ± 0.003 | 0.782 ± 0.012 | 0.691 ± 0.018 | 6.693 ± 1.280 | 0.920 ± 0.015 | 0.102 ± 0.019 | 1.069 ± 0.025 | -0.035 ± 0.022 |
| | DSM(Weibull) | 1.352 ± 0.020 | 0.063 ± 0.003 | 0.768 ± 0.017 | 0.683 ± 0.018 | 3.963 ± 0.695 | 0.984 ± 0.020 | 0.038 ± 0.011 | 1.041 ± 0.015 | -0.025 ± 0.019 |
| Exponential | ALD | 2.942 ± 2.389 | 0.309 ± 0.018 | **0.560 ± 0.008** | **0.560 ± 0.007** | 0.432 ± 0.405 | 0.978 ± 0.047 | -0.015 ± 0.014 | 0.964 ± 0.049 | 0.016 ± 0.053 |
| | CQRNN | 1.943 ± 0.297 | 0.317 ± 0.013 | 0.558 ± 0.013 | 0.557 ± 0.011 | 0.305 ± 0.123 | 0.976 ± 0.066 | 0.012 ± 0.043 | **1.001 ± 0.027** | -0.008 ± 0.019 |
| | LogNorm | 3.223 ± 0.823 | 0.455 ± 0.010 | 0.527 ± 0.028 | 0.528 ± 0.028 | 0.419 ± 0.141 | 0.983 ± 0.026 | 0.042 ± 0.018 | 1.057 ± 0.022 | -0.051 ± 0.021 |
| | DeepSurv | **1.913 ± 0.269** | 0.486 ± 0.015 | 0.558 ± 0.007 | 0.558 ± 0.006 | **0.119 ± 0.066** | 0.986 ± 0.033 | 0.009 ± 0.022 | 1.003 ± 0.018 | -0.008 ± 0.018 |
| | DeepHit | 2.626 ± 2.759 | 0.471 ± 0.012 | 0.526 ± 0.032 | 0.526 ± 0.031 | 1.205 ± 1.060 | 0.960 ± 0.027 | -0.012 ± 0.021 | 0.907 ± 0.055 | 0.127 ± 0.066 |
| | GBM | 2.089 ± 0.249 | 0.295 ± 0.007 | 0.541 ± 0.011 | 0.540 ± 0.010 | 0.137 ± 0.055 | 0.977 ± 0.034 | **0.007 ± 0.025** | 0.995 ± 0.019 | -0.004 ± 0.017 |
| | RSF | 3.422 ± 0.208 | 0.343 ± 0.016 | 0.515 ± 0.015 | 0.513 ± 0.014 | 0.778 ± 0.335 | 0.885 ± 0.026 | -0.009 ± 0.011 | 0.897 ± 0.024 | 0.066 ± 0.020 |
| | DSM(LogNorm) | 2.374 ± 0.229 | 0.294 ± 0.007 | 0.547 ± 0.013 | 0.546 ± 0.012 | 1.935 ± 0.344 | 0.976 ± 0.025 | 0.040 ± 0.016 | 1.040 ± 0.025 | -0.028 ± 0.024 |
| | DSM(Weibull) | 1.921 ± 0.230 | **0.292 ± 0.007** | 0.557 ± 0.007 | 0.557 ± 0.006 | 1.571 ± 0.345 | **0.990 ± 0.022** | 0.012 ± 0.010 | 1.004 ± 0.022 | **0.002 ± 0.022** |

| Dataset | Method | MAE | IBS | Harrell's C-index | Uno's C-index | CensDcal | Cal[S(t\|x)](Slope) | Cal[S(t\|x)](Intercept) | Cal[f(t\|x)](Slope) | Cal[f(t\|x)](Intercept) |
|---|---|---|---|---|---|---|---|---|---|---|
| Weibull | ALD | 5.135 ± 9.533 | **0.219 ± 0.028** | 0.767 ± 0.009 | 0.763 ± 0.009 | 0.648 ± 0.511 | 1.044 ± 0.023 | -0.023 ± 0.033 | 0.993 ± 0.049 | 0.021 ± 0.060 |
| | CQRNN | **0.350 ± 0.098** | 0.461 ± 0.030 | **0.775 ± 0.005** | **0.769 ± 0.005** | 0.346 ± 0.131 | 0.989 ± 0.057 | **-0.001 ± 0.042** | 0.995 ± 0.022 | **-0.003 ± 0.023** |
| | LogNorm | 0.862 ± 0.121 | 0.840 ± 0.021 | 0.773 ± 0.006 | 0.767 ± 0.006 | 0.598 ± 0.172 | 0.993 ± 0.026 | 0.029 ± 0.016 | 1.050 ± 0.028 | -0.053 ± 0.038 |
| | DeepSurv | 0.381 ± 0.098 | 0.969 ± 0.019 | 0.772 ± 0.004 | 0.766 ± 0.006 | **0.118 ± 0.049** | 0.989 ± 0.023 | 0.006 ± 0.012 | **1.005 ± 0.018** | -0.009 ± 0.021 |
| | DeepHit | 1.975 ± 0.172 | 0.618 ± 0.032 | 0.769 ± 0.006 | 0.763 ± 0.007 | 3.020 ± 1.157 | **0.998 ± 0.058** | 0.122 ± 0.071 | 1.206 ± 0.056 | -0.187 ± 0.069 |
| | GBM | 1.467 ± 0.100 | 0.254 ± 0.006 | 0.766 ± 0.005 | 0.759 ± 0.006 | 0.359 ± 0.143 | 1.025 ± 0.022 | 0.012 ± 0.020 | 1.053 ± 0.025 | -0.057 ± 0.021 |
| | RSF | 1.259 ± 0.155 | 0.233 ± 0.011 | 0.747 ± 0.009 | 0.740 ± 0.010 | 0.986 ± 0.299 | 0.863 ± 0.030 | -0.009 ± 0.014 | 0.886 ± 0.024 | 0.066 ± 0.021 |
| | DSM(LogNorm) | 2.608 ± 0.072 | 0.328 ± 0.007 | 0.745 ± 0.005 | 0.743 ± 0.005 | 2.713 ± 0.469 | 1.015 ± 0.016 | -0.004 ± 0.022 | 0.985 ± 0.033 | 0.046 ± 0.026 |
| | DSM(Weibull) | 2.661 ± 0.073 | 0.329 ± 0.007 | 0.745 ± 0.004 | 0.742 ± 0.004 | 3.569 ± 0.486 | 1.079 ± 0.015 | -0.013 ± 0.020 | 1.026 ± 0.027 | 0.018 ± 0.023 |
| LogNorm | ALD | 0.363 ± 0.068 | **0.376 ± 0.013** | 0.588 ± 0.014 | 0.585 ± 0.014 | 0.256 ± 0.150 | 1.005 ± 0.021 | 0.006 ± 0.011 | 1.011 ± 0.028 | -0.004 ± 0.029 |
| | CQRNN | 0.950 ± 0.091 | 0.407 ± 0.019 | 0.588 ± 0.016 | 0.584 ± 0.016 | 0.459 ± 0.220 | 1.024 ± 0.066 | -0.019 ± 0.034 | 0.996 ± 0.042 | **0.000 ± 0.031** |
| | LogNorm | **0.267 ± 0.062** | 0.645 ± 0.021 | 0.588 ± 0.015 | 0.584 ± 0.015 | **0.103 ± 0.020** | 1.009 ± 0.012 | 0.006 ± 0.009 | 1.016 ± 0.015 | -0.010 ± 0.017 |
| | DeepSurv | 0.963 ± 0.058 | 0.658 ± 0.029 | **0.589 ± 0.016** | **0.586 ± 0.016** | 0.137 ± 0.049 | 0.996 ± 0.021 | **0.001 ± 0.020** | 0.997 ± 0.025 | 0.002 ± 0.021 |
| | DeepHit | 0.902 ± 0.504 | 0.568 ± 0.025 | 0.551 ± 0.032 | 0.548 ± 0.031 | 2.088 ± 1.666 | 0.988 ± 0.031 | -0.026 ± 0.050 | 0.892 ± 0.072 | 0.162 ± 0.090 |
| | GBM | 1.034 ± 0.049 | 0.385 ± 0.010 | 0.579 ± 0.015 | 0.575 ± 0.015 | 0.163 ± 0.052 | **0.999 ± 0.028** | -0.002 ± 0.018 | **0.999 ± 0.018** | -0.003 ± 0.016 |
| | RSF | 1.182 ± 0.074 | 0.428 ± 0.012 | 0.542 ± 0.019 | 0.541 ± 0.018 | 0.907 ± 0.247 | 0.881 ± 0.019 | -0.017 ± 0.015 | 0.887 ± 0.019 | 0.062 ± 0.020 |
| | DSM(LogNorm) | 0.939 ± 0.060 | 0.388 ± 0.011 | 0.506 ± 0.009 | 0.505 ± 0.008 | 1.191 ± 0.335 | 1.008 ± 0.022 | -0.006 ± 0.016 | 0.988 ± 0.021 | 0.020 ± 0.020 |
| | DSM(Weibull) | 0.762 ± 0.055 | 0.382 ± 0.011 | 0.501 ± 0.013 | 0.501 ± 0.012 | 2.153 ± 0.480 | 1.059 ± 0.019 | -0.010 ± 0.011 | 1.021 ± 0.017 | 0.011 ± 0.016 |
| Norm heavy | ALD | 0.667 ± 0.139 | **0.019 ± 0.001** | **0.919 ± 0.007** | **0.870 ± 0.029** | 0.036 ± 0.017 | 1.009 ± 0.005 | -0.004 ± 0.004 | 1.001 ± 0.009 | **-0.002 ± 0.010** |
| | CQRNN | **0.574 ± 0.031** | 0.538 ± 0.006 | 0.914 ± 0.008 | 0.863 ± 0.033 | 0.062 ± 0.009 | **1.000 ± 0.019** | -0.002 ± 0.007 | **1.000 ± 0.012** | -0.004 ± 0.011 |
| | LogNorm | 33.140 ± 12.004 | 0.411 ± 0.014 | 0.781 ± 0.071 | 0.679 ± 0.126 | 2.249 ± 0.490 | 1.122 ± 0.022 | **0.001 ± 0.029** | 1.111 ± 0.031 | -0.074 ± 0.032 |
| | DeepSurv | 1.662 ± 0.157 | 0.558 ± 0.007 | 0.726 ± 0.035 | 0.582 ± 0.056 | 0.577 ± 0.067 | 1.070 ± 0.006 | -0.002 ± 0.009 | 1.065 ± 0.011 | -0.065 ± 0.010 |
| | DeepHit | 0.814 ± 0.104 | 0.475 ± 0.037 | 0.913 ± 0.009 | 0.856 ± 0.034 | 1.349 ± 0.374 | 1.051 ± 0.044 | 0.055 ± 0.035 | 1.139 ± 0.027 | -0.121 ± 0.036 |
| | GBM | 1.627 ± 0.034 | 0.041 ± 0.002 | 0.874 ± 0.009 | 0.796 ± 0.084 | 0.043 ± 0.011 | 1.003 ± 0.009 | 0.011 ± 0.008 | 1.019 ± 0.006 | -0.015 ± 0.007 |
| | RSF | 0.607 ± 0.016 | 0.020 ± 0.001 | 0.910 ± 0.008 | 0.802 ± 0.111 | 0.045 ± 0.016 | 1.015 ± 0.006 | 0.002 ± 0.006 | 1.015 ± 0.009 | -0.008 ± 0.010 |
| | DSM(LogNorm) | 1.915 ± 0.037 | 0.047 ± 0.002 | 0.782 ± 0.026 | 0.706 ± 0.067 | 7.049 ± 0.555 | 0.997 ± 0.008 | 0.023 ± 0.007 | 1.027 ± 0.005 | -0.015 ± 0.005 |
| | DSM(Weibull) | 1.906 ± 0.037 | 0.047 ± 0.002 | 0.827 ± 0.014 | 0.771 ± 0.032 | 6.828 ± 0.605 | 1.008 ± 0.008 | 0.006 ± 0.007 | 1.013 ± 0.005 | -0.006 ± 0.005 |
| Norm med. | ALD | **0.238 ± 0.036** | **0.047 ± 0.003** | **0.894 ± 0.005** | **0.872 ± 0.004** | 0.157 ± 0.044 | 1.058 ± 0.012 | -0.035 ± 0.011 | **0.997 ± 0.012** | 0.004 ± 0.014 |
| | CQRNN | 0.312 ± 0.033 | 0.608 ± 0.010 | 0.888 ± 0.005 | 0.867 ± 0.005 | 0.097 ± 0.045 | 0.984 ± 0.026 | **0.001 ± 0.013** | 0.989 ± 0.019 | 0.007 ± 0.020 |
| | LogNorm | 7.300 ± 2.579 | 0.430 ± 0.019 | 0.810 ± 0.048 | 0.777 ± 0.048 | 8.192 ± 0.660 | 0.751 ± 0.073 | 0.350 ± 0.052 | 1.280 ± 0.021 | -0.192 ± 0.036 |
| | DeepSurv | 0.253 ± 0.026 | 0.722 ± 0.012 | 0.893 ± 0.004 | 0.871 ± 0.004 | 0.609 ± 0.111 | 1.054 ± 0.014 | 0.008 ± 0.015 | 1.061 ± 0.019 | -0.051 ± 0.017 |
| | DeepHit | 0.916 ± 0.077 | 0.576 ± 0.012 | 0.886 ± 0.006 | 0.863 ± 0.005 | 1.655 ± 0.409 | 1.056 ± 0.032 | 0.038 ± 0.026 | 1.130 ± 0.032 | -0.130 ± 0.066 |
| | GBM | 1.443 ± 0.035 | 0.096 ± 0.004 | 0.859 ± 0.004 | 0.837 ± 0.003 | 0.121 ± 0.045 | **1.014 ± 0.011** | 0.015 ± 0.009 | 1.037 ± 0.011 | -0.033 ± 0.011 |
| | RSF | 0.406 ± 0.022 | 0.051 ± 0.003 | 0.884 ± 0.004 | 0.861 ± 0.004 | **0.072 ± 0.035** | 1.016 ± 0.016 | 0.003 ± 0.010 | 1.017 ± 0.012 | -0.012 ± 0.014 |
| | DSM(LogNorm) | 1.918 ± 0.035 | 0.118 ± 0.006 | 0.692 ± 0.028 | 0.662 ± 0.027 | 1.404 ± 0.176 | 1.019 ± 0.014 | 0.030 ± 0.009 | 1.058 ± 0.009 | -0.036 ± 0.012 |
| | DSM(Weibull) | 1.920 ± 0.038 | 0.117 ± 0.005 | 0.757 ± 0.018 | 0.728 ± 0.017 | 1.174 ± 0.191 | 1.031 ± 0.018 | 0.009 ± 0.013 | 1.032 ± 0.010 | -0.010 ± 0.012 |
| Norm light | ALD | 0.236 ± 0.051 | **0.090 ± 0.007** | **0.882 ± 0.004** | **0.874 ± 0.004** | 0.339 ± 0.076 | 1.087 ± 0.017 | -0.050 ± 0.011 | **0.999 ± 0.014** | 0.005 ± 0.021 |
| | CQRNN | 0.271 ± 0.032 | 0.671 ± 0.013 | 0.879 ± 0.002 | 0.871 ± 0.002 | 0.149 ± 0.097 | **0.999 ± 0.027** | -0.014 ± 0.017 | 0.985 ± 0.022 | **0.004 ± 0.027** |
| | LogNorm | 3.152 ± 2.154 | 0.548 ± 0.023 | 0.832 ± 0.022 | 0.821 ± 0.022 | 12.884 ± 1.700 | 0.804 ± 0.162 | 0.358 ± 0.126 | 1.351 ± 0.018 | -0.256 ± 0.038 |
| | DeepSurv | 0.247 ± 0.016 | 0.941 ± 0.024 | **0.882 ± 0.002** | **0.874 ± 0.002** | 0.582 ± 0.127 | 1.038 ± 0.014 | 0.016 ± 0.019 | 1.057 ± 0.023 | -0.040 ± 0.023 |
| | DeepHit | 0.959 ± 0.051 | 0.691 ± 0.030 | 0.875 ± 0.004 | 0.867 ± 0.004 | 1.854 ± 0.461 | 1.044 ± 0.041 | 0.063 ± 0.022 | 1.159 ± 0.023 | -0.157 ± 0.053 |
| | GBM | 1.329 ± 0.029 | 0.174 ± 0.011 | 0.849 ± 0.004 | 0.841 ± 0.004 | 0.250 ± 0.079 | 1.015 ± 0.013 | 0.023 ± 0.013 | 1.056 ± 0.013 | -0.055 ± 0.013 |
| | RSF | 0.365 ± 0.013 | 0.096 ± 0.006 | 0.875 ± 0.003 | 0.866 ± 0.003 | **0.113 ± 0.053** | 1.014 ± 0.014 | -0.001 ± 0.013 | 1.014 ± 0.016 | -0.013 ± 0.015 |
| | DSM(LogNorm) | 1.931 ± 0.035 | 0.227 ± 0.014 | 0.653 ± 0.020 | 0.643 ± 0.020 | 0.888 ± 0.119 | 1.040 ± 0.016 | 0.024 ± 0.008 | 1.069 ± 0.012 | -0.048 ± 0.014 |
| | DSM(Weibull) | 1.926 ± 0.039 | 0.222 ± 0.014 | 0.727 ± 0.016 | 0.717 ± 0.016 | 0.531 ± 0.130 | 1.044 ± 0.016 | 0.013 ± 0.012 | 1.044 ± 0.012 | -0.014 ± 0.014 |
| Norm same | ALD | 0.405 ± 0.079 | **0.066 ± 0.003** | 0.890 ± 0.005 | 0.847 ± 0.008 | 0.114 ± 0.036 | 1.007 ± 0.014 | 0.006 ± 0.012 | 1.004 ± 0.018 | 0.010 ± 0.022 |
| | CQRNN | 0.301 ± 0.024 | 0.568 ± 0.018 | 0.886 ± 0.006 | 0.841 ± 0.016 | 0.147 ± 0.109 | 0.988 ± 0.028 | 0.003 ± 0.014 | **0.999 ± 0.024** | -0.007 ± 0.031 |
| | LogNorm | 0.379 ± 0.202 | 0.770 ± 0.039 | **0.894 ± 0.005** | **0.850 ± 0.009** | 0.900 ± 0.801 | **0.994 ± 0.032** | 0.017 ± 0.030 | 1.036 ± 0.087 | -0.048 ± 0.110 |
| | DeepSurv | **0.254 ± 0.036** | 0.787 ± 0.015 | 0.889 ± 0.004 | 0.837 ± 0.025 | 0.227 ± 0.053 | 1.033 ± 0.014 | -0.014 ± 0.011 | 1.010 ± 0.016 | -0.010 ± 0.018 |
| | DeepHit | 1.303 ± 0.132 | 0.572 ± 0.032 | 0.882 ± 0.006 | 0.832 ± 0.017 | 1.798 ± 0.770 | 1.041 ± 0.049 | 0.060 ± 0.047 | 1.142 ± 0.041 | -0.124 ± 0.059 |
| | GBM | 1.534 ± 0.062 | 0.142 ± 0.005 | 0.838 ± 0.009 | 0.798 ± 0.007 | 0.130 ± 0.042 | 0.991 ± 0.019 | 0.029 ± 0.011 | 1.038 ± 0.010 | -0.031 ± 0.009 |
| | RSF | 0.444 ± 0.020 | 0.076 ± 0.003 | 0.878 ± 0.005 | 0.828 ± 0.010 | **0.072 ± 0.030** | 1.014 ± 0.007 | **0.001 ± 0.009** | 1.005 ± 0.016 | 0.008 ± 0.016 |
| | DSM(LogNorm) | 2.135 ± 0.054 | 0.179 ± 0.006 | 0.651 ± 0.018 | 0.615 ± 0.016 | 3.858 ± 0.295 | 0.981 ± 0.015 | 0.049 ± 0.008 | 1.052 ± 0.013 | -0.030 ± 0.011 |
| | DSM(Weibull) | 2.182 ± 0.053 | 0.178 ± 0.006 | 0.738 ± 0.017 | 0.689 ± 0.013 | 3.834 ± 0.328 | 1.012 ± 0.013 | 0.011 ± 0.007 | 1.021 ± 0.012 | **-0.007 ± 0.011** |
| LogNorm heavy | ALD | **0.385 ± 0.193** | **0.095 ± 0.006** | **0.777 ± 0.012** | **0.727 ± 0.021** | 0.043 ± 0.019 | 1.003 ± 0.014 | -0.005 ± 0.005 | 0.998 ± 0.014 | **-0.003 ± 0.014** |
| | CQRNN | 0.717 ± 0.027 | 0.436 ± 0.035 | 0.767 ± 0.009 | 0.718 ± 0.018 | 0.235 ± 0.104 | 0.992 ± 0.026 | -0.007 ± 0.013 | 0.998 ± 0.032 | -0.019 ± 0.035 |
| | LogNorm | 0.755 ± 0.194 | 0.401 ± 0.012 | 0.643 ± 0.053 | 0.609 ± 0.046 | 0.066 ± 0.056 | 1.018 ± 0.012 | **-0.002 ± 0.005** | 1.011 ± 0.019 | -0.003 ± 0.020 |
| | DeepSurv | 0.842 ± 0.019 | 0.459 ± 0.013 | 0.497 ± 0.034 | 0.465 ± 0.029 | 0.102 ± 0.068 | 1.031 ± 0.008 | -0.010 ± 0.007 | 1.015 ± 0.017 | -0.018 ± 0.015 |
| | DeepHit | 0.724 ± 0.020 | 0.402 ± 0.012 | 0.756 ± 0.012 | 0.712 ± 0.019 | 0.282 ± 0.121 | 1.036 ± 0.020 | -0.012 ± 0.006 | 1.030 ± 0.014 | -0.045 ± 0.018 |
| | GBM | 0.689 ± 0.018 | 0.118 ± 0.006 | 0.666 ± 0.032 | 0.618 ± 0.029 | **0.030 ± 0.016** | 1.004 ± 0.009 | 0.003 ± 0.003 | 1.009 ± 0.011 | -0.007 ± 0.012 |
| | RSF | 0.695 ± 0.018 | 0.099 ± 0.006 | 0.715 ± 0.016 | 0.663 ± 0.018 | 0.031 ± 0.020 | 1.008 ± 0.007 | **0.002 ± 0.004** | 1.010 ± 0.011 | -0.006 ± 0.012 |
| | DSM(LogNorm) | 0.656 ± 0.016 | 0.125 ± 0.007 | 0.695 ± 0.014 | 0.654 ± 0.015 | 18.733 ± 1.302 | 0.994 ± 0.010 | -0.004 ± 0.003 | 0.984 ± 0.012 | 0.020 ± 0.013 |
| | DSM(Weibull) | 0.680 ± 0.018 | 0.123 ± 0.007 | 0.697 ± 0.009 | 0.659 ± 0.026 | 18.884 ± 1.516 | 0.997 ± 0.011 | -0.007 ± 0.004 | 0.980 ± 0.013 | 0.021 ± 0.013 |
| LogNorm med. | ALD | **0.178 ± 0.046** | **0.174 ± 0.005** | **0.747 ± 0.004** | **0.718 ± 0.007** | 0.087 ± 0.052 | 1.008 ± 0.017 | -0.004 ± 0.010 | **1.002 ± 0.009** | -0.001 ± 0.012 |
| | CQRNN | 0.540 ± 0.059 | 0.368 ± 0.053 | 0.746 ± 0.005 | 0.716 ± 0.006 | 0.376 ± 0.166 | 0.985 ± 0.069 | **-0.001 ± 0.037** | 0.994 ± 0.035 | -0.006 ± 0.041 |
| | LogNorm | 0.549 ± 0.101 | 0.452 ± 0.012 | 0.694 ± 0.024 | 0.665 ± 0.021 | 0.085 ± 0.067 | **1.002 ± 0.016** | 0.007 ± 0.011 | 1.008 ± 0.022 | **0.000 ± 0.026** |
| | DeepSurv | 0.654 ± 0.029 | 0.545 ± 0.015 | 0.638 ± 0.011 | 0.596 ± 0.011 | 0.138 ± 0.058 | 1.020 ± 0.017 | -0.015 ± 0.009 | 0.994 ± 0.016 | 0.005 ± 0.018 |
| | DeepHit | 0.600 ± 0.018 | 0.426 ± 0.010 | 0.729 ± 0.019 | 0.702 ± 0.015 | 0.344 ± 0.118 | 1.046 ± 0.018 | -0.032 ± 0.017 | 0.986 ± 0.018 | 0.018 ± 0.020 |
| | GBM | 0.626 ± 0.016 | 0.205 ± 0.008 | 0.711 ± 0.008 | 0.684 ± 0.008 | 0.067 ± 0.012 | 1.006 ± 0.017 | 0.008 ± 0.011 | 1.017 ± 0.011 | -0.014 ± 0.014 |
| | RSF | 0.506 ± 0.016 | 0.180 ± 0.008 | 0.733 ± 0.008 | 0.702 ± 0.005 | **0.060 ± 0.028** | 1.015 ± 0.017 | 0.002 ± 0.012 | 1.014 ± 0.010 | -0.005 ± 0.014 |
| | DSM(LogNorm) | 0.643 ± 0.014 | 0.223 ± 0.008 | 0.720 ± 0.006 | 0.694 ± 0.008 | 7.020 ± 0.465 | 0.996 ± 0.022 | -0.003 ± 0.014 | 0.983 ± 0.013 | 0.025 ± 0.015 |
| | DSM(Weibull) | 0.637 ± 0.014 | 0.221 ± 0.008 | 0.670 ± 0.011 | 0.648 ± 0.009 | 7.371 ± 0.618 | 1.039 ± 0.021 | -0.036 ± 0.014 | 0.973 ± 0.012 | 0.030 ± 0.014 |
| LogNorm light | ALD | **0.184 ± 0.035** | **0.310 ± 0.011** | **0.725 ± 0.007** | 0.713 ± 0.008 | 0.185 ± 0.095 | 0.985 ± 0.015 | 0.007 ± 0.009 | **1.001 ± 0.014** | -0.001 ± 0.017 |
| | CQRNN | 0.356 ± 0.073 | 0.418 ± 0.045 | **0.725 ± 0.007** | **0.714 ± 0.008** | 0.976 ± 0.602 | 0.988 ± 0.077 | -0.012 ± 0.045 | 0.962 ± 0.071 | 0.044 ± 0.072 |
| | LogNorm | 0.311 ± 0.022 | 0.794 ± 0.026 | 0.709 ± 0.009 | 0.698 ± 0.010 | 0.231 ± 0.170 | 0.972 ± 0.027 | -0.007 ± 0.014 | 0.964 ± 0.035 | 0.029 ± 0.041 |
| | DeepSurv | 0.403 ± 0.027 | 0.833 ± 0.025 | 0.715 ± 0.009 | 0.700 ± 0.011 | 0.211 ± 0.123 | 1.010 ± 0.017 | **-0.000 ± 0.012** | 1.004 ± 0.017 | 0.005 ± 0.018 |
| | DeepHit | 0.581 ± 0.018 | 0.654 ± 0.017 | 0.702 ± 0.008 | 0.692 ± 0.008 | 0.253 ± 0.174 | 1.006 ± 0.030 | -0.013 ± 0.016 | 0.974 ± 0.021 | 0.042 ± 0.026 |
| | GBM | 0.616 ± 0.017 | 0.363 ± 0.010 | 0.693 ± 0.005 | 0.683 ± 0.006 | **0.104 ± 0.042** | 1.007 ± 0.029 | 0.009 ± 0.017 | 1.022 ± 0.016 | -0.019 ± 0.018 |
| | RSF | 0.442 ± 0.019 | 0.317 ± 0.009 | 0.714 ± 0.006 | 0.703 ± 0.007 | 0.114 ± 0.032 | 1.025 ± 0.020 | **-0.000 ± 0.012** | 1.017 ± 0.018 | -0.006 ± 0.020 |
| | DSM(LogNorm) | 0.643 ± 0.014 | 0.386 ± 0.010 | 0.698 ± 0.007 | 0.688 ± 0.008 | 1.809 ± 0.275 | **0.997 ± 0.029** | 0.002 ± 0.017 | 0.988 ± 0.018 | 0.026 ± 0.019 |
| | DSM(Weibull) | 0.622 ± 0.014 | 0.384 ± 0.011 | 0.647 ± 0.010 | 0.639 ± 0.010 | 2.292 ± 0.343 | 1.037 ± 0.028 | -0.013 ± 0.016 | 0.998 ± 0.018 | 0.023 ± 0.019 |
| LogNorm same | ALD | **0.191 ± 0.044** | **0.154 ± 0.006** | 0.739 ± 0.009 | 0.697 ± 0.008 | 0.076 ± 0.057 | 1.012 ± 0.011 | **-0.001 ± 0.008** | 1.009 ± 0.011 | -0.005 ± 0.010 |
| | CQRNN | 0.319 ± 0.079 | 0.300 ± 0.049 | 0.740 ± 0.008 | 0.698 ± 0.009 | 0.787 ± 0.336 | 0.986 ± 0.086 | -0.003 ± 0.040 | 0.971 ± 0.058 | 0.041 ± 0.056 |
| | LogNorm | 0.273 ± 0.068 | 0.528 ± 0.017 | 0.736 ± 0.012 | 0.695 ± 0.010 | 0.213 ± 0.117 | 0.972 ± 0.015 | -0.006 ± 0.010 | 0.963 ± 0.028 | 0.033 ± 0.040 |
| | DeepSurv | 0.362 ± 0.026 | 0.511 ± 0.012 | **0.743 ± 0.010** | **0.700 ± 0.007** | 0.138 ± 0.040 | 1.017 ± 0.013 | -0.005 ± 0.012 | **1.004 ± 0.014** | **0.001 ± 0.013** |
| | DeepHit | 0.560 ± 0.098 | 0.385 ± 0.022 | 0.652 ± 0.066 | 0.633 ± 0.047 | 1.265 ± 1.911 | 0.925 ± 0.071 | -0.010 ± 0.011 | 0.925 ± 0.088 | 0.058 ± 0.108 |
| | GBM | 0.571 ± 0.023 | 0.196 ± 0.006 | 0.696 ± 0.011 | 0.661 ± 0.011 | **0.064 ± 0.031** | 1.011 ± 0.007 | 0.008 ± 0.009 | 1.023 ± 0.010 | -0.017 ± 0.007 |
| | RSF | 0.397 ± 0.019 | 0.167 ± 0.007 | 0.724 ± 0.011 | 0.682 ± 0.010 | 0.123 ± 0.045 | 1.006 ± 0.012 | 0.024 ± 0.012 | 1.032 ± 0.010 | -0.009 ± 0.010 |
| | DSM(LogNorm) | 0.606 ± 0.025 | 0.215 ± 0.006 | 0.678 ± 0.026 | 0.646 ± 0.020 | 4.464 ± 0.674 | **0.996 ± 0.014** | -0.007 ± 0.007 | 0.978 ± 0.013 | 0.029 ± 0.012 |
| | DSM(Weibull) | 0.597 ± 0.024 | 0.216 ± 0.006 | 0.639 ± 0.013 | 0.612 ± 0.010 | 4.947 ± 0.520 | 1.036 ± 0.011 | -0.025 ± 0.007 | 0.983 ± 0.009 | 0.029 ± 0.008 |

# Learning Survival Distributions with the Asymmetric Laplace Distribution

| Dataset | Method | MAE | IBS | Harrell's C-index | Uno's C-index | CensDcal | Cal$[S(t\|x)]$(Slope) | Cal$[S(t\|x)]$(Intercept) | Cal$[f(t\|x)]$(Slope) | Cal$[f(t\|x)]$(Intercept) |
|---|---|---|---|---|---|---|---|---|---|---|
| METABRIC | ALD | 1.626 ± 0.194 | 0.245 ± 0.012 | 0.637 ± 0.021 | 0.633 ± 0.031 | 0.293 ± 0.125 | **1.001 ± 0.033** | -0.011 ± 0.016 | 0.993 ± 0.028 | -0.009 ± 0.024 |
| | CQRNN | 0.998 ± 0.074 | 0.344 ± 0.027 | 0.632 ± 0.017 | 0.630 ± 0.033 | 0.641 ± 0.391 | 0.972 ± 0.048 | 0.007 ± 0.018 | **1.001 ± 0.042** | -0.022 ± 0.051 |
| | LogNorm | 1.329 ± 0.041 | 0.526 ± 0.015 | 0.609 ± 0.019 | 0.613 ± 0.046 | 0.619 ± 0.247 | 0.964 ± 0.026 | -0.024 ± 0.011 | 0.937 ± 0.017 | 0.040 ± 0.018 |
| | DeepSurv | 0.981 ± 0.029 | 0.533 ± 0.019 | **0.645 ± 0.016** | **0.635 ± 0.035** | **0.159 ± 0.075** | 1.009 ± 0.011 | -0.008 ± 0.013 | 1.003 ± 0.022 | -0.010 ± 0.023 |
| | DeepHit | 1.177 ± 0.065 | 0.462 ± 0.008 | 0.563 ± 0.040 | 0.577 ± 0.053 | 0.659 ± 0.212 | 1.070 ± 0.022 | -0.036 ± 0.014 | 1.018 ± 0.015 | -0.024 ± 0.029 |
| | GBM | **0.949 ± 0.043** | 0.250 ± 0.007 | 0.639 ± 0.018 | 0.632 ± 0.042 | 0.199 ± 0.049 | 1.016 ± 0.025 | **-0.001 ± 0.019** | 1.017 ± 0.021 | -0.018 ± 0.021 |
| | RSF | 1.079 ± 0.059 | **0.241 ± 0.008** | 0.632 ± 0.017 | 0.619 ± 0.043 | 0.180 ± 0.071 | 1.020 ± 0.022 | -0.004 ± 0.019 | 1.011 ± 0.020 | -0.007 ± 0.020 |
| | DSM(LogNorm) | 0.980 ± 0.031 | 0.265 ± 0.007 | 0.613 ± 0.024 | 0.591 ± 0.051 | 4.601 ± 0.307 | 1.002 ± 0.026 | 0.004 ± 0.016 | 1.004 ± 0.018 | **0.001 ± 0.018** |
| | DSM(Weibull) | 1.025 ± 0.032 | 0.266 ± 0.007 | 0.602 ± 0.025 | 0.598 ± 0.024 | 4.998 ± 0.510 | 1.042 ± 0.023 | -0.035 ± 0.014 | 0.982 ± 0.019 | 0.016 ± 0.019 |
| WHAS | ALD | 2.196 ± 0.612 | 0.134 ± 0.013 | 0.823 ± 0.016 | 0.824 ± 0.014 | **0.198 ± 0.094** | 0.972 ± 0.027 | 0.003 ± 0.016 | 0.981 ± 0.021 | 0.009 ± 0.023 |
| | CQRNN | 0.798 ± 0.249 | 0.636 ± 0.018 | 0.838 ± 0.016 | 0.846 ± 0.016 | 0.564 ± 0.248 | 0.974 ± 0.060 | **0.002 ± 0.022** | 0.998 ± 0.057 | -0.024 ± 0.053 |
| | LogNorm | 1.976 ± 0.232 | 0.614 ± 0.025 | 0.600 ± 0.042 | 0.575 ± 0.039 | 0.584 ± 0.233 | 0.920 ± 0.032 | 0.041 ± 0.021 | 0.994 ± 0.032 | -0.002 ± 0.033 |
| | DeepSurv | 0.867 ± 0.050 | 0.699 ± 0.023 | 0.711 ± 0.014 | 0.637 ± 0.025 | 0.228 ± 0.101 | **0.997 ± 0.020** | 0.005 ± 0.018 | 1.012 ± 0.020 | -0.019 ± 0.019 |
| | DeepHit | 0.966 ± 0.077 | 0.604 ± 0.023 | 0.806 ± 0.018 | 0.811 ± 0.018 | 0.269 ± 0.172 | 0.963 ± 0.036 | 0.018 ± 0.022 | 1.008 ± 0.026 | -0.015 ± 0.031 |
| | GBM | 1.157 ± 0.073 | 0.165 ± 0.009 | 0.817 ± 0.018 | 0.816 ± 0.015 | 0.363 ± 0.174 | 1.055 ± 0.034 | 0.042 ± 0.020 | 1.076 ± 0.022 | -0.044 ± 0.023 |
| | RSF | **0.603 ± 0.055** | **0.082 ± 0.011** | **0.866 ± 0.017** | **0.898 ± 0.015** | 11.243 ± 3.013 | 3.177 ± 0.275 | 0.163 ± 0.049 | 1.607 ± 0.091 | 0.113 ± 0.024 |
| | DSM(LogNorm) | 2.055 ± 0.175 | 0.211 ± 0.009 | 0.779 ± 0.016 | 0.788 ± 0.015 | 11.200 ± 1.181 | 0.908 ± 0.026 | 0.070 ± 0.021 | 1.015 ± 0.025 | **-0.001 ± 0.027** |
| | DSM(Weibull) | 1.748 ± 0.127 | 0.208 ± 0.010 | 0.786 ± 0.023 | 0.794 ± 0.017 | 10.569 ± 1.295 | 0.907 ± 0.028 | 0.064 ± 0.022 | **1.001 ± 0.024** | 0.010 ± 0.026 |
| SUPPORT | ALD | 1.121 ± 0.107 | 0.362 ± 0.013 | 0.568 ± 0.015 | 0.572 ± 0.015 | 2.197 ± 0.667 | 1.084 ± 0.043 | -0.113 ± 0.023 | 0.900 ± 0.056 | 0.084 ± 0.046 |
| | CQRNN | 0.659 ± 0.047 | 0.344 ± 0.007 | 0.612 ± 0.005 | 0.613 ± 0.006 | 0.724 ± 0.428 | 1.034 ± 0.066 | -0.019 ± 0.034 | 0.992 ± 0.051 | 0.019 ± 0.049 |
| | LogNorm | 1.311 ± 0.150 | 0.688 ± 0.020 | 0.597 ± 0.011 | 0.597 ± 0.011 | 2.792 ± 0.942 | 0.942 ± 0.040 | -0.114 ± 0.008 | 0.769 ± 0.041 | 0.169 ± 0.042 |
| | DeepSurv | 0.511 ± 0.021 | 0.629 ± 0.014 | 0.599 ± 0.008 | 0.597 ± 0.009 | **0.092 ± 0.036** | **0.989 ± 0.016** | -0.003 ± 0.011 | 0.986 ± 0.012 | **0.006 ± 0.014** |
| | DeepHit | 0.574 ± 0.034 | 0.530 ± 0.009 | 0.577 ± 0.008 | 0.582 ± 0.009 | 0.829 ± 0.213 | 0.891 ± 0.013 | 0.006 ± 0.008 | 0.909 ± 0.014 | 0.086 ± 0.021 |
| | GBM | **0.427 ± 0.008** | 0.359 ± 0.005 | 0.597 ± 0.006 | 0.601 ± 0.007 | 0.111 ± 0.041 | 1.065 ± 0.018 | -0.010 ± 0.011 | **1.005 ± 0.010** | 0.019 ± 0.015 |
| | RSF | 0.647 ± 0.027 | **0.339 ± 0.005** | **0.619 ± 0.006** | **0.618 ± 0.006** | 0.122 ± 0.058 | 1.132 ± 0.013 | -0.027 ± 0.009 | 1.006 ± 0.010 | 0.037 ± 0.015 |
| | DSM(LogNorm) | 0.512 ± 0.012 | 0.379 ± 0.006 | 0.570 ± 0.007 | 0.570 ± 0.007 | 16.664 ± 1.442 | 1.147 ± 0.011 | -0.145 ± 0.010 | 0.928 ± 0.009 | 0.057 ± 0.012 |
| | DSM(Weibull) | 0.581 ± 0.013 | 0.375 ± 0.007 | 0.565 ± 0.010 | 0.569 ± 0.010 | 14.725 ± 1.066 | 1.209 ± 0.010 | -0.147 ± 0.010 | 0.941 ± 0.009 | 0.056 ± 0.012 |
| GBSG | ALD | 1.713 ± 0.208 | **0.279 ± 0.014** | 0.671 ± 0.013 | 0.665 ± 0.013 | 0.283 ± 0.106 | **1.000 ± 0.035** | -0.018 ± 0.016 | 0.977 ± 0.025 | 0.014 ± 0.034 |
| | CQRNN | 0.865 ± 0.070 | 0.357 ± 0.021 | **0.680 ± 0.015** | **0.672 ± 0.014** | 0.573 ± 0.577 | 0.953 ± 0.043 | -0.008 ± 0.016 | 0.967 ± 0.030 | **0.002 ± 0.040** |
| | LogNorm | 1.469 ± 0.105 | 0.577 ± 0.015 | 0.660 ± 0.012 | 0.653 ± 0.012 | 0.817 ± 0.303 | 0.968 ± 0.025 | -0.057 ± 0.011 | 0.886 ± 0.025 | 0.086 ± 0.035 |
| | DeepSurv | **0.709 ± 0.036** | 0.569 ± 0.016 | 0.611 ± 0.017 | 0.602 ± 0.015 | 0.180 ± 0.126 | 1.002 ± 0.021 | **-0.003 ± 0.013** | **0.996 ± 0.013** | 0.004 ± 0.018 |
| | DeepHit | 0.773 ± 0.037 | 0.495 ± 0.016 | 0.649 ± 0.016 | 0.644 ± 0.016 | 2.020 ± 1.450 | 0.967 ± 0.049 | -0.045 ± 0.014 | 0.952 ± 0.031 | -0.025 ± 0.016 |
| | GBM | 0.837 ± 0.037 | 0.289 ± 0.008 | 0.669 ± 0.013 | 0.662 ± 0.012 | **0.165 ± 0.139** | 1.006 ± 0.022 | 0.009 ± 0.017 | 1.018 ± 0.020 | -0.012 ± 0.020 |
| | RSF | 0.900 ± 0.052 | 0.283 ± 0.011 | 0.658 ± 0.010 | 0.648 ± 0.010 | 0.192 ± 0.054 | 0.995 ± 0.029 | -0.007 ± 0.016 | 0.984 ± 0.024 | 0.011 ± 0.026 |
| | DSM(LogNorm) | 0.989 ± 0.025 | 0.308 ± 0.009 | 0.636 ± 0.023 | 0.629 ± 0.020 | 7.079 ± 0.384 | 1.048 ± 0.017 | -0.052 ± 0.010 | 0.960 ± 0.015 | 0.034 ± 0.017 |
| | DSM(Weibull) | 1.075 ± 0.026 | 0.308 ± 0.009 | 0.638 ± 0.011 | 0.632 ± 0.011 | 8.698 ± 0.718 | 1.102 ± 0.015 | -0.077 ± 0.009 | 0.968 ± 0.014 | 0.031 ± 0.016 |
| TMBImmuno | ALD | 3.002 ± 1.497 | 0.245 ± 0.015 | 0.561 ± 0.037 | 0.547 ± 0.040 | 0.835 ± 0.604 | 1.053 ± 0.045 | -0.038 ± 0.025 | **0.994 ± 0.021** | 0.004 ± 0.025 |
| | CQRNN | 1.008 ± 0.053 | 0.272 ± 0.013 | 0.567 ± 0.022 | **0.557 ± 0.017** | 0.251 ± 0.123 | 0.967 ± 0.037 | 0.011 ± 0.026 | 0.988 ± 0.027 | 0.009 ± 0.020 |
| | LogNorm | 1.880 ± 0.156 | 0.420 ± 0.011 | 0.561 ± 0.028 | **0.557 ± 0.028** | 0.617 ± 0.196 | 0.949 ± 0.028 | -0.027 ± 0.019 | 0.913 ± 0.025 | 0.066 ± 0.027 |
| | DeepSurv | 0.948 ± 0.097 | 0.395 ± 0.012 | 0.543 ± 0.034 | 0.526 ± 0.039 | **0.246 ± 0.168** | 1.019 ± 0.030 | **-0.001 ± 0.023** | 1.009 ± 0.020 | **-0.003 ± 0.018** |
| | DeepHit | 1.117 ± 0.141 | 0.400 ± 0.011 | 0.560 ± 0.023 | 0.554 ± 0.021 | 0.464 ± 0.214 | 0.963 ± 0.039 | -0.018 ± 0.026 | 0.935 ± 0.020 | 0.058 ± 0.026 |
| | GBM | **0.897 ± 0.032** | **0.243 ± 0.009** | **0.576 ± 0.021** | 0.555 ± 0.023 | 0.694 ± 0.279 | 1.320 ± 0.057 | -0.019 ± 0.034 | 1.049 ± 0.028 | 0.079 ± 0.026 |
| | RSF | 1.607 ± 0.069 | 0.263 ± 0.008 | 0.550 ± 0.017 | 0.536 ± 0.018 | 0.448 ± 0.112 | 1.266 ± 0.046 | -0.018 ± 0.019 | 1.017 ± 0.021 | 0.081 ± 0.019 |
| | DSM(LogNorm) | 0.956 ± 0.018 | 0.248 ± 0.009 | 0.518 ± 0.033 | 0.522 ± 0.033 | 7.414 ± 0.791 | **1.005 ± 0.022** | -0.028 ± 0.019 | 0.963 ± 0.016 | 0.035 ± 0.015 |
| | DSM(Weibull) | 1.020 ± 0.020 | 0.248 ± 0.009 | 0.551 ± 0.026 | 0.542 ± 0.029 | 9.625 ± 1.314 | 1.077 ± 0.017 | -0.053 ± 0.015 | 0.984 ± 0.016 | 0.021 ± 0.016 |
| BreastMSK | ALD | 2.593 ± 0.289 | **0.086 ± 0.008** | 0.617 ± 0.032 | 0.568 ± 0.035 | **0.066 ± 0.027** | 1.002 ± 0.019 | -0.007 ± 0.010 | 0.993 ± 0.020 | **0.003 ± 0.021** |
| | CQRNN | 1.864 ± 0.354 | 0.316 ± 0.035 | 0.599 ± 0.044 | 0.561 ± 0.036 | 0.172 ± 0.083 | 0.993 ± 0.036 | -0.005 ± 0.013 | 0.990 ± 0.030 | **0.003 ± 0.034** |
| | LogNorm | 6.675 ± 0.597 | 0.310 ± 0.015 | 0.610 ± 0.029 | 0.573 ± 0.046 | 0.208 ± 0.089 | 1.044 ± 0.010 | -0.004 ± 0.009 | 1.031 ± 0.012 | -0.023 ± 0.015 |
| | DeepSurv | 1.639 ± 0.217 | 0.334 ± 0.018 | 0.614 ± 0.033 | **0.582 ± 0.049** | 0.212 ± 0.099 | 1.046 ± 0.024 | -0.006 ± 0.011 | 1.036 ± 0.019 | -0.036 ± 0.019 |
| | DeepHit | **1.523 ± 0.076** | 0.303 ± 0.016 | 0.614 ± 0.036 | 0.563 ± 0.046 | 0.411 ± 0.213 | 1.062 ± 0.011 | -0.021 ± 0.006 | 1.032 ± 0.011 | -0.040 ± 0.014 |
| | GBM | 1.598 ± 0.087 | 0.091 ± 0.007 | **0.635 ± 0.030** | **0.582 ± 0.026** | 0.071 ± 0.053 | 1.011 ± 0.007 | **0.000 ± 0.007** | 1.006 ± 0.015 | -0.003 ± 0.017 |
| | RSF | 1.640 ± 0.163 | 0.087 ± 0.008 | 0.628 ± 0.037 | 0.575 ± 0.033 | 0.077 ± 0.057 | 1.016 ± 0.033 | 0.002 ± 0.009 | **1.001 ± 0.020** | 0.005 ± 0.017 |
| | DSM(LogNorm) | 1.613 ± 0.088 | 0.095 ± 0.008 | 0.622 ± 0.051 | 0.548 ± 0.036 | 13.508 ± 1.830 | **1.002 ± 0.013** | -0.010 ± 0.005 | 0.983 ± 0.015 | 0.014 ± 0.017 |
| | DSM(Weibull) | 1.643 ± 0.075 | 0.097 ± 0.008 | 0.620 ± 0.047 | 0.549 ± 0.027 | 15.107 ± 1.812 | 1.018 ± 0.014 | -0.021 ± 0.006 | 0.978 ± 0.015 | 0.017 ± 0.017 |
| LCGGBM | ALD | 1.232 ± 0.325 | **0.108 ± 0.011** | 0.778 ± 0.021 | 0.736 ± 0.030 | 0.450 ± 0.267 | 0.995 ± 0.047 | 0.003 ± 0.022 | **0.996 ± 0.038** | **0.009 ± 0.040** |
| | CQRNN | 0.808 ± 0.197 | 0.375 ± 0.041 | 0.790 ± 0.024 | 0.754 ± 0.034 | 0.543 ± 0.273 | 0.989 ± 0.071 | **0.001 ± 0.037** | 0.990 ± 0.052 | 0.011 ± 0.058 |
| | LogNorm | 1.191 ± 0.214 | 0.382 ± 0.017 | **0.795 ± 0.022** | **0.758 ± 0.037** | 0.327 ± 0.190 | **1.005 ± 0.025** | 0.007 ± 0.026 | 1.020 ± 0.040 | -0.018 ± 0.044 |
| | DeepSurv | 0.785 ± 0.155 | 0.472 ± 0.024 | 0.728 ± 0.057 | 0.664 ± 0.079 | 0.481 ± 0.219 | 1.022 ± 0.027 | 0.002 ± 0.025 | 1.018 ± 0.040 | -0.012 ± 0.046 |
| | DeepHit | 2.062 ± 0.285 | 0.377 ± 0.024 | 0.769 ± 0.022 | 0.734 ± 0.035 | 1.176 ± 0.539 | 1.085 ± 0.034 | -0.052 ± 0.023 | 0.968 ± 0.035 | 0.066 ± 0.037 |
| | GBM | **0.621 ± 0.094** | 0.138 ± 0.012 | 0.765 ± 0.013 | 0.734 ± 0.028 | 0.392 ± 0.163 | 1.032 ± 0.029 | 0.009 ± 0.028 | 1.032 ± 0.035 | -0.018 ± 0.037 |
| | RSF | 1.046 ± 0.325 | 0.114 ± 0.016 | 0.759 ± 0.035 | 0.723 ± 0.031 | 0.333 ± 0.106 | 1.030 ± 0.043 | -0.005 ± 0.027 | 0.995 ± 0.038 | 0.024 ± 0.045 |
| | DSM(LogNorm) | 0.955 ± 0.083 | 0.169 ± 0.015 | 0.577 ± 0.048 | 0.623 ± 0.052 | 9.428 ± 1.991 | 1.026 ± 0.021 | -0.034 ± 0.013 | 0.964 ± 0.034 | 0.035 ± 0.036 |
| | DSM(Weibull) | 1.042 ± 0.082 | 0.171 ± 0.016 | 0.766 ± 0.023 | 0.741 ± 0.033 | 10.756 ± 2.509 | 1.079 ± 0.017 | -0.051 ± 0.012 | 0.987 ± 0.033 | 0.019 ± 0.034 |

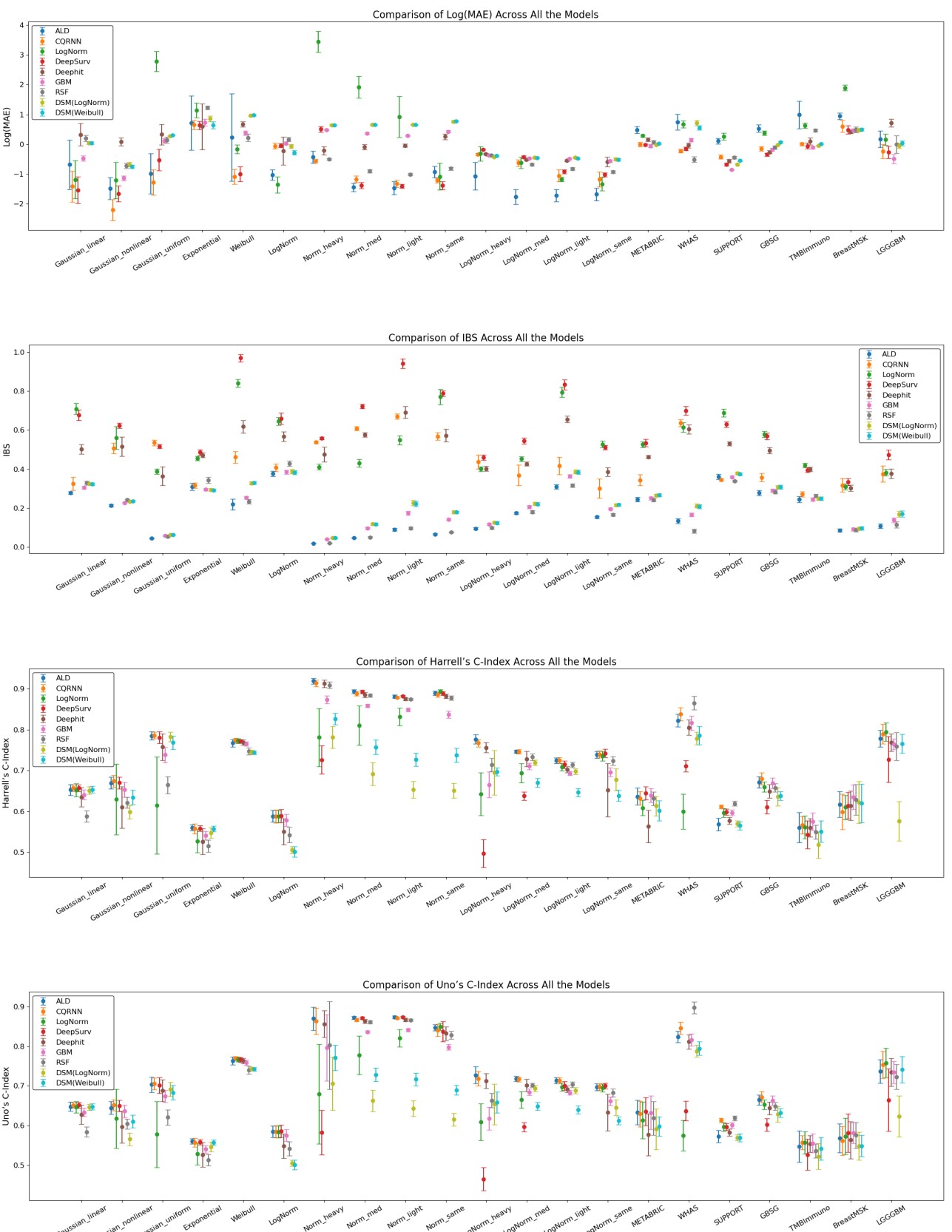

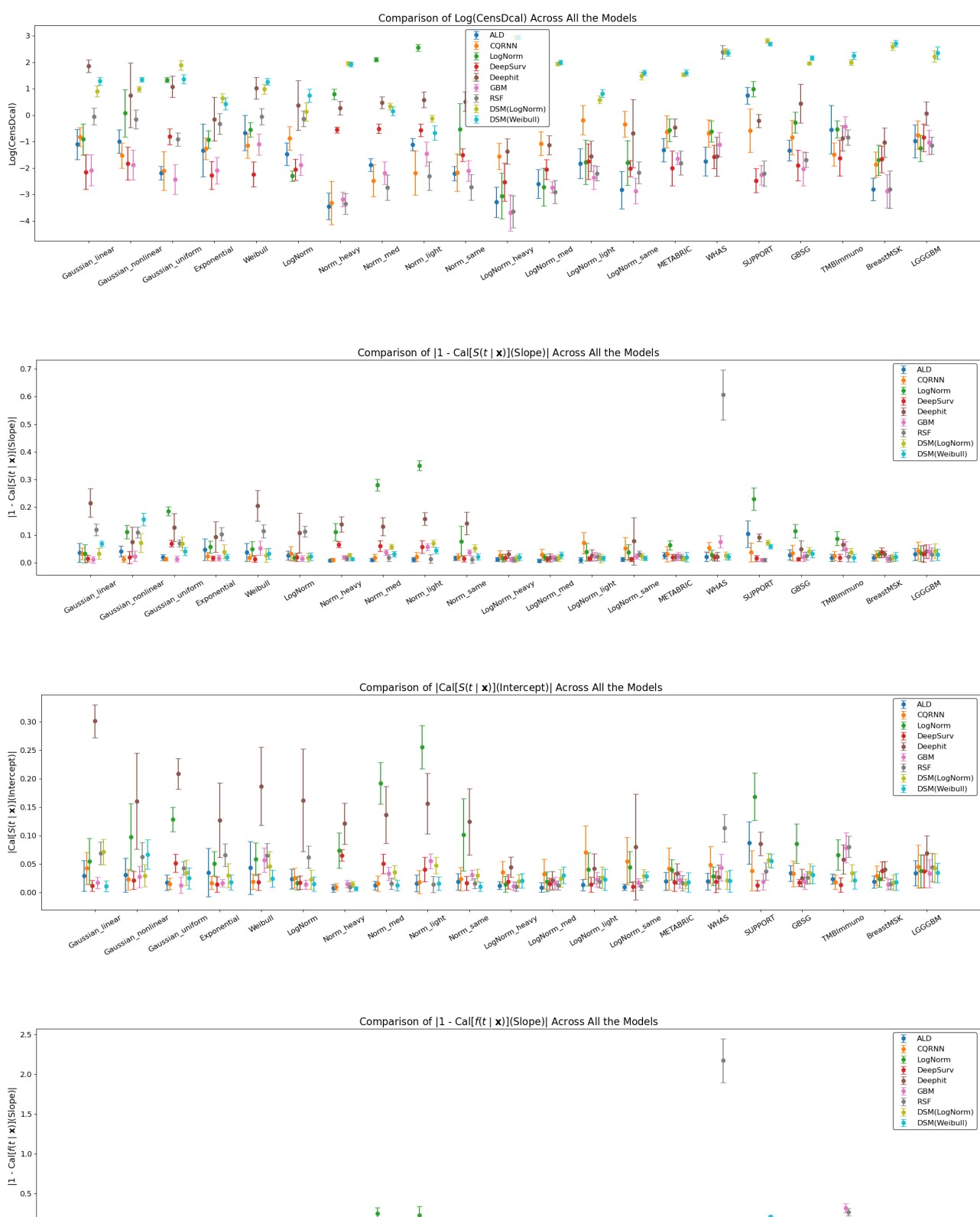

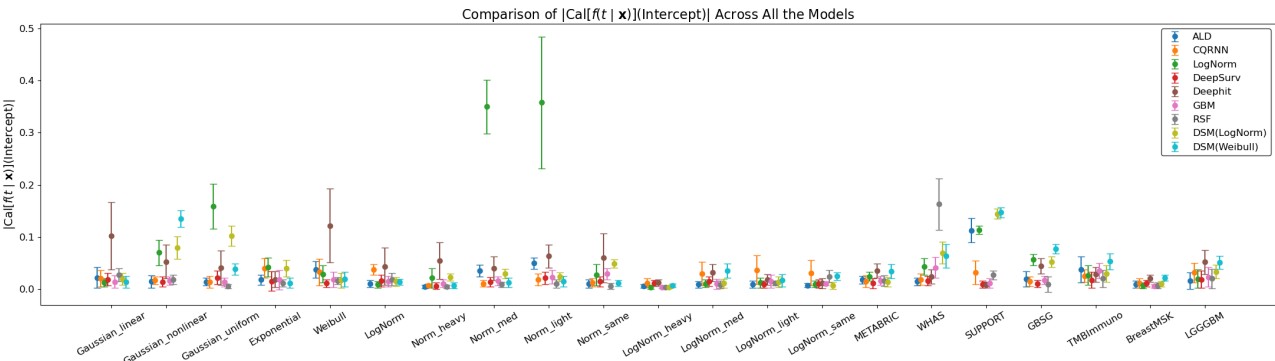

*Figure 4.* Performance on all datasets, methods, and metrics.

## C.2. Calibration Plots.

Figures 5 and 6 present the complete calibration curve fitting results, offering a comprehensive and visual assessment of model calibration performance. Specifically, Figure 5 compares our method (ALD) with CQRNN, LogNorm, DeepSurv, and DeepHit, while Figure 6 extends the comparison to GBM, RSF, and both DSM variants (LogNorm and Weibull). Each plot shows the observed event probabilities against the expected target proportions $[0.1, 0.2, \ldots, 0.9, 1.0]$. The ideal calibration corresponds to the identity line, where predictions align perfectly with the empirical outcomes. Deviations from this line highlight calibration errors. These plots provide intuitive insight into the degree and nature of miscalibration across models.

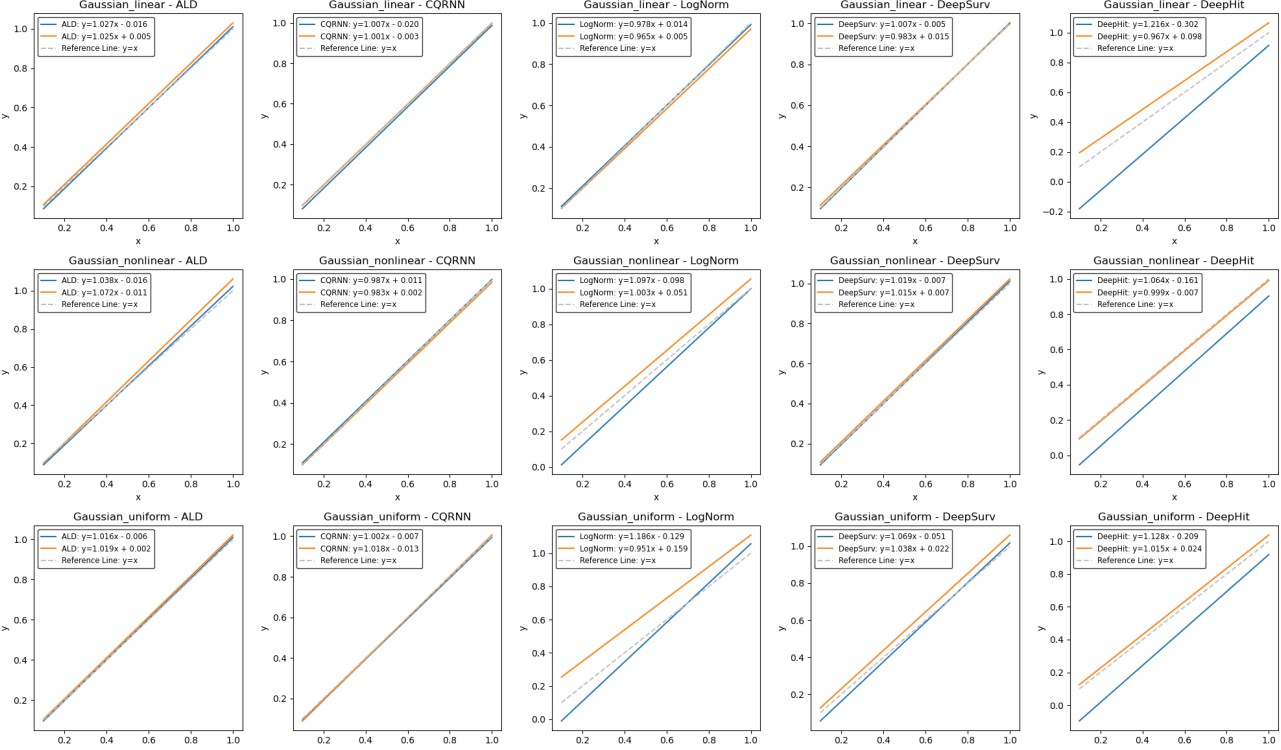

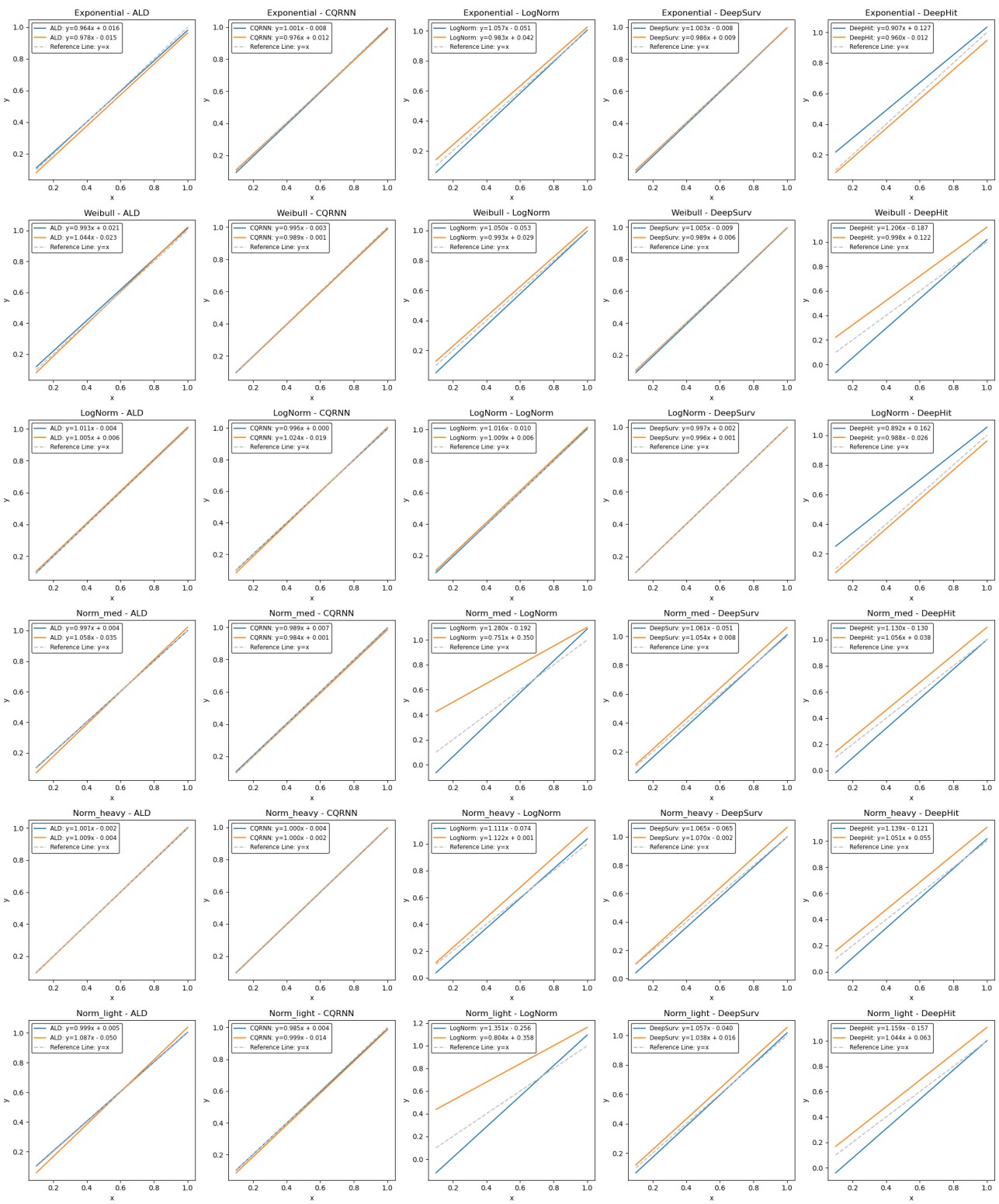

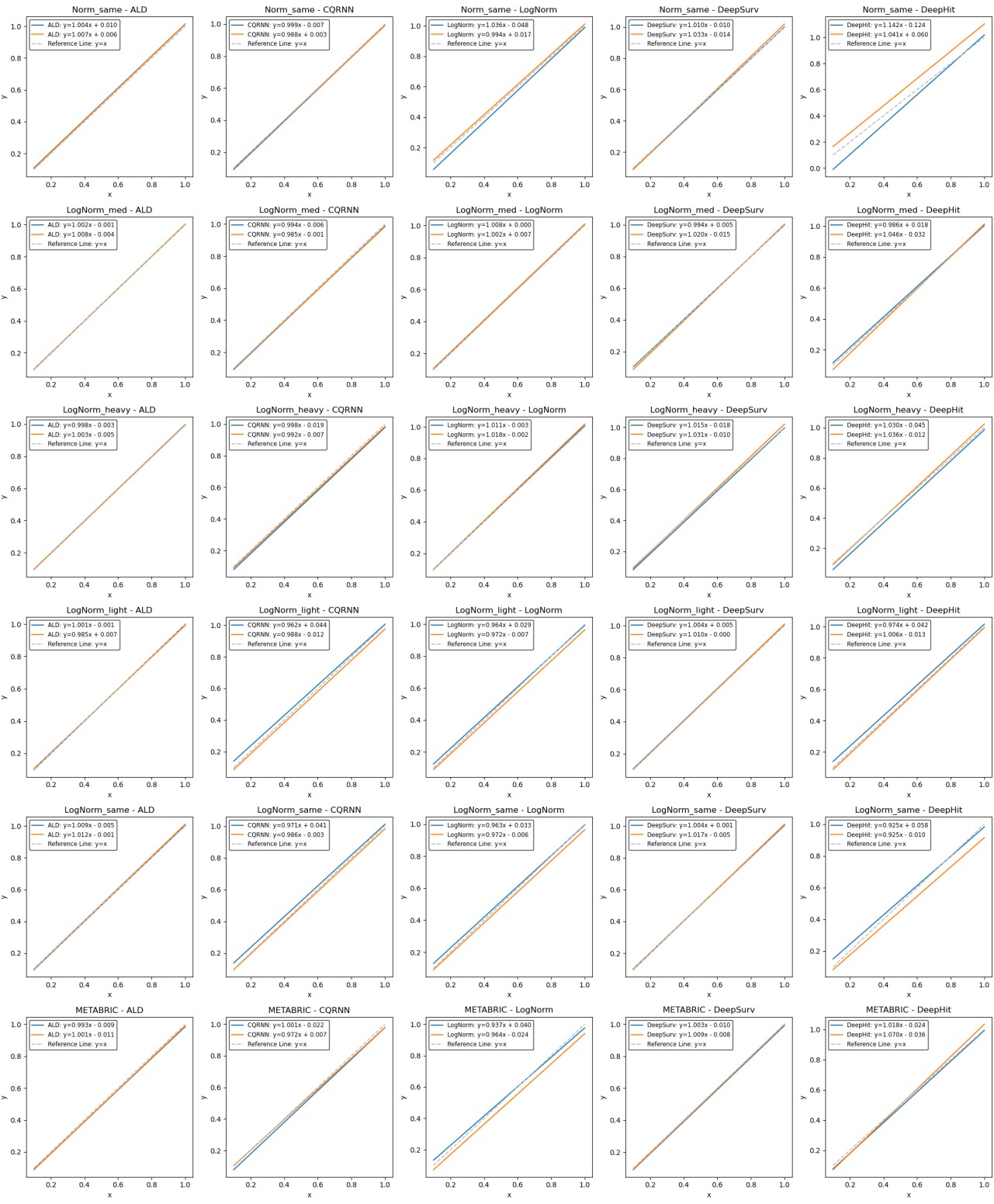

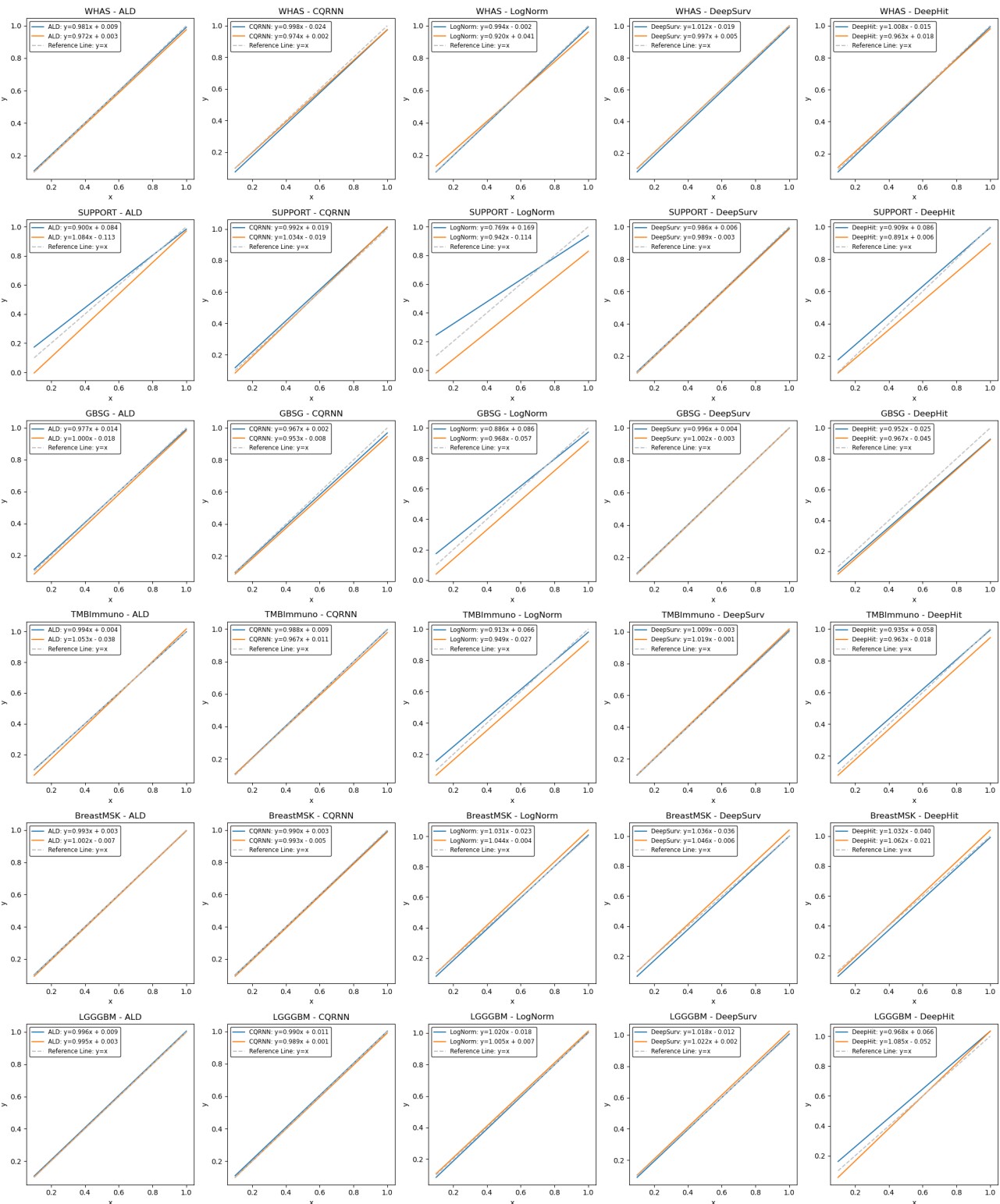

*Figure 5.* Calibration curves (linear fit) for ALD, CQRNN, LogNorm, DeepSurv and DeepHit. The blue and orange lines represent the curves for Cal$[S(t \mid \mathbf{x})]$ and Cal$[f(t \mid \mathbf{x})]$, respectively. The gray dashed line represents the idealized result where the slope is one and the intercept is zero.

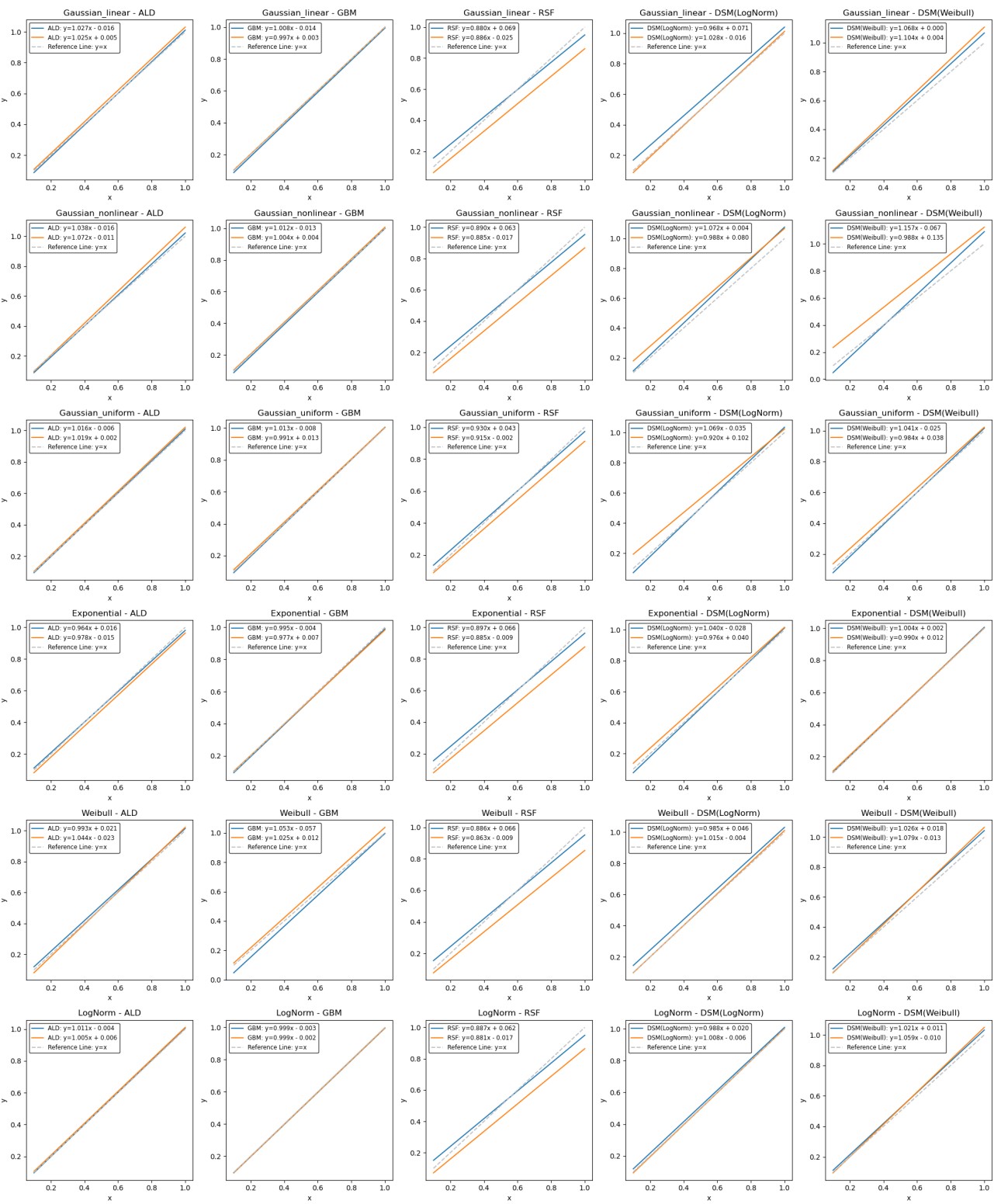

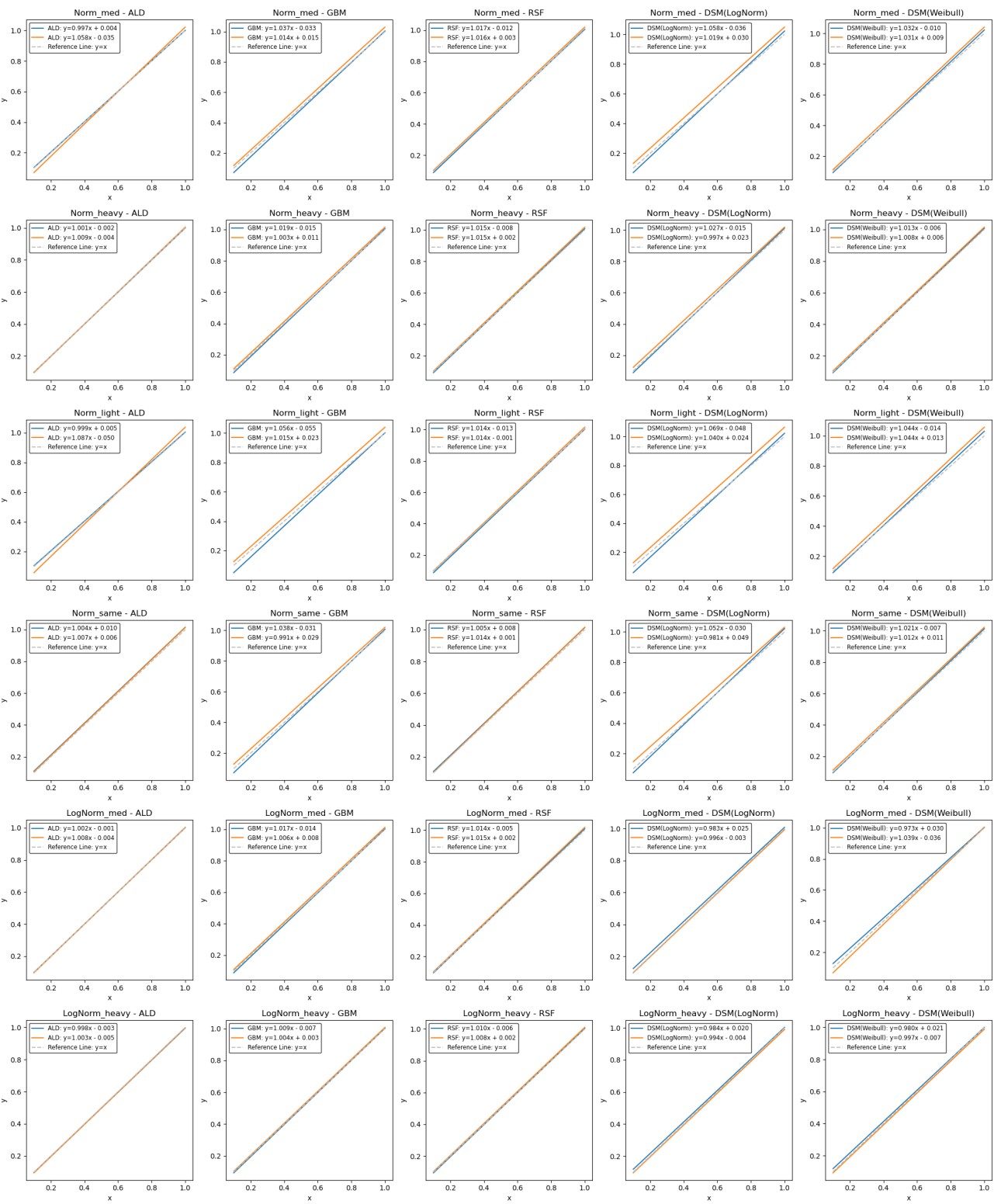

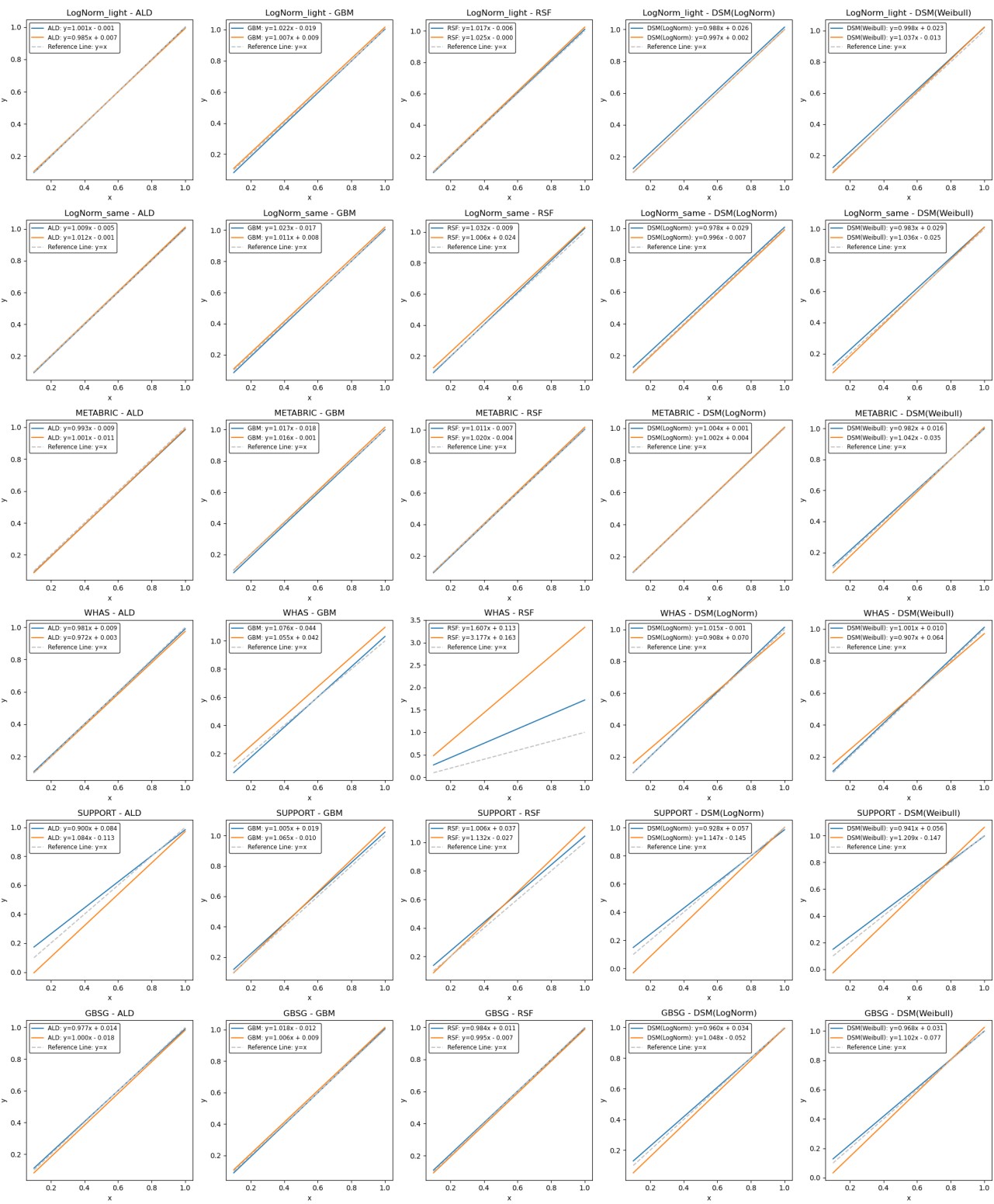

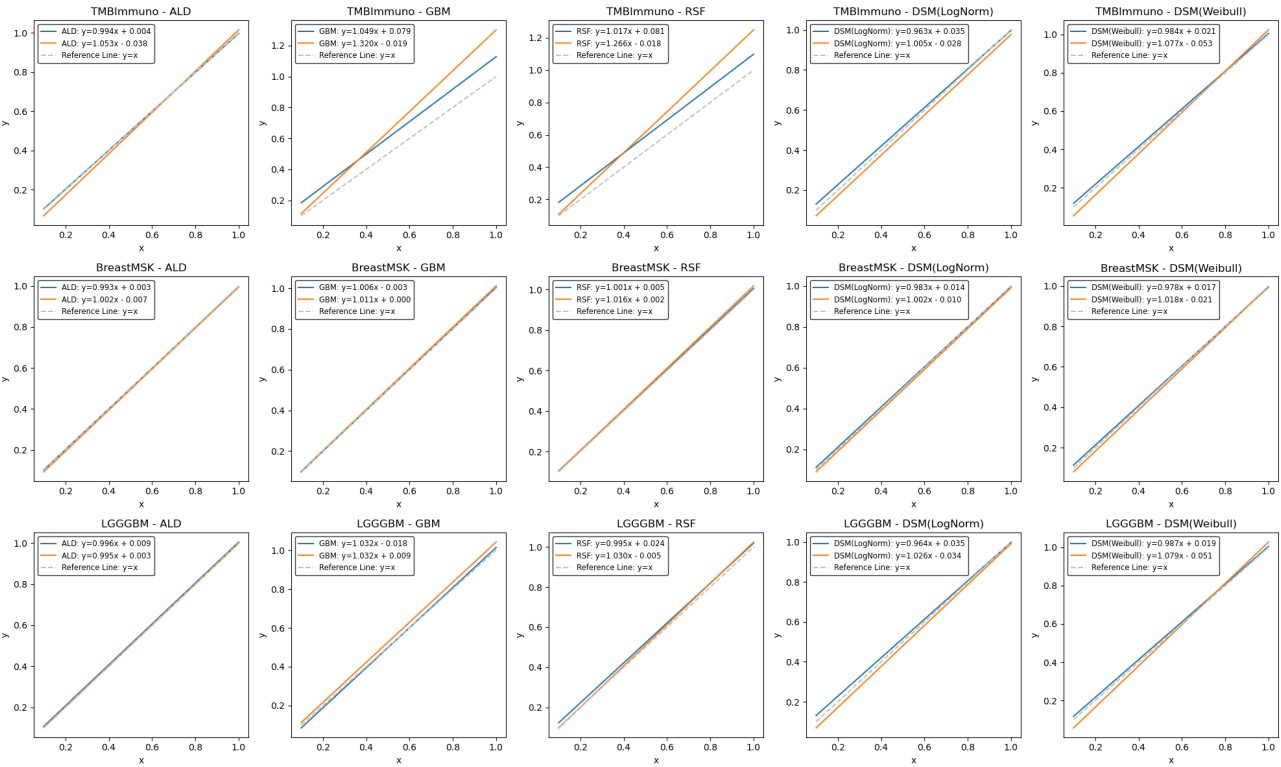

*Figure 6.* Calibration curves (linear fit) for ALD, GBM, RSF, DSM(LogNorm) and DSM(Weibull). The blue and orange lines represent the curves for $\mathrm{Cal}[S(t \mid \mathbf{x})]$ and $\mathrm{Cal}[f(t \mid \mathbf{x})]$, respectively. The gray dashed line represents the idealized result where the slope is one and the intercept is zero.

### C.3. Case Studies

**Case Study 1: Robustness Under High Censoring and Quantile Extremes**

To further assess the robustness of our model under varying levels of data censoring and across different event-time quantiles, we conduct experiments following the setup proposed by Nagpal et al. (2021). Specifically, we simulate additional censoring by uniformly sampling censoring times from the interval $[0, T]$ and applying them to a randomly selected subset of uncensored training samples, reducing the proportion of uncensored data to approximately 50% and 25%. Test splits remain unchanged to ensure unbiased estimation of the time-dependent concordance index ($C^{td}$). This procedure is applied to the METABRIC and SUPPORT datasets, which differ markedly in their censoring rates and event time distributions.

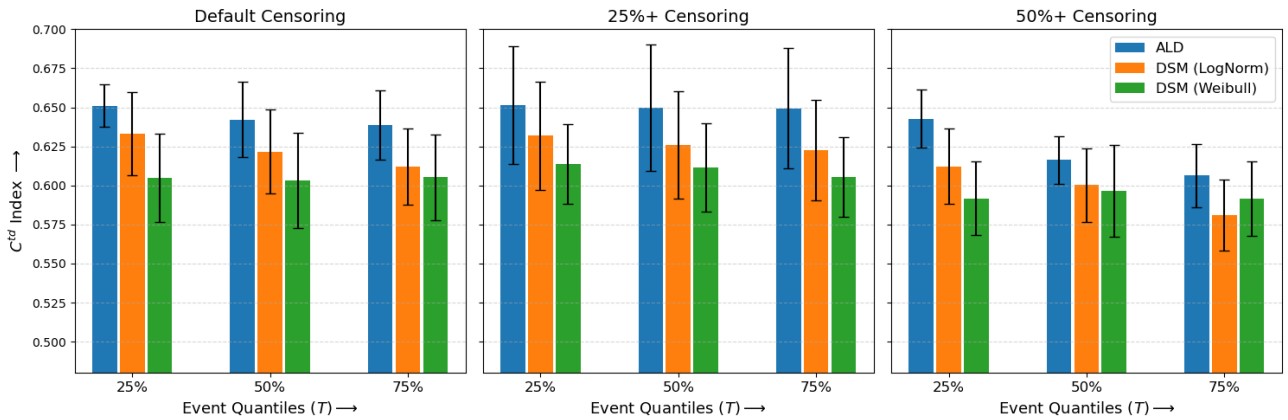

*Figure 7.* $C^{td}$ for METABRIC dataset at different quantiles of event times for different levels of censoring.

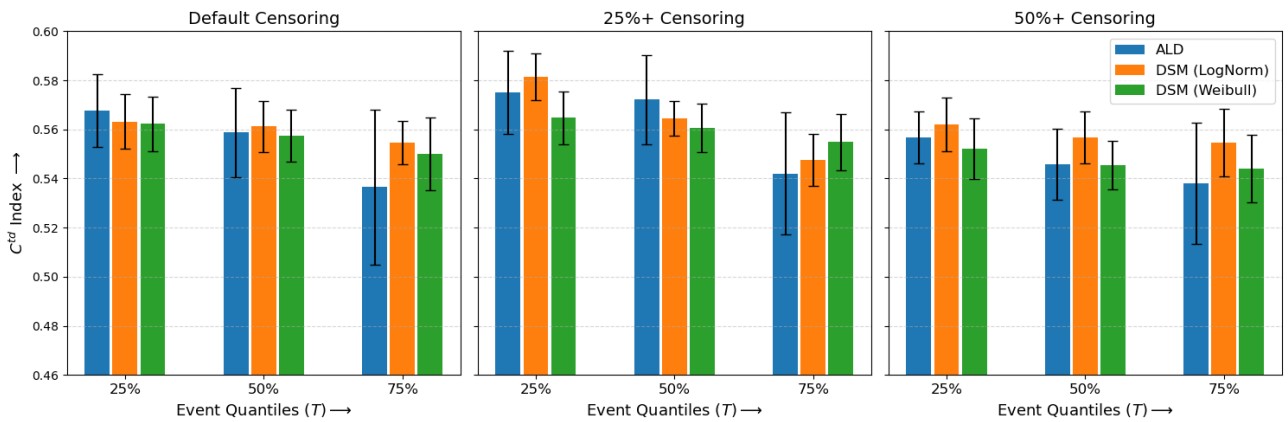

*Figure 8.* $C^{td}$ for SUPPORT dataset at different quantiles of event times for different levels of censoring.

Figures 7 and 8 illustrate $C^{td}$ values across three quantiles (25%, 50%, 75%) under default, 25%, and 50% censoring regimes. On the METABRIC dataset, our method demonstrates strong performance across all settings, particularly at higher quantiles where skewness and right-censoring are more pronounced. This suggests that our model is well-suited for capturing long-term survival signals even under substantial information loss due to censoring.

On the more challenging SUPPORT dataset, our model's performance is comparatively lower, especially under extreme censoring. This outcome can be attributed to the SUPPORT data's distributional properties, which deviate significantly from the ALD assumption and exhibit heavy skewness and short survival times. Nevertheless, our method still matches the performance of DSM-based baselines, indicating that it retains stable and competitive predictive behavior even under distributional mismatch and extreme censoring.

Overall, this analysis confirms the resilience of our method to both reduced observed events and variability in event-time horizons, with particularly strong performance on datasets aligned with ALD assumptions.

### Case Study 2: Capturing Diverse Survival Patterns

To assess the expressiveness of our model in characterizing heterogeneous survival behaviors, we perform a clustering-based analysis on both real-world and synthetic datasets.

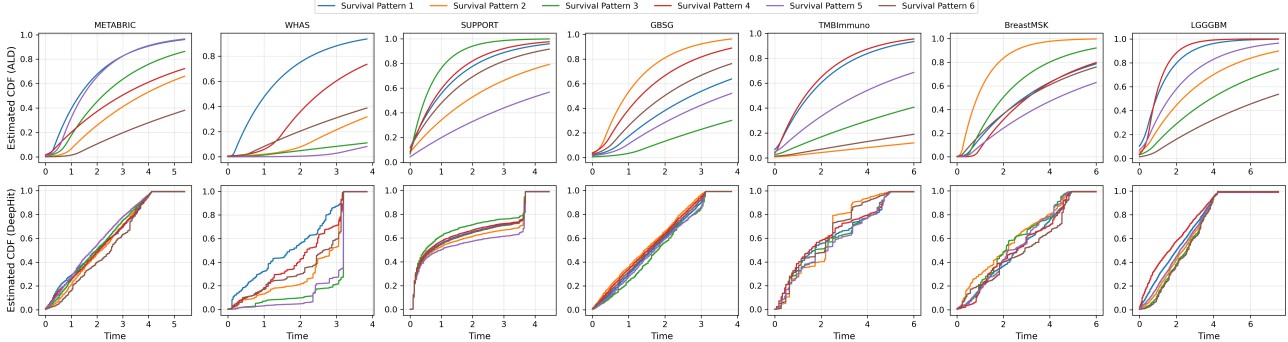

*Figure 9.* Clustered survival patterns in real datasets estimated by ALD and DeepHit.

Figure 9 visualizes six representative survival patterns identified via $K$-means clustering on the estimated parameters from our model and DeepHit across seven real-world datasets. Each curve represents the average cumulative distribution function (CDF) within a cluster. Although both models demonstrate the ability to represent distinct survival behaviors, the CDFs produced by DeepHit consistently converge to one at $1.2 \max_i y_i$, which implies that all events are predicted to occur by that time with high probability (close to 1). This behavior is unlikely in the presence of censoring and indicates a limited capacity to model long-term survival. In contrast, our model more accurately reflects plausible survival behavior under right-censoring, with CDFs that appropriately plateau before reaching one.

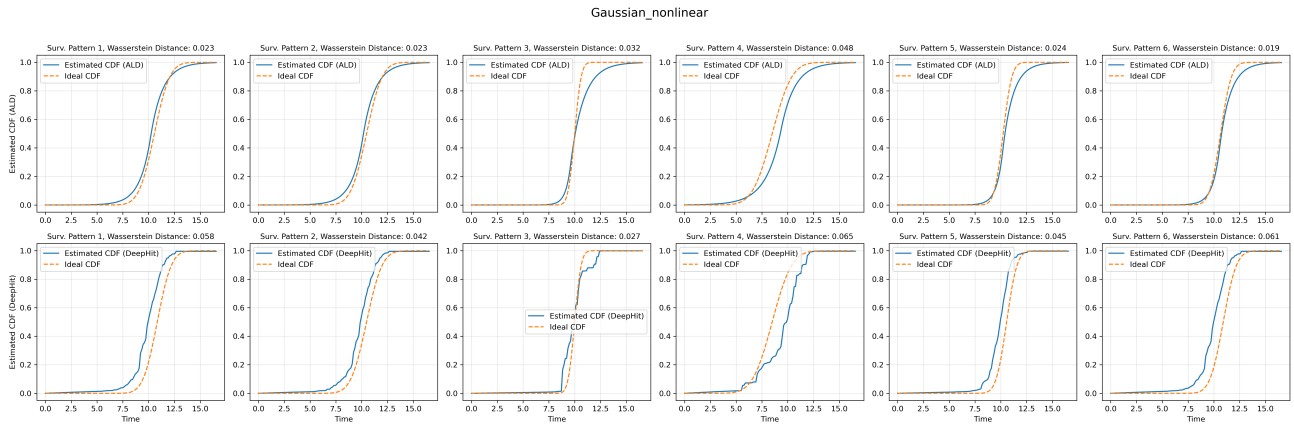

*Figure 10.* Worst-estimated instance in each cluster for Gaussian_nonlinear dataset (best synthetic dataset).

We extend this analysis to synthetic datasets where ground-truth CDFs are available, allowing a more rigorous evaluation. In Figures 10 and 11, we compare the worst-estimated instance (based on Wasserstein distance) within each of six clusters for the best-case dataset (Gaussian_nonlinear) and most challenging dataset (LogNorm_med), respectively. In both cases, our method consistently yields lower Wasserstein distances compared to DeepHit, highlighting its robustness and precision in modeling diverse survival distributions across various difficulty regimes.

These results collectively demonstrate our model's enhanced ability to recover realistic and heterogeneous survival patterns, both in real-world settings and under controlled synthetic evaluations.

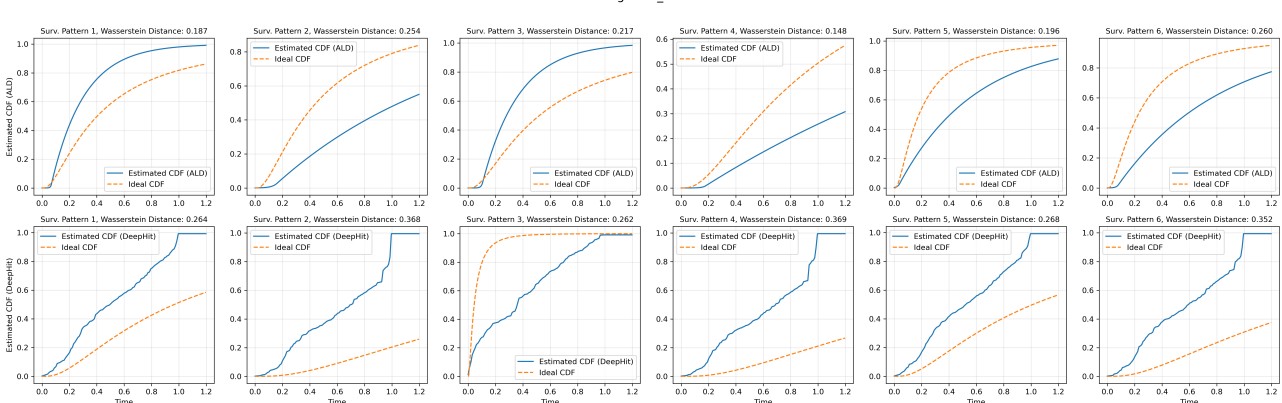

*Figure 11.* Worst-estimated instance in each cluster for LogNorm_med dataset (most challenging synthetic dataset).

## Case Study 3: Alternative Distribution Summaries

To examine the flexibility of our probabilistic model, we evaluate its predictive performance under three different distribution summaries: mean, median, and mode. While the mean serves as the default choice in our main experiments, Table 5 shows that using the median or mode can lead to improved performance on certain datasets, particularly in MAE and concordance metrics. This highlights the adaptability of our approach, where different summary statistics can be selected based on the requirements of downstream evaluation. For example, in datasets with high skewness (*e.g.*, Exponential, LogNorm_heavy), the median often outperforms the mean in terms of MAE. Conversely, the mode may yield better alignment with the underlying survival dynamics on the real datasets. This case study emphasizes the benefit of having access to a closed-form survival distribution, allowing flexible downstream use of different summaries without retraining.

*Table 5.* Full results table for all datasets, the ALD method (Mean, Median, Mode), and metrics. The values represent the mean ± 1 standard error on the test set over 10 runs.

| Dataset | Method | MAE | IBS | Harrell's C-index | Uno's C-index | CensDcal | Cal$[S(t\|\mathbf{x})]$(Slope) | Cal$[S(t\|\mathbf{x})]$(Intercept) | Cal$[f(t\|\mathbf{x})]$(Slope) | Cal$[f(t\|\mathbf{x})]$(Intercept) |
|---|---|---|---|---|---|---|---|---|---|---|
| Norm linear | ald (Mean) | 0.865 ± 1.336 | | 0.653 ± 0.014 | 0.648 ± 0.011 | | | | | |
| | ald (Median) | **0.217 ± 0.037** | 0.278 ± 0.008 | 0.654 ± 0.012 | 0.682 ± 0.037 | 0.407 ± 0.343 | 1.027 ± 0.042 | -0.016 ± 0.037 | 1.025 ± 0.016 | 0.005 ± 0.030 |
| | ald (Mode) | 0.689 ± 0.186 | | **0.657 ± 0.008** | **0.718 ± 0.006** | | | | | |
| Norm non-lin | ald (Mean) | **0.243 ± 0.080** | | **0.670 ± 0.015** | **0.644 ± 0.016** | | | | | |
| | ald (Median) | 0.253 ± 0.073 | 0.212 ± 0.006 | 0.667 ± 0.015 | 0.582 ± 0.029 | 0.406 ± 0.179 | 1.038 ± 0.025 | -0.016 ± 0.040 | 1.072 ± 0.021 | -0.011 ± 0.015 |
| | ald (Mode) | 0.438 ± 0.089 | | 0.632 ± 0.060 | 0.573 ± 0.054 | | | | | |
| Norm uniform | ald (Mean) | 0.473 ± 0.344 | | 0.785 ± 0.010 | **0.703 ± 0.020** | | | | | |
| | ald (Median) | **0.392 ± 0.196** | 0.045 ± 0.002 | 0.785 ± 0.011 | 0.696 ± 0.013 | 0.115 ± 0.030 | 1.016 ± 0.014 | -0.006 ± 0.021 | 1.019 ± 0.020 | 0.002 ± 0.016 |
| | ald (Mode) | 0.613 ± 0.118 | | **0.788 ± 0.012** | 0.696 ± 0.014 | | | | | |
| Exponential | ald (Mean) | 2.942 ± 2.389 | | **0.560 ± 0.008** | **0.560 ± 0.007** | | | | | |
| | ald (Median) | **1.088 ± 0.308** | 0.309 ± 0.018 | 0.559 ± 0.010 | 0.553 ± 0.020 | 0.432 ± 0.405 | 0.964 ± 0.049 | 0.016 ± 0.053 | 0.978 ± 0.047 | -0.015 ± 0.014 |
| | ald (Mode) | 5.009 ± 0.235 | | 0.556 ± 0.011 | 0.555 ± 0.020 | | | | | |
| Weibull | ald (Mean) | 5.134 ± 9.533 | | **0.768 ± 0.009** | **0.763 ± 0.010** | | | | | |
| | ald (Median) | **0.484 ± 0.059** | 0.219 ± 0.028 | 0.767 ± 0.006 | 0.691 ± 0.023 | 0.648 ± 0.511 | 0.993 ± 0.049 | 0.021 ± 0.060 | 1.044 ± 0.023 | -0.023 ± 0.033 |
| | ald (Mode) | 1.163 ± 0.340 | | 0.750 ± 0.008 | 0.689 ± 0.023 | | | | | |
| LogNorm | ald (Mean) | **0.363 ± 0.068** | | 0.588 ± 0.014 | **0.585 ± 0.014** | | | | | |
| | ald (Median) | 0.533 ± 0.097 | 0.376 ± 0.013 | **0.589 ± 0.015** | 0.510 ± 0.023 | 0.256 ± 0.150 | 1.011 ± 0.028 | -0.004 ± 0.029 | 1.005 ± 0.021 | 0.006 ± 0.011 |
| | ald (Mode) | 1.733 ± 0.190 | | 0.549 ± 0.043 | 0.496 ± 0.020 | | | | | |
| Norm heavy | ald (Mean) | 0.667 ± 0.139 | | **0.919 ± 0.007** | **0.870 ± 0.029** | | | | | |
| | ald (Median) | **0.454 ± 0.081** | 0.019 ± 0.001 | 0.916 ± 0.009 | 0.802 ± 0.008 | 0.256 ± 0.150 | 1.011 ± 0.028 | -0.004 ± 0.029 | 1.005 ± 0.021 | 0.006 ± 0.011 |
| | ald (Mode) | 0.627 ± 0.072 | | 0.911 ± 0.012 | 0.802 ± 0.008 | | | | | |
| Norm med. | ald (Mean) | **0.238 ± 0.036** | | **0.894 ± 0.005** | **0.872 ± 0.004** | | | | | |
| | ald (Median) | 0.298 ± 0.036 | 0.047 ± 0.003 | 0.889 ± 0.006 | 0.868 ± 0.011 | 0.157 ± 0.044 | 0.997 ± 0.012 | 0.004 ± 0.014 | 1.058 ± 0.012 | -0.036 ± 0.011 |
| | ald (Mode) | 0.388 ± 0.047 | | 0.884 ± 0.007 | 0.849 ± 0.011 | | | | | |
| Norm light | ald (Mean) | **0.236 ± 0.051** | | **0.882 ± 0.004** | **0.874 ± 0.004** | | | | | |
| | ald (Median) | 0.255 ± 0.016 | 0.090 ± 0.007 | 0.880 ± 0.003 | 0.853 ± 0.017 | 0.339 ± 0.076 | 0.998 ± 0.014 | 0.005 ± 0.021 | 1.087 ± 0.017 | -0.050 ± 0.011 |
| | ald (Mode) | 0.328 ± 0.029 | | 0.876 ± 0.003 | 0.850 ± 0.017 | | | | | |
| Norm same | ald (Mean) | 0.404 ± 0.078 | | **0.890 ± 0.005** | 0.847 ± 0.008 | | | | | |
| | ald (Median) | **0.281 ± 0.022** | 0.066 ± 0.003 | 0.888 ± 0.006 | **0.886 ± 0.004** | 0.114 ± 0.036 | 1.004 ± 0.018 | 0.010 ± 0.022 | 1.007 ± 0.014 | 0.006 ± 0.012 |
| | ald (Mode) | 0.518 ± 0.065 | | 0.881 ± 0.008 | 0.880 ± 0.004 | | | | | |
| LogNorm heavy | ald (Mean) | 0.385 ± 0.193 | | 0.777 ± 0.012 | 0.727 ± 0.022 | | | | | |
| | ald (Median) | **0.244 ± 0.042** | 0.095 ± 0.006 | **0.779 ± 0.011** | **0.749 ± 0.011** | 0.043 ± 0.019 | 0.998 ± 0.014 | -0.002 ± 0.014 | 1.003 ± 0.014 | -0.005 ± 0.005 |
| | ald (Mode) | 0.898 ± 0.045 | | 0.756 ± 0.029 | 0.724 ± 0.012 | | | | | |

| Dataset | Method | MAE | IBS | Harrell's C-index | Uno's C-index | CensDcal | Cal[$S(t\|x)$](Slope) | Cal[$S(t\|x)$](Intercept) | Cal[$f(t\|x)$](Slope) | Cal[$f(t\|x)$](Intercept) |
|---|---|---|---|---|---|---|---|---|---|---|
| LogNorm med. | ald (Mean) | **0.178 ± 0.046** | | 0.747 ± 0.004 | 0.718 ± 0.007 | | | | | |
| | ald (Median) | 0.247 ± 0.024 | 0.174 ± 0.006 | **0.748 ± 0.004** | **0.749 ± 0.013** | 0.087 ± 0.052 | 1.002 ± 0.009 | -0.001 ± 0.012 | 1.008 ± 0.017 | -0.004 ± 0.010 |
| | ald (Mode) | 0.896 ± 0.082 | | 0.723 ± 0.013 | 0.709 ± 0.012 | | | | | |
| LogNorm light | ald (Mean) | **0.184 ± 0.035** | | **0.725 ± 0.007** | **0.713 ± 0.008** | | | | | |
| | ald (Median) | 0.221 ± 0.064 | 0.310 ± 0.011 | **0.725 ± 0.007** | 0.696 ± 0.020 | 0.185 ± 0.095 | 1.001 ± 0.014 | -0.001 ± 0.016 | 0.985 ± 0.015 | 0.008 ± 0.009 |
| | ald (Mode) | 0.921 ± 0.053 | | 0.702 ± 0.014 | 0.697 ± 0.016 | | | | | |
| LogNorm same | ald (Mean) | **0.191 ± 0.044** | | 0.739 ± 0.009 | 0.697 ± 0.008 | | | | | |
| | ald (Median) | 0.259 ± 0.062 | 0.154 ± 0.006 | **0.740 ± 0.010** | **0.751 ± 0.014** | 0.076 ± 0.057 | 1.009 ± 0.011 | -0.005 ± 0.010 | 1.012 ± 0.011 | -0.001 ± 0.008 |
| | ald (Mode) | 0.943 ± 0.043 | | 0.710 ± 0.007 | 0.715 ± 0.014 | | | | | |
| METABRIC | ald (Mean) | 1.626 ± 0.194 | | 0.637 ± 0.021 | **0.633 ± 0.031** | | | | | |
| | ald (Median) | 1.123 ± 0.088 | 0.245 ± 0.012 | **0.640 ± 0.018** | 0.588 ± 0.031 | 0.293 ± 0.125 | 0.993 ± 0.028 | -0.008 ± 0.024 | 1.001 ± 0.033 | -0.012 ± 0.016 |
| | ald (Mode) | **0.856 ± 0.039** | | 0.605 ± 0.021 | 0.547 ± 0.018 | | | | | |
| WHAS | ald (Mean) | 2.196 ± 0.612 | | **0.823 ± 0.016** | **0.824 ± 0.014** | | | | | |
| | ald (Median) | 1.118 ± 0.152 | 0.134 ± 0.013 | 0.784 ± 0.043 | 0.765 ± 0.017 | 0.198 ± 0.094 | 0.981 ± 0.021 | 0.009 ± 0.023 | 0.972 ± 0.027 | 0.003 ± 0.016 |
| | ald (Mode) | **0.916 ± 0.101** | | 0.802 ± 0.018 | 0.806 ± 0.022 | | | | | |
| SUPPORT | ald (Mean) | 1.121 ± 0.107 | | 0.568 ± 0.015 | **0.572 ± 0.015** | | | | | |
| | ald (Median) | 0.856 ± 0.062 | 0.362 ± 0.013 | **0.572 ± 0.015** | 0.561 ± 0.015 | 2.197 ± 0.667 | 0.900 ± 0.056 | 0.084 ± 0.046 | 1.084 ± 0.043 | -0.113 ± 0.023 |
| | ald (Mode) | **0.421 ± 0.051** | | 0.532 ± 0.016 | 0.522 ± 0.044 | | | | | |
| GBSG | ald (Mean) | 1.713 ± 0.208 | | 0.671 ± 0.014 | **0.665 ± 0.013** | | | | | |
| | ald (Median) | 1.161 ± 0.094 | 0.278 ± 0.014 | **0.672 ± 0.010** | 0.590 ± 0.035 | 0.283 ± 0.106 | 0.977 ± 0.025 | 0.014 ± 0.034 | 1.000 ± 0.035 | -0.018 ± 0.016 |
| | ald (Mode) | **0.664 ± 0.072** | | 0.657 ± 0.023 | 0.554 ± 0.062 | | | | | |
| TMBImmuno | ald (Mean) | 3.002 ± 1.497 | | 0.561 ± 0.037 | 0.547 ± 0.040 | | | | | |
| | ald (Median) | 1.085 ± 0.191 | 0.245 ± 0.015 | **0.562 ± 0.032** | **0.548 ± 0.030** | 0.835 ± 0.604 | 0.994 ± 0.021 | 0.004 ± 0.025 | 1.053 ± 0.045 | -0.038 ± 0.025 |
| | ald (Mode) | **0.609 ± 0.069** | | 0.546 ± 0.024 | 0.531 ± 0.025 | | | | | |
| BreastMSK | ald (Mean) | 2.593 ± 0.289 | | **0.617 ± 0.032** | **0.568 ± 0.036** | | | | | |
| | ald (Median) | 1.116 ± 0.394 | 0.086 ± 0.008 | 0.457 ± 0.068 | 0.538 ± 0.083 | 0.066 ± 0.027 | 0.993 ± 0.020 | 0.003 ± 0.021 | 1.002 ± 0.019 | -0.007 ± 0.010 |
| | ald (Mode) | **0.686 ± 0.077** | | 0.591 ± 0.071 | 0.515 ± 0.090 | | | | | |
| LGGGBM | ald (Mean) | 1.232 ± 0.325 | | 0.778 ± 0.021 | 0.736 ± 0.030 | | | | | |
| | ald (Median) | 0.846 ± 0.239 | 0.108 ± 0.011 | **0.785 ± 0.030** | **0.750 ± 0.043** | 0.450 ± 0.267 | 0.996 ± 0.038 | 0.008 ± 0.040 | 0.995 ± 0.047 | 0.003 ± 0.022 |
| | ald (Mode) | **0.497 ± 0.100** | | 0.777 ± 0.023 | 0.739 ± 0.058 | | | | | |

## Case Study 4: Empirical Behavior of $F_{\mathbf{ALD}}(0|\mathbf{x})$

Given that the Asymmetric Laplace Distribution (ALD) has non-zero support over the negative time domain, we investigate whether the model assigns substantial probability mass to implausible event times $t < 0$. Specifically, we analyze the empirical distribution of $F_{\mathrm{ALD}}(0 \mid \mathbf{x})$—the predicted probability that an event occurs before time zero.

*Table 6.* The 50th, 75th and 95th percentiles of the CDF estimation for $t = 0$, $F_{\mathrm{ALD}}(0 \mid \mathbf{x})$, under the Asymmetric Laplace Distribution.

| Dataset | 50th Percentile | 75th Percentile | 95th Percentile |
|---|---|---|---|
| Norm linear | 0.0001 | 0.0007 | 0.0018 |
| Norm non-linear | 1.9878e-06 | 0.0001 | 0.0007 |
| Norm uniform | 2.9879e-05 | 0.0028 | 0.0124 |
| Exponential | 0.0194 | 0.0665 | 0.1204 |
| Weibull | 0.0015 | 0.0032 | 0.0046 |
| LogNorm | 0.0031 | 0.0109 | 0.0134 |
| Norm heavy | 1.1804e-06 | 2.6128e-05 | 0.0007 |
| Norm med | 4.2222e-06 | 3.5778e-05 | 0.0004 |
| Norm light | 1.1978e-05 | 0.0001 | 0.0009 |
| Norm same | 7.8051e-07 | 4.8624e-06 | 0.0001 |
| LogNorm heavy | 0.0001 | 0.0014 | 0.0142 |
| LogNorm med | 0.0001 | 0.0007 | 0.0082 |
| LogNorm light | 0.0004 | 0.0024 | 0.0150 |
| LogNorm same | 0.0004 | 0.0021 | 0.0123 |
| METABRIC | 0.0068 | 0.0123 | 0.0292 |
| WHAS | 0.0046 | 0.0151 | 0.0507 |
| SUPPORT | 0.0957 | 0.1393 | 0.2035 |
| GBSG | 0.0248 | 0.0394 | 0.0668 |
| TMBImmuno | 0.0523 | 0.0681 | 0.0878 |
| BreastMSK | 0.0006 | 0.0008 | 0.0130 |
| LGGGBM | 0.0570 | 0.0842 | 0.1356 |

Table 6 reports the 50th, 75th, and 95th percentiles of $F_{\mathrm{ALD}}(0 \mid \mathbf{x})$ across 21 datasets. The results indicate that this quantity remains close to zero in the vast majority of cases. Even at the 95th percentile, most values remain below 1–2%, suggesting that the model rarely assigns significant mass to invalid time regions. This empirically confirms that, despite the ALD's unbounded support, the learned parameters stay consistent with the temporal constraints of survival analysis.

## Case Study 5: Limitations of ALD on SUPPORT

The model's performance on the SUPPORT dataset is comparatively lower than on other benchmarks. This shortcoming

can largely be attributed to the assumption of the Asymmetric Laplace Distribution (ALD), which may not be universally suitable across datasets. In particular, the SUPPORT dataset exhibits pronounced skewness and a narrow range of event times, as visualized in Figure 12. A substantial concentration of events occurs near zero, resulting in extreme quantile values and rendering the distribution challenging to calibrate, especially in the early time horizon (*i.e.*, $t \rightarrow 0$).

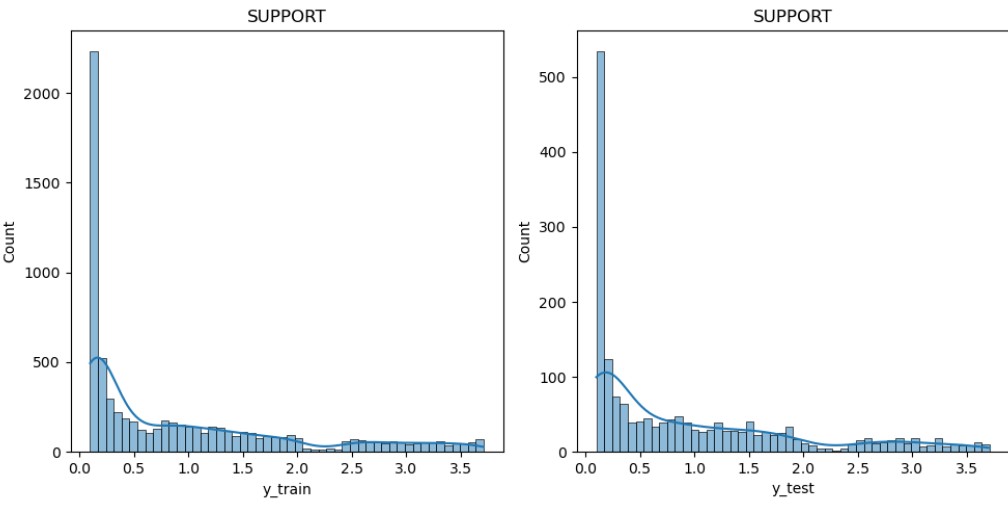

*Figure 12.* Distribution of the SUPPORT dataset for the training set and test set.

In response to this skewness, the model assigns high probability mass to early intervals while allocating limited capacity to later time points. This imbalance compromises calibration accuracy across the full temporal range. Nonetheless, the resulting calibration metrics remain within acceptable bounds and are generally comparable to those of the baselines. Furthermore, as shown in Appendix C.1 and Table 2, our model still achieves competitive performance on slope and intercept-based calibration metrics, reaffirming its robustness despite the dataset-specific limitations.

