# OpenReview forum: "Learning Survival Distributions with the Asymmetric Laplace Distribution"
_ICML.cc/2025/Conference — ICML 2025 poster_

### Official Review · Reviewer_6hhK · 2025-03-05

**Overall Recommendation:** 4

**Summary:**

The paper introduces a novel parametric survival analysis framework based on the Asymmetric Laplace Distribution (ALD). By formulating the survival problem through ALD, the authors derive closed-form expressions for key distributional summaries (e.g., mean, median, mode, and quantiles) and propose an architecture where a shared encoder feeds into three independent heads to predict the ALD parameters. The model is trained via maximum likelihood estimation (MLE) with tailored loss functions for both observed and censored data. Comprehensive experiments on 14 synthetic and 7 real-world datasets demonstrate that the proposed approach often outperforms established methods such as LogNorm MLE, DeepSurv, DeepHit, and CQRNN in terms of predictive accuracy, discrimination (using C-index metrics), and calibration.

**Updates after Rebuttal**

I thank the authors for their rebuttal. Although I am not sure if including additional results in anonymous githubs is legitimate, the revised experiments look much more promising and well-address my concern on empirical evaluation. I will increase my score by one point.

**Claims And Evidence:**

* While the paper asserts that the ALD-based method is broadly effective, its reliance on the ALD assumption is problematic in cases with highly skewed event distributions (e.g., the SUPPORT dataset). The evidence does not fully demonstrate robustness when the underlying survival distribution deviates significantly from an ALD. This point requires more convincing ablation studies or experiments.
* The paper makes claims regarding the efficiency and stability of the proposed model, bolstered by the use of a shared encoder and residual connections. Still, the experiments do not sufficiently address how the approach scales to very high-dimensional data or extremely censored datasets --- especially some heavily censored datasets.

**Essential References Not Discussed:**

* Deep Survival Machine (DSM) —which models survival times using a learned mixture of parametric distributions—shares the goal of providing a full survival distribution without relying on the proportional hazards assumption. However, while DSM flexibly combines several parametric components (such as Weibull or log-normal distributions) to capture heterogeneous survival patterns, the proposed ALD-based method offers a closed-form formulation for key event summaries (mean, median, quantiles) via maximum likelihood estimation. This paper was not discussed and compared.
* Similarly the infamous DeepCox model in this field was also not compared.

**Experimental Designs Or Analyses:**

I have checked all experiment designs and analyses. Essentially I think there are several baselines missing, such as DeepCox, Deep Survival Machine, Neural Frality Machine, and Deep Cox Mixture.

**Methods And Evaluation Criteria:**

Yes

**Other Comments Or Suggestions:**

N.A.

**Other Strengths And Weaknesses:**

Strengths:

* The paper offers detailed derivations, theoretical insights, and even proofs (presented in the appendix) that explain the loss formulation and the properties of the ALD. This depth of analysis underpins the methodological contributions.
* The use of the ALD provides a neat way to capture survival distributions in closed form. This results in efficient computation of various summary statistics, which enhances interpretability compared to discretized methods.

Weaknesses:

* The model’s performance is contingent upon the assumption that survival times are well modeled by an ALD. In cases such as the SUPPORT dataset, where the event distribution is highly skewed, this assumption may not hold, potentially affecting calibration.
* Experiments on highly censoring scenarios (e.g., refer to the experiments by DSM) are encouraged to more comprehensively evaluate the model at tails.

**Questions For Authors:**

* How sensitive is the proposed ALD-based method when the true underlying survival distribution deviates from the ALD assumption? Can you provide results from ablation studies or additional experiments that test this robustness?
* In datasets with very high censoring rates (e.g., the experiment settings of DSM), what modifications or additional regularizations might be necessary to maintain reliable calibration and discrimination? Have you considered how the method might be adapted to such scenarios?

**Relation To Broader Scientific Literature:**

The paper’s contributions are deeply rooted in—and extend—the established literature on survival analysis and quantile regression. In particular, it builds on classical parametric models (such as exponential, Weibull, and log‐normal models) by introducing a flexible, ALD‐based approach that provides closed-form expressions for key survival summaries (mean, median, quantiles). This idea is an evolution of earlier work in quantile regression where the asymmetric Laplace distribution was used as a working likelihood (see Koenker and Bassett, 1978; Yu and Moyeed, 2001) to capture the conditional quantiles of the response.

Furthermore, by integrating a neural network architecture with a shared encoder and independent heads for estimating the ALD parameters, the paper aligns with recent deep learning efforts in survival analysis (e.g., DeepSurv, DSM, Deep Cox, Neural Frality Machine, DeepHit, and CQRNN) that aim to model complex, non-linear relationships and to address limitations of traditional models like the Cox proportional hazards model.

**Theoretical Claims:**

Yes I chwecked all theoretical proofs and claims.

---

> ### Author Rebuttal · Authors · 2025-03-31
>
> We sincerely thank you for your comprehensive evaluation and insightful comments.
> We also appreciate the detailed review of our theory, experiments, and connections to the broader literature.
> Below we respond to the specific points raised:
>
> **R1[Claims And Evidence, Other Strengths And Weaknesses (Weaknesses), Questions For Authors 1]:**
> We would like to emphasize that our benchmark covers a diverse range of survival scenarios with varying censoring and skewness levels. All 21 datasets used in our experiments differ in their censoring rates, and for synthetic datasets (*e.g.*, multivariate Normal, Log-Normal), we explicitly vary the censoring levels to create more challenging conditions. Additionally, distributions like LogNormal and Exponential inherently introduce strong skewness. Importantly, these synthetic datasets are distributionally distinct from ALD, yet our method performs well across them, demonstrating robustness beyond the assumed distribution.
> The latest Table 2 [https://anonymous.4open.science/r/ICML25/Fig1.png ] (4 more baselines added) highlights the superiority of our method compared to a broad spectrum of baselines, including (semi-)parametric and nonparametric models, covering both neural and non-neural architectures.
> Although our method shows relatively weaker results on the SUPPORT dataset, the same trend is seen in several baselines. Nevertheless, our performance remains competitive and stable across metrics.
>
> Following your suggestions, we added case studies inspired by DSM to evaluate robustness under high censoring and at key quantiles (25%, 50%, 75%). Fig 2 and 3 [https://anonymous.4open.science/r/ICML25/Fig2.png; https://anonymous.4open.science/r/ICML25/Fig3.png ] present results on METABRIC and SUPPORT datasets, which differ significantly in their censoring and distributional properties.
> Our method consistently achieves strong or competitive time-dependent concordance scores across censoring levels.
> On METABRIC, it performs especially well at higher quantiles, where skewness and censoring are more pronounced.
> Although our method shows relatively weaker results on SUPPORT, this is likely due to the substantial deviation of its underlying distribution from the ALD assumption. However, our method still performs comparably to DSM-based baselines without significant degradation.
>
> We also appreciate the point and agree that scaling to higher-dimensional datasets is important.
> While our current experiments focus on structured survival datasets with up to 14 covariates (*i.e.*, SUPPORT), our method is lightweight, requires only 3 estimated parameters of ALD per instance, and leverages compact neural architectures.
> It does not rely on large time grids or mixture components, making it computationally efficient.
> Intuitively, we expect our method to retain its advantages even in higher-dimensional settings due to its compact parameterization and stable training behavior.
> We are currently exploring extensions to such settings and plan to evaluate their scalability and performance on high-dimensional survival data in future work.
>
> **R2[Experimental Designs Or Analyses, Essential References Not Discussed]:**
> In our current version, we have already reported results for DeepCox (referred to as DeepSurv in our initial submission), and have additionally incorporated new baselines such as DSM with Log-Normal and Weibull components, as well as classical non-neural models like GBM and RSF.
> For other models such as Neural Frailty Machine and Deep Cox Mixture, we will include a detailed discussion in the Related Work section (similar to the additional comparisons presented in our response to **Reviewer rnhn**) and are happy to incorporate further empirical evaluations in the revision if needed.
>
> Here, we would like to highlight that, compared to these mixture-based models, our approach offers two key advantages:
>
> 1. Computational efficiency. DSM models the survival function as a mixture of $K$ Log-Normal or Weibull components, which increases computational complexity as $K$ grows. In contrast, our model requires estimating only 3 parameters per instance, enabling faster training and inference with a much simpler architecture.
>
> 2. Performance stability. DSM is sensitive to $K$ and architecture choices. Misaligned components can lead to instability. In contrast, our model relies on straightforward neural network hyperparameters and exhibits stable performance across diverse datasets without extensive tuning.
>
> **R3[Questions For Authors 2]:**
> While we acknowledge that more sophisticated loss functions or architectural enhancements could potentially further improve performance, our current focus is on maintaining a simple and interpretable design.
> Despite its simplicity, our approach already achieves strong empirical results.
> We highly agree that integrating more advanced components is an interesting and valuable direction and plan to explore this in future work.

---

### Official Review · Reviewer_QAQg · 2025-03-10

**Overall Recommendation:** 3

**Summary:**

The paper proposes a parametric survival analysis method based on asymmetric Laplace distribution. It enables predicting continuous distribution-based predictions, unlike existing discretized nonparametric methods.

**Claims And Evidence:**

The paper mentions several limitations of the existing methods 40-54, such as strong assumptions about the underlying distribution, only predicting discretized distribution.

The experiments show the proposed method is significantly better than the selected baselines. However, there seems to be not many discussion about how important each module is. Specifically,
1. the paper claims predicting continuous distribution is an advantage. I acknowledge this and it is achieved obviously through the theoretical foundation. However, how many benefits are brought by this?
2. The paper claims the strong assumptions about the underlying distribution. Why does the proposed method release the assumption and what is some evidence to show that?

**Essential References Not Discussed:**

No

**Experimental Designs Or Analyses:**

The experiment statistics looks good, but as mentioned in previous sections, it does not sufficiently support the claims made by the paper. If possible, I would recommend adding more experiments:
1. baselines of predicting the coefficients of a Fourier transform or Taylor series.
2. compare what survival patterns can the proposed method succeeds and when it fails.

**Methods And Evaluation Criteria:**

The proposed method predicts the parameters of the asymmetric Laplace distribution, and thereby predicts the survival distribution. The corollary 3.2 provides the justification of using asymmetric Laplace distribution, that tells the relationship between it and quantile regression. This method makes sense and works as expected, but the contribution is limited, mainly because of these:
1. There are many ways to model a continuous parametrized distribution other than Asymmetric Laplace Distribution. For example, predicting the coefficients of a Fourier transform, or Taylor series. They are also continuous in nature. What are the unique advantages of ALD?
2. The paper claims the proposed method offers superior flexibility in capturing diverse survival patterns. But there seems no experiments supporting this claim.

**Other Comments Or Suggestions:**

No

**Other Strengths And Weaknesses:**

No

**Questions For Authors:**

No

**Relation To Broader Scientific Literature:**

I am unsure about this question.

**Theoretical Claims:**

The theoretical claims make sense to me. But the detailed math is not carefully checked.

---

> ### Author Rebuttal · Authors · 2025-03-31
>
> We sincerely thank you for your careful reading and constructive feedback.
> We added four more baselines shown in Table 2 [https://anonymous.4open.science/r/ICML25/Fig1.png ] and more comparisons (See the reply for **Reviewer rnhn**) will be added.
> Below, we will address your concerns.
>
> **R1[Claims And Evidence 1]:**
> Unlike discretized nonparametric approaches, our method estimates only 3 parameters of the ALD per instance, providing computational efficiency and smooth, closed-form CDFs.
> In contrast, models such as DeepHit must balance computational cost and accuracy by discretizing time, which introduces approximation errors, especially, under coarse time grids, and limits their ability to capture fine-grained survival dynamics.
>
> Beyond theory, across 14 synthetic and 7 real-world datasets, our method outperforms DeepHit in 73% of 189 benchmark settings and underperforms in less than 4% (Table 2), underscoring the strength of continuous modeling.
> This advantage is most evident in the IBS metric, where our method achieves superior performance across all 21 datasets.
> Notably, IBS evaluates both discrimination and calibration over time, making it especially sensitive to distributional smoothness and continuity.
>
> **R2[Claims And Evidence 2]:**
> Indeed, any parametric model makes certain assumptions about the underlying distribution.
> We adopt the ALD assumption primarily due to its strong theoretical and practical connections to quantile regression.
> The widely used pinball (checkmark) loss, foundational in methods like CQRNN, is essentially derived from ALD. However, such methods lack full distributional modeling and continuity.
>
> Our work bridges this gap, retaining the empirical strengths of quantile-based approaches while introducing a fully parametric, continuous, and interpretable survival model. As noted by **Reviewer 6hhK**, this represents a principled evolution from loss-based to distribution-based modeling.
> Other parametric options (*e.g.*, Gamma, Log-Normal) are possible but often suffer from the absence of closed-form CDFs, unstable gradients, and reduced robustness (see the reply for **Reviewer rnhn** and Appendix A.2). In contrast, ALD offers stable training and strong empirical results, making it a compelling choice.
>
> **R3[Methods And Evaluation Criteria 1]**:
> To the best of our knowledge, there is limited prior work exploring the use of Fourier or Taylor series expansions for continuous survival distribution modeling, making this a novel and interesting direction.
> We would greatly appreciate any specific references or formulations the reviewer has in mind for further consideration, and we are open to exploring this line of research in the revision.
>
> While many alternatives exist, our goal is to balance simplicity, interpretability, and performance. The ALD-based framework achieves this effectively, as demonstrated in our results.
>
> **R4[Methods And Evaluation Criteria 2, Experimental Designs Or Analyses]:**
> We now provide new analyses to support this claim.
> Fig 4 [https://anonymous.4open.science/r/ICML25/Fig4.png ] shows the flexibility of our method and DeepHit in capturing diverse survival patterns in the 7 real-world datasets.
> Specifically, we applied K-means clustering to the estimated parameters from our model, with the number of clusters set to 6.
> This allows us to visualize 6 representative survival patterns, where each CDF curve corresponds to the average parameters within a cluster.
> As shown in Fig 4, both our method and DeepHit are capable of capturing diverse survival patterns in real-world datasets.
> However, the patterns generated by DeepHit often appear less realistic.
> Its CDFs consistently converge to 1 across all identified patterns, suggesting that every individual eventually experiences the event.
> This outcome contradicts the existence of censoring in the data, where a non-zero censoring rate implies that a portion of individuals should remain event-free over time.
> Although the true underlying distribution is not observable in real-world settings, this behavior illustrates DeepHit’s limited ability to model long-term survival.
>
> To further compare what survival patterns can the proposed method succeed in and when it fails, we leverage the fact that ground-truth CDFs are available for each instance in the synthetic datasets.
> This allows us to perform the same clustering analysis as in the real-world setting.
> Specifically, Fig 5 [https://anonymous.4open.science/r/ICML25/Fig5.png ] presents the worst-estimated instance (based on Wasserstein distance) within each of the 6 clusters for the best-performing synthetic dataset, comparing our method and DeepHit.
> Fig 6 [https://anonymous.4open.science/r/ICML25/Fig6.png ] repeats this for the most challenging dataset.
> These results demonstrate that our method consistently yields lower Wasserstein distances across different survival patterns, indicating a more reliable estimation of survival distributions, even in difficult scenarios.

---

### Official Review · Reviewer_rnhn · 2025-03-12

**Overall Recommendation:** 3

**Summary:**

This paper proposes a parametric survival analysis model which uses asymmetric Laplace distributions (ALDs) to represent survival distributions, where the non-linear dependence of ALD’s parameters on static covariates is modeled by neural networks. The experiments on synthetic and real-world data confirmed the model's validity.

**Claims And Evidence:**

The primal contribution of this paper is to introduce asymmetric Laplace distributions (ALDs) to represent survival distributions. But it seems not to be clear why ALDs should be adopted or why ALDs are superior to other parametric/nonparametric approaches. For example, do ALDs have any advantages over mixture models (e.g., mixture of log-normal or gamma distributions) in terms of accuracy or computational cost?

**Essential References Not Discussed:**

To overcome the limitation of Cox proportional hazards model, a vast number of non-neural network survival models have been proposed that replace the log-linear parametric function with a non-linear one. Especially, generalized boosted models [a], random survival forests [b], and Gaussian process models [c,d,e], which are not cited in the paper, should be discussed in Related Work. Some of them should be included in the comparative experiments (e.g., generalized boosted models can be easily implemented via gbm(distribution = ’coxph’) in R).

[a] Ridgeway G, Developers G (2025). gbm: Generalized Boosted Regression Models. R package version 2.2.2, https://github.com/gbm-developers/gbm.

[b] Hemant Ishwaran, Udaya B. Kogalur, Eugene H. Blackstone, and Michael S. Lauer. Random survival forests. The Annals of Applied Statistics, 2(3):841–860, 2008.

[c] Tamara Fernández, Nicolás Rivera, and Yee Whye Teh. Gaussian processes for survival analysis. In Advances in Neural Information Processing Systems 29, 2016.

[d] Minyoung Kim and Vladimir Pavlovic. Variational inference for Gaussian process models for survival analysis. In Uncertainty in Artificial Intelligence, pages 435–445, 2018.

[e] Hideaki Kim. Survival Permanental Processes for Survival Analysis with Time-Varying Covariates. In Advances in Neural Information Processing Systems 36, 2023.

**Experimental Designs Or Analyses:**

The compared reference models seem to focus heavily on neural network-based models. More standard nonparametric approaches in survival analysis (e.g., random survival forest) should be included in the comparison. See below about essential references.

**Methods And Evaluation Criteria:**

The proposed methods and evaluation criteria make sense.

**Other Comments Or Suggestions:**

No other comments.

**Other Strengths And Weaknesses:**

Strength
- The validity of the proposed model was evaluated on many real-world data.

Weakness
- See above.

**Questions For Authors:**

I cannot follow the sentence on pp. 4, "However, semiparametric models face challenges in effectively handling censored data, particularly when censoring rates are very high." Could you explain why semiparametric approaches work poorly in high-rate censoring?

**Relation To Broader Scientific Literature:**

This paper provides an accurate survival analysis, which could be beneficial especially in clinical fields.

**Theoretical Claims:**

Theoretical claims seem to be correct.

---

> ### Author Rebuttal · Authors · 2025-03-31
>
> Thank you for your thoughtful and constructive feedback, and for generously suggesting many references that we had not previously considered.
> We added 4 more baselines and the results are available in Table 2 [https://anonymous.4open.science/r/ICML25/Fig1.png ].
> Below, we provide a detailed discussion comparing our proposed method with the baselines you mentioned and address your specific concerns.
>
> **R1[Claims And Evidence]:**
>
> **Compared to parametric methods, our approach offers two key advantages:**
>
> 1. For our method, both the PDF and CDF of the ALD are available in closed form and are smooth to their parameters.
> While it is possible to use any distribution to model survival functions, some commonly used alternatives, such as the Gamma distribution, often suffer from gradient instability during training.
> Specifically, the Gamma distribution lacks a closed-form expression for its CDF, which instead relies on numerically approximated special functions (*i.e.*, the incomplete Gamma function).
> These operations introduce significant numerical instability when computing gradients via backpropagation, leading to unstable training dynamics or poor convergence.
> In contrast, the ALD offers closed-form expressions with stable gradients, making it well suited for optimization in neural network-based survival models.
>
> 2. Experimental results on 14 synthetic and 7 real-world datasets across 9 metrics show that our method consistently achieves strong performance, even under high skewness and heavy censoring.
> As shown in Table 2, it outperforms LogNormal in 60\% of 189 settings, with worse performance in less than 5\%, highlighting its robustness and generalizability.
>
> **Compared to nonparametric methods, our approach offers two key advantages:**
>
> 1. Our method provides continuous and closed-form estimates of CDFs and summary statistics, avoiding the discretization inherent in many nonparametric models.
> Notably, these discretization CDFs can introduce significant approximation error, especially when the interval between time steps is large, which limits their precision in capturing fine-grained temporal dynamics.
>
> 2. The parametric nature of the ALD enables faster training and inference, with fewer parameters and lower memory overhead compared to nonparametric methods.
> For example, our model only needs to estimate 3 parameters per instance.
> In contrast, nonparametric approaches, such as DeepHit, require computing and storing hundreds of discrete CDF values to achieve comparable resolution, resulting in significantly higher computational costs.
>
> **Compared to the mixture models, our approach offers two key advantages:**
>
> 1. DSM models the survival function as a mixture of $K$ Log-Normal or Weibull components, leading to increased computational complexity as $K$ grows.
> In contrast, our ALD-based model is simple and estimates only 3 parameters per instance, enabling faster training and inference.
>
> 2. In practice, DSM is highly sensitive to the choice of parameter $K$ and architectural design, partly due to its loss function, which aggregates multiple distributional components.
> This can make it vulnerable to instability when some mixture components fail to capture the underlying survival patterns effectively.
> In contrast, our model relies on simple neural network hyperparameters and delivers more stable performance across diverse settings, without the need for extensive tuning.
>
> **R2[Experimental Designs Or Analyses, Essential References Not Discussed]:**
> We will include and discuss these important works in the related work section.
> We have implemented both GBM and RSF, as reported in Table 2.
>
> **Compared to these two standard nonparametric approaches:**
>
> 1. Our method outperforms GBM in 52\% of the 189 benchmark settings while underperforming in fewer than 7\% of them. Similarly, it surpasses RSF in 55\% of the settings and performs worse in fewer than 9\%, demonstrating strong robustness and generalizability across diverse survival scenarios.
>
> 2. Ensemble-based nonparametric methods such as GBM and RSF typically require training 100 or more decision trees, which can be computationally intensive and memory demanding, especially for large datasets or when tuning hyperparameters.
> Moreover, once trained, our method enables fast evaluation of continuous survival functions in closed form, whereas tree-based models require traversing ensembles for each prediction.
>
> **R3[Questions For Authors]:**
> Semiparametric models like the Cox proportional hazards model incorporate censored samples only indirectly through the risk set in the partial likelihood.
> As a result, when censoring is high, the model relies on very limited observed events, reducing its statistical efficiency.
> In contrast, our parametric approach models both censored and uncensored data explicitly through a full likelihood, allowing it to make better use of all available information and remain robust under high censoring.

---

> > ### Comment · Reviewer_rnhn · 2025-04-03
> >
> > Thank the authors for the clarification and for including additional reference methods in the experiments. While I remain somewhat unconvinced about the superiority of ALDs over other parametric distributions or mixture models, the results appear remarkably good, and I believe the proposed method offers practical value. I appreciate the authors’ efforts to better articulate the paper's contributions. I will raise my score to 3: weak accept, **on the condition that** the appendix includes a detailed description of how the hyper-parameters for all comparison methods were determined (e.g., the procedure for selecting the number of components in DSM, the number of trees in ensemble methods, and the number of time discretization points used for numerical integration in DeepSurv and DeepHit).

---

> > > ### Author Response · Authors · 2025-04-03
> > >
> > > We sincerely thank you for your positive feedback and for raising the score.
> > >
> > > We are glad to hear that the proposed methods are considered to have practical value, and we also appreciate your recognition of our efforts in clarifying our contribution and enriching our experimental comparison.
> > >
> > > As per your suggestion, we will detailedly describe the hyperparameter selection process of these methods and all the compared methods in our Appendix. This will cover but is not limited to: 1. the procedure for selecting the number of components in DSM. 2. the number of trees in ensemble methods. 3. the number of time discretization points used for numerical integration in DeepSurv and DeepHit. We believe that this supplement will further improve the transparency and reproducibility of our experiments. Additionally, we will update our code to include the added baselines and the case study examples.
> > >
> > > Thank you again for your thoughtful review of both our paper and rebuttal. We truly appreciate the time and effort you’ve put into evaluating our work, as well as the insightful references you’ve suggested. Your comments have consistently been constructive and valuable, and they’ve played a key role in helping us improve the clarity and quality of our paper.

---

### Official Review · Reviewer_jgfN · 2025-03-14

**Overall Recommendation:** 3

**Summary:**

- Authors introduce a parametric survival analysis model, which utilises the Asymmetric Laplace Distribution in the Quantile Regression inspired loss function.

- Inspired by the mean absolute deviation loss function, which models the hyperplane with median distance from all points, the quantile regression is a modelling approach where we use weighted absolute value as a loss function to model the hyperplane which separates p and 1-p observations in the optimal way. To estimate the parameters of the Asymmetric Laplace normal, authors present their architecture based on the shared encoder, residual skip connection  and 3 heads that predict the parameters of the Asymmetric Laplace Distribution.

- The findings are supported by the experiments on the real data.

**Claims And Evidence:**

Claims are well supported. Evidence is provided in the form of:
-  the derivations,
-  experiments,
- relevant citations.

**Essential References Not Discussed:**

- All relevant text books coverring Quantile regression are cited.
- The benchmark models are well described and discussedI .

**Experimental Designs Or Analyses:**

Authors compare the performance against other 4 benchmark models: CQRNN, DeepSurv, DeepHit and LogNorn

**Methods And Evaluation Criteria:**

Yes.

 - Authors carry experiments on the synthetic datasets and 7 real-life dataset related to survival statistics task to support their claim.

 - Authors vary the number of the features, sizes of the datasets and various  censored fractions of the datates.

 - Each experiment is repetead 10 times.

**Other Comments Or Suggestions:**

Not convinced that the results itself is strong enough as a topic to publish the paper (however methodologically is the paper well composed and written). Happy to be convinced though.

**Other Strengths And Weaknesses:**

Well written paper, easy to follow, methodology is great.

**Questions For Authors:**

1. Looking at the size of the architecture and datasets: when it comes to selection of the architecture, how can you achieve that given the sample and architecture size, the networks are generalising reasonably well rather than memorising.
2. Did you experiment with different architectures?
3. Did you consider experiments containing shifts of the test sets to verify how well the selected method generalises to domain shift?
4. How does the model behave for the edge cases, i.e. is your methodology suitable for the extreme quantiles?

**Relation To Broader Scientific Literature:**

The model design seems to be novel.

**Theoretical Claims:**

Verified Corollary 3.2 in Appendix A.1. Seems correct.

---

> ### Author Rebuttal · Authors · 2025-03-31
>
> Thank you for your constructive and insightful feedback.
> We appreciate the time and effort spent evaluating our work and will address the concerns and questions below.
>
> **R1[Other Comments Or Suggestions]:**
> In response, we would like to emphasize that our method introduces a novel but simple loss formulation based on the ALD for survival modeling.
> Compared to non-parametric methods, our approach offers great flexibility and interpretability, as it enables closed-form expressions for key survival summaries (*e.g.*, mean, median and quantiles) and continuous cumulative distribution functions.
> In contrast to existing (semi-)parametric methods, our approach is easy to train and offers balanced performance metrics, making it well-suited for modeling diverse survival patterns across datasets.
>
> For the experiments, we conducted extensive evaluations in 14 synthetic and 7 real-world datasets to comprehensively assess our method under diverse survival scenarios.
> We compare our approach against 8 competitive baselines (4 more added), covering a broad spectrum of survival models — (semi-)parametric and non-parametric —as well as both neural and non-neural architectures.
> Our evaluation spans 9 performance metrics, capturing the model's accuracy, discrimination, and calibration.
>
> As a result, Table 2 summarizes the superiority of our method, and the Appendix further analyzes the details of the other baselines for a better understanding of the difference between our methods from both theoretical and empirical aspects.
>
> Thanks to the insightful feedback from all reviewers, we have included additional results and case studies to highlight the following:
>
> 1. The superiority of our method compared to mixture-based models (*e.g.*, DSM with Log-Normal and Weibull components) and non-neural survival models (*e.g.*, Generalized Boosted Models and Random Survival Forests) [https://anonymous.4open.science/r/ICML25/Fig1.png ];
>
> 2. The flexibility of our method in capturing diverse survival patterns in real-world datasets [https://anonymous.4open.science/r/ICML25/Fig4.png ] and synthetic datasets [https://anonymous.4open.science/r/ICML25/Fig5.png; https://anonymous.4open.science/r/ICML25/Fig6.png ], complementing our synthetic benchmarks that already span a wide range of structural variations distinct from ALD;
>
> 3. The robustness of our approach under challenging real-world conditions [https://anonymous.4open.science/r/ICML25/Fig2.png; https://anonymous.4open.science/r/ICML25/Fig3.png ], such as pronounced skewness and heavy censoring, provides further evidence of its reliability beyond the varied censoring and skewness levels considered in our initial experiments.
>
> While time constraints limited the extent of results we could include in this rebuttal, we would be more than happy to conduct and report additional experiments or case studies if the reviewers find them helpful.
>
> **R2[Questions For Authors 1-2]:**
> We did not explore alternative architectures in our work for the following reasons:
>
> 1. Dataset simplicity. Most of our datasets are low-dimensional (the highest being SUPPORT with 14 features).
> Given this, we focused on fully connected architectures, which offered sufficient capacity while avoiding overfitting. Empirically, we found them to generalize well across datasets with varying feature sizes and censoring levels.
>
> 2. Comparability with prior work. Most recent neural methods for survival analysis, such as CQRNN, DeepSurv, DeepHit and DeepSurv, also use fully connected architectures (see Appendix B.3).
> To ensure a fair comparison, we adopted the same architectural choice.
>
> While an exhaustive architecture search was beyond the scope of this work, we fully agree that this is a valuable direction.
> We would be happy to explore alternative configurations in future work and greatly appreciate any concrete suggestions regarding alternative architectures.
>
> **R3[Questions For Authors 3]:**
> The point about domain shift is excellent and domain shift is indeed an important and valuable direction for future research.
> While such an analysis falls outside the scope of our current work, we fully acknowledge its significance.
> In future extensions, we plan to incorporate experiments under domain shift scenarios.
> We will also add a discussion paragraph in the final version to explicitly reflect this limitation and outline potential directions for addressing it.
>
> **R4[Questions For Authors 4]:**
> Thank you for this valuable question.
> Your concern aligns closely with the feedback from **Reviewer 6hhK**, and we have further conducted additional case studies following the evaluation settings used in DSM. As illustrated in Fig2 and 3 [https://anonymous.4open.science/r/ICML25/Fig2.png; https://anonymous.4open.science/r/ICML25/Fig3.png ], our method consistently demonstrates superior or competitive performance across different censoring regimes and event-time quantiles. The full analysis can be found in the reply for **Reviewer 6hhK**.

---

### Decision · Program_Chairs · 2025-05-01

**Decision:**

Accept (poster)

**Comment:**

The reviewers overall leaned toward acceptance after the discussion period, where the authors provided detailed comments including running additional experiments. Overall, the proposed ALD method is a valuable contribution to the survival analysis community and achieves reasonably good experimental results. There remain concerns that it is not always better than other survival methods, which I think is a reasonable concern although I wouldn't expect such a method (or just about any other method) to be uniformly better/the "best" at every dataset.